# The quantum perfect fluid in 2D

**Aurélien Dersy[1⋆], Andrei Khmelnitsky[2†] and Riccardo Rattazzi[3‡]**

**1** Department of Physics, Harvard University, 02138 Cambridge, MA, USA
**2** Department of Physics, Imperial College, London, United Kingdom
**3** Theoretical Particle Physics Laboratory (LPTP), Institute of Physics,
École Polytechnique Fédérale de Lausanne, 1015 Lausanne, Switzerland

⋆ adersy@g.harvard.edu , † andrew.khmelnitskiy@gmail.com , ‡ riccardo.rattazzi@epfl.ch

## Abstract

We consider the field theory that defines a perfect incompressible 2D fluid. One distinctive property of this system is that the quadratic action for fluctuations around the ground state features neither mass nor gradient term. Quantum mechanically this poses a technical puzzle, as it implies the Hilbert space of fluctuations is not a Fock space and perturbation theory is useless. As we show, the proper treatment must instead use that the configuration space is the area preserving Lie group $S$Diff. Quantum mechanics on Lie groups is basically a group theory problem, but a harder one in our case, since $S$Diff is infinite dimensional. Focusing on a fluid on the 2-torus $T^2$, we could however exploit the well known result $S\text{Diff}(T^2) \sim SU(N)$ for $N \to \infty$, reducing for finite $N$ to a tractable case. $SU(N)$ offers a UV-regulation, but physical quantities can be robustly defined in the continuum limit $N \to \infty$. The main result of our study is the existence of ungapped localized excitations, the vortons, satisfying a dispersion $\omega \propto k^2$ and carrying a vorticity dipole. The vortons are also characterized by very distinctive derivative interactions whose structure is fixed by symmetry. Departing from the original incompressible fluid, we constructed a class of field theories where the vortons appear, right from the start, as the quanta of either bosonic or fermionic local fields.

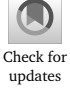

# 1 Introduction

Symmetries play a central role in the characterization of the universality classes of infinite systems. When non-linearly realized, or spontaneously broken, symmetries play in some sense an even greater role. That is because of Goldstone theorem [1,2] in all its variants, classical or quantum and relativistic or non-relativistic, which controls the occurrence of soft modes as well as the structure of their interactions. The latter, in particular, are controlled by field derivatives in such a way that they are weak at low momentum, while higher order effects are organized in a systematic derivative expansion. Systems with spontaneously broken continuous symmetries thus ideally implement the concept of Effective Field Theory (EFT) [3].

In its relativistic invariant incarnation, that is on backgrounds respecting the Poincaré group, Goldstone theorem is particularly neat and powerful [4,5]. The set of light degrees of freedom is in one to one correspondence with the broken symmetry generators, while Lorentz invariance nails their dispersion relation to the light cone and strongly constrains their interactions. The chiral Lagrangian of QCD mesons beautifully concretizes the concept, with the added illustrative benefits stemming from the presence of small breaking effects, (quark masses and the fine structure constant) and from the relevance of topology [6].

The cases where relativistic invariance, boosts in particular, is also spontaneously broken correspond instead to systems at finite density. Intuitively that is because at finite density there exists a preferential inertial frame. Here the implications of Goldstone theorem are less tight and consequently the zoology of options is much richer. In particular the number of mandated soft degrees of freedom is not in correspondence with the number of broken

generators, and is actually normally smaller [7, 8]. Moreover the latter are not necessarily spinless bosons, but they can have any helicity (see e.g. [9]), and even be composed of two ungapped fermions, like it happens in Fermi liquids [10]. Similarly the dispersion relation is not fixed and frequencies can range from linear or quadratic in momentum to gapped, with the gap completely fixed by group theory constraints [11–14]. Nonetheless, beside this variety, the long wavelength fluctuations of basic quantities like energy and charge density is always controlled by ungapped modes. The fact that sound waves universally satisfy an ungapped dispersion relation is just a simple consequence of their being associated with the spontaneous breakdown of the Poincaré (or Galilei) symmetry.

One remarkable implication of what we just outlined is that finite density systems can be classified according to patterns of spontaneous symmetry breaking [15]. Focusing on systems at (virtually) zero temperature, the resulting universality classes should expectedly be Quantum Field Theories whose low energy quanta correspond to the soft hydrodynamic modes. Such universality classes should tell us what states of matter are at all possible according to the broad principles of relativity and quantum mechanics. Developing this knowledge may perhaps sound academic when aiming at systems that can be concretely created in a laboratory, where besides the grand principles, a reality just made of electrons and nuclei also matters. However if one considers the early stages of the universe, of which we still know little, it seems sensible to entertain the possibility for more exotic dynamics, and thus ask what properties could the cosmic medium then have according to basic principles.

This paper is devoted to the perhaps most obvious finite density system, the perfect fluid. Its characterization in terms of symmetries and its classical Lagrangian description have been discussed multiple times in the literature (see for instance [16–19]). However, the quantum description of the perfect fluid poses basic conceptual issues, as was discussed in [20] without reaching any firm conclusion. Indeed one option left open by that study is that the perfect fluid does not make sense as an Effective Quantum Field Theory, that is as a closed quantum system at zero temperature. That result would match the empirical observation that fluids transition to other phases, for instance superfluid or solid phases, when cooled to sufficiently low temperature. But another option could be that the perfect quantum fluid does make sense, though in a very non-trivial manner, such that it is not immediate to identify a physical system in its universality class. In this paper we shall reconsider the problem and explicitly construct a quantum mechanical system that consistently realizes the dynamics of the perfect fluid in two spacial dimensions. As we shall see, in accord with the second option mentioned above, the result is quite non-trivial when comparing to the classical field theory description given for instance in ref. [20]. Our construction, honestly, does not address all the questions, but it provides concrete results and predictions, building upon which further progress can hopefully be made.[1]

We would now like to illustrate the difficulty posed by the quantum mechanical treatment of the perfect fluid. To help make our point, we shall first review the symmetry based characterization of the simplest finite density systems.

## 1.1 Superfluids and solids

From a symmetry perspective, the simplest finite density system is undoubtedly the (relativistic) superfluid. Besides the Poincaré group $ISO(3,1)$ generated by space-time translations $P_\mu$, Lorentz boosts $K_i$ and rotations $J_i$, the corresponding system is endowed with an internal $U(1)_Q$ symmetry (either compact or not) generated by a charge $Q$. The superfluid state can then be abstractly characterized as a configuration that realizes the spontaneous breaking

---

[1]A significant part of this paper, in particular sections 2-3 and 6, is based on results derived in a previous abandoned project [21]. The broader picture we have now developed gives us sufficient confidence to present all our results.

$ISO(3,1) \times U(1)_Q \rightarrow ISO(3) \times H'$ [22]. Here $ISO(3)$ is the euclidean group of rotations and translations in 3D while $H'$ is a residual time translation generated by $H' = P_0 - \mu Q$. This pattern of symmetry breaking dictates a single soft Goldstone mode. The long distance effective description can be constructed by considering the theory of a single scalar field $\varphi$ that shifts under $U(1)_Q$: $\varphi(x) \rightarrow \varphi(x) + \alpha$. The most general lowest derivative Lagrangian is given by the most general function of the $ISO(3,1) \times U(1)_Q$ invariant $X \equiv \partial_\mu \varphi \partial^\mu \varphi$

$$\mathcal{L} = F(X). \tag{1}$$

A superfluid state is then obtained by considering the solution $\varphi = \mu t$, which realizes $ISO(3,1) \times U(1)_Q \rightarrow ISO(3) \times H'$, and by expanding around it: $\varphi(x) = \mu t + \pi(x)$. The equation of state is in one-to-one correspondence with the form of the function $F$, as the latter controls the energy momentum tensor. However $F$ also controls the properties and the interactions of the fluctuation $\pi$. Therefore the equation of state controls not only the dispersion relation of $\pi$ but also the scattering amplitudes of the $\pi$ quanta at lowest order in the derivative expansion. In particular, the dispersion relation has generically the form $\omega = c_s k$, with $0 < c_s < 1$, while the interactions are controlled by derivatives. At the quantum level the low energy states of the system are therefore given by a Fock space of weakly coupled quanta.

On the background solution $\varphi = \mu t$, time translations are non-linearly realized and $\varphi$ is in one-to-one correspondence with $t$. The scalar field of a superfluid can thus be viewed as a *clock*. That directly connects the superfluid to the simplest incarnation of inflation, where a slow rolling scalar field clocks the evolution of the universe. In fact slow roll inflation can be viewed as a slightly deformed superfluid where the shift symmetry $\varphi(x) \rightarrow \varphi(x) + \alpha$ is explicitly broken by an approximately flat potential.

Another relevant simple example is provided by a homogeneous solid [18, 23]. Three scalars $\phi^I$ ($I = 1, 2, 3$) now play the role of *rulers* in space. That is realized by assuming the internal shift symmetry $\phi^I \rightarrow \phi^I + \alpha^I$ and a background configuration linear in the spacial coordinates: $\phi^I = \mu^I_j x^j$. Further assuming internal rotational symmetry $\phi^I \rightarrow R^I_J \phi^J$, with $R^I_J \in SO(3)$, and taking a solution with $\mu^I_j = \mu \delta^I_j$, isotropy is added to homogeneity. The internal symmetry in the latter case is the euclidean group $ISO(3)$ in $\phi$-space and the solution operates the spontaneous breaking of Poincaré $\times ISO(3)$ to a *diagonal $ISO(3)'$*. At the lowest derivative order an invariant action is constructed through the bilinears $X^{IJ} = \partial_\mu \phi^I \partial^\mu \phi^J$. Focusing on the special case of an isotropic solid, the dependence is further limited to invariants of the internal $SO(3)$, which can be taken to be $I_k \equiv \mathrm{Tr} X^k$ for $k = 1, 2, 3$. The most general Lagrangian for a relativistic homogeneous and isotropic solid can then be written as

$$\mathcal{L} = F(I_1, I_2, I_3). \tag{2}$$

The fluctuations around the background are described by a field $\pi^I(x)$ transforming as a vector under the residual $ISO(3)'$: $\phi^I(x) = \mu x^I + \pi^I(x)$. At the quantum level the low energy states of the system are again given by a Fock space of weakly coupled quanta. One main difference with respect to the superfluid is that now phonons come in three polarizations, one longitudinal and two transverse, with linear dispersion relations (and in general different speeds of sound for the longitudinal and transverse modes).

## 1.2 The fluid and its puzzling quantum mechanics

The natural next case to consider is the subject of this paper: the perfect fluid [18]. This can be viewed as a very special limit of the solid, where the internal symmetry $ISO(3)$ is extended to the full group of volume preserving diffeomorphisms $S$Diff

$$\phi^I \rightarrow f^I(\phi), \qquad \left| \frac{\partial f}{\partial \phi} \right| \equiv \det \left( \frac{\partial f^I}{\partial \phi^J} \right) = 1. \tag{3}$$

At the lowest derivative order, the Lagrangian is now constrained to purely depend on the determinant of $X^{IJ} = \partial_\mu \phi^I \partial^\mu \phi^J$

$$\mathcal{L} = F(X), \qquad X \equiv \det(X^{IJ}). \tag{4}$$

Contact with the standard description of relativistic fluids (see for instance the treatment in [24]) is then made by first defining the entropy current $J_\mu$ [19] via

$$J_\mu dx^\mu \equiv * \frac{1}{6} \epsilon_{IJK} d\phi^I \wedge d\phi^J \wedge d\phi^K. \tag{5}$$

$J^\mu$ is conserved ($\partial_\mu J^\mu = 0$) by construction, that is independently of the equations of motion. One also has $X = -J_\mu J^\mu$ so that $J^\mu = \sqrt{X} u^\mu$, with $u^\mu$ a unit norm 4-vector ($u^\mu u_\mu = -1$): $\sqrt{X}$ and $u^\mu$ have then the natural interpretation of respectively entropy density and fluid 4-velocity. The latter interpretation is also consistent with the form of the energy momentum tensor, which is indeed found to be that of a relativistic fluid with 4-velocity $u^\mu$

$$T^{\mu\nu} = (p + \rho)u^\mu u^\nu + p\, \eta^{\mu\nu}, \tag{6}$$

where energy density and pressure are respectively given by $\rho = -F(X)$ and $p = F(X) - 2F'(X)X$.[2] Finally one finds that the equations of motion dictated by eq. (4) are precisely equivalent to energy momentum conservation $\partial_\mu T^{\mu\nu} = 0$. Together with the trivial equation $\partial_\mu J^\mu = 0$ this shows the full equivalence of our system to a relativistic fluid in its ordinary description.

The effective field theoretic description of (relativistic) fluids offers an alternative perspective on standard results and also a systematic methodology to address concrete physics questions. For instance, Kelvin's Theorem, which basically states the convective conservation of vorticity (see [18] for a relativistic discussion), here coincides with Noether's theorem for the local currents of the $S$Diff symmetry. Instances of applications include the study of the effects of global symmetries and of their anomalies [19, 25], the systematic description of vortex-sound interactions [26], the computation of relativistic corrections to sound emission from turbulent flow [26] and also the non-linear treatment of cosmological density perturbations [27].

All the concrete results so far have been obtained treating eq. (4) classically. It is thus natural to ask where a quantum treatment would take us. In the case of solids and superfluids, as we have mentioned, quantum mechanics leads us to the known grounds of a weakly interacting QFT for the phonon quanta. However, as we shall now review, this ordinary route is barred in the case of the fluid. This very fact is what makes the issue of the quantized perfect fluid interesting: does it make sense? And if it does, what is it?

The basic novelty of fluids, compared with the other finite density systems, is appreciated by studying fluctuations around the homogeneous isotropic solution: $\phi^I = x^I + \pi^I(x)$. Expanding eq. (4) at quadratic order, one finds a Lagrangian $\mathcal{L}^{(2)}$ of the form

$$\mathcal{L}^{(2)} \propto \dot{\pi}^I \dot{\pi}^I - c_s^2 (\partial_I \pi^I)^2 \equiv (\dot{\pi})^2 - c_s^2 (\nabla \cdot \pi)^2, \tag{7}$$

where, given $\pi^I$ transforms as a vector field under the unbroken $ISO(3)$, we used 3D vector calculus notation. We can now decompose $\pi$ into longitudinal and transverse components, $\pi = \pi_\perp + \pi_\parallel$, satisfying respectively $\nabla \cdot \pi_\perp = 0$ and $\nabla \wedge \pi_\parallel = 0$. By eq. (7) the corresponding waves then satisfy the dispersion relation

$$\omega_\parallel(k) = c_s k, \qquad \omega_\perp(k) = 0. \tag{8}$$

---

[2]$\rho$, $p$ and entropy density $\sqrt{X}$ are all in one-to-one correspondence, given ours is by construction a purely mechanical system, where entropy cannot change. In fact, we could also simply drop the interpretation of $\sqrt{X}$ as the entropy density, and simply view our system as field theory constructed on symmetry principles.

This result expresses the well known fact that in a classical fluid only longitudinal waves propagate with a finite sound speed. The degenerate dispersion relation of transverse modes corresponds to the existence of non propagating stationary vortex configurations. An example of that is a vortex flow with cylindrical symmetry and suitable profile for $\rho$, $p$ and the velocity in the angular direction $v_\theta$.[3] The vanishing, for any wave vector $k$, of the proper frequencies of transverse modes is also at the basis of the phenomenon of turbulence. Indeed, given transverse motions do not have a frequency gap growing with their wave vector, an external slow and long wavelength perturbation can "resonantly" excite transverse modes with arbitrarily large wave vector.

The absence of a gradient term for $\pi_\perp$ in eq. (7) is a direct consequence of the $S$Diff symmetry. In fact this result holds to all orders in the fluctuation when expanding around $\phi^I = x^I$. That is because time independent transverse modes[4] $\pi_\perp(\boldsymbol{x}, t) = \pi_\perp(\boldsymbol{x})$ correspond to the linearized action of $S$Diff on the background $\phi^I = x^I$. In other words, $\pi_\perp(\boldsymbol{x})$ are just the Lie parameters of $S$Diff

$$\phi^I = x^I \longrightarrow f^I(\phi) = f^I(\boldsymbol{x}) \equiv x^I + \pi_\perp^I + O(\pi_\perp^2). \tag{9}$$

The Lagrangian is exactly invariant for any $f \in S$Diff, and thus order by order in the Lie parameter $\pi_\perp$. At lowest order this gives eq. (7).

The peculiarity of the fluid (e.g. turbulence) associated with $\omega_\perp(k) = 0$ becomes even more dramatic if we want to treat it as a quantum field theory. That is because $\omega_\perp(k) = 0$ implies that the set of transverse modes does not reduce to a system of weakly coupled harmonic oscillators. In other words, the absence of any gradient or potential contribution to the action of the transverse modes implies that the wave functionals describing the ground state and the excited states are not *localized* in field space. That is unlike what happens instead for fields with ordinary quadratic action. The relation existing between the fluid and any other system with ordinary quadratic action bears a close analogy with that existing between a free particle on the line, $\mathcal{L} = m\dot{q}^2/2$, and the harmonic oscillator $\mathcal{L} = m\dot{q}^2/2 - m\omega^2 q^2/2$. While in the latter system the energy eigenstates are localized in position space, the eigenstates of the free particle are fully delocalized. Compactifying the line on a circle does not significantly change the story: the eigenstates of the free particle are fully delocalized on the circle and their properties cannot be described meaningfully by perturbing around a point in $q$ space. The resulting Hilbert spaces and spectrum are thus vastly different. In particular the Hilbert space of the free particle is not a Fock space constructed from a vacuum $|0\rangle$ by acting with creation operators. Similarly, in the case of the fluid, the Hilbert space does not consist of a Fock space of weakly coupled transverse and longitudinal phonons. Hence the question: what should we make quantum mechanically of such a bizarre field theory?

To our knowledge, this question was studied twice in the recent literature [20,28], though it had already surfaced, in a different form, in the famous 1941 paper on superfluids by Landau [29]. We believe none of these studies fully addressed it. The approach of ref. [20] is to slightly deform the fluid into a soft (low shear) solid to obtain a manageable system. That is done by adding to eq. (4) a small term breaking $S$Diff down to $ISO(3)$. Its main effect is the appearance of a small velocity $c_\perp$ for the transverse modes associated with a transverse gradient term $(\boldsymbol{\nabla} \wedge \boldsymbol{\pi})^2$ in the action. The idea would be to construct the effective QFT at finite $c_\perp$ and see what happens when $c_\perp \to 0$. However, by studying the scattering of the now propagating transverse quanta, one finds, perhaps not unexpectedly, that the strength of their interaction grows like an inverse power of $c_\perp$. As the interaction is derivative, that implies that the momentum cut-off scale $\Lambda$, above which the system is out of any perturbative control,

---

[3]In the limit of non-relativistic velocities $F = ma$ simply dictates $v_\theta^2(r) = (r/\rho(r))dp(r)/dr$.

[4]Throughout the paper we indicate the spacial coordinates in boldface, $\boldsymbol{x}$, whenever we need to distinguish them from the full space-time ones and whenever we need to emphasize they form a vector.

goes to zero with a power of $c_\perp$. The limitation of the approach of ref. [20] is then that it carves out a controllable and weakly coupled QFT in the very long wavelength regime, while the interesting fluid dynamics lies at finite wavelength. Ref. [28] proposed to remedy this state of things by limiting observables to the set of $S$Diff invariant operators. That seems sensible, also because, that set includes all the standard quantities $\rho$, $p$ and $\boldsymbol{v}$. The idea is that these quantities, unlike the $S$-matrix for transverse modes, will behave smoothly in the limit $c_\perp \to 0$. This idea is inspired by the analogy with 2D $\sigma$-models, where the wild IR fluctuations in the field are projected out by gauging the symmetry and admitting only invariants as observables. However, on the basis of the discussion at the end of section 3, we think the case of the 2D $\sigma$-model and the present one are quite different. In the former, the gauging of the global symmetry eliminates only a small set of global degrees of freedom, while if properly carried out in the present case, it would eliminate all the transverse modes, leading to a system that is indistinguishable from a superfluid. Indeed the results of this paper, which we believe are based on a proper quantization of the fluid, do not correspond to those of ref. [28]. We will comment on the differences among the two approaches in section 5.1. As concerns finally Landau, the view expressed in his paper [29] is that the transverse modes are simply gapped at the cut-off of the theory and hence play no role in the low energy EFT. Landau's argument is not based on an explicit computation, but on the expectation that variables described by a non-commuting algebra ($S$Diff) are discretized in quantum mechanics, like it happens for angular momentum. Landau was perhaps also guided in this deduction by the empirical observation that no light mode of this sort is observed in superfluids.

## 1.3 The incompressible limit

According to our discussion, the transverse modes are at the heart of the difficulty, while the longitudinal ones are secondary. To simplify the discussion and zoom on the essential problem it does make sense to do away with the latter class of modes. That can indeed be done by considering the dynamical regime of low velocities ($v \ll c_s$) where the fluid behaves as incompressible, and sound waves are not emitted. As discussed in [20], by integrating out the longitudinal fluctuation $\pi_\parallel$, one obtains an effective Lagrangian written as an expansion in time derivatives and suitable for the incompressible regime. The Lagrangian is spacially non-local, but at low velocities, where retardation effects are small, this does not pose any problem. One could also concretely define this regime by compactifying space on a manifold (for instance a torus or a sphere) of size $R$ and by considering the limit of very low energies. The field $\pi_\parallel$ decomposes into a tower of Kaluza-Klein modes with frequency/energy gap $\sim c_s/R$, while the $\pi_\perp$ modes remain ungapped. At energies $\ll c_s/R$ the longitudinal modes can be integrated out and the resulting effective description corresponds to a slowly moving incompressible fluid whose degrees of freedom coincide with the transverse modes. Notice that $E \ll c_s/R$ corresponds to time scales $T \gg R/c_s$ and thus to velocities $R/T \ll c_s$, that is to a slowly moving fluid. Notice also that, given $c_s < 1$, the incompressible regime is necessarily non-relativistic.

After eliminating the compressional mode, the dynamics is described by fields $\phi^a(\boldsymbol{x}, t)$ subject to the constraint $\det(\partial\phi/\partial x) = 1$. The dynamical variables consist then of the (time dependent) volume preserving maps between $x$-space and $\phi$-space $\phi : \mathbf{X} \to \mathbf{\Phi}$, see also the discussion in section 3. The condition on the Jacobian ensures that $\phi^a(\boldsymbol{x}, t)$ is bijective, so it can be inverted, thus taking $x^I(\phi, t)$ as dynamical variables on $\phi$-space. The fluid velocity is then simply given by $v^I = \partial x^I(\phi, t)/\partial t \equiv \dot{x}^I$, while at lowest order in the time derivative expansion, see [20], the effective Lagrangian resulting from eq. (4) reduces to the well known result

$$\mathcal{L} = F(1 - \boldsymbol{v}^2) = -F'(1)v^2 + O(v^4) \equiv \frac{\rho_m}{2}v^2 + O(v^4), \qquad \rho_m = \rho(1) + p(1), \qquad (10)$$

where we have used that $\rho(X) + p(X) = -2F'(x)$ to define the mass density $\rho_m$. The dynamics of the incompressible fluid can be meaningfully studied by focusing on the leading $O(v^2)$ term. In this paper we shall study the quantum mechanics of this system. Notice that, even though the form of the Lagrangian appears simple, the dynamics is non-trivial because the variables $x^I(\phi, t)$ are constrained. As we shall better discuss later in section 3, the $x^I(\phi, t)$ (and equivalently $\phi^a(\boldsymbol{x}, t)$) are volume preserving mappings between isomorphic spaces, $\mathbf{X} \sim \boldsymbol{\Phi}$, and can thus be viewed as elements of $S$Diff on $\mathbf{X}$. The incompressible fluid can be viewed as a $\sigma$-model with base space $S$Diff. In fact this will be the basis of our approach.

Before proceeding to a technical study of eq. (10) it is useful to try and guess what energy spectrum to expect. In the canonical approach, the first step is to derive the canonical variables and their Hamiltonian. This task is not immediate, given our variables are constrained. While a complete discussion will be given in the next section, it is intuitively clear that the canonical momentum resulting from this procedure will be proportional to the 3-momentum density $\rho_m v$. This implies that, when written in terms of canonical variables, the Hamiltonian will be proportional to $1/\rho_m$. Using dimensional analysis we can then guess what to expect for the energy gap of modes of momentum $k$. If $k$ and $\rho_m$ were the only parameters at hand we would conclude the gap is proportional to $k^5/\rho_m$. More generally for a fluid in $D$-spacial dimensions it would be $k^{D+2}/\rho_m$. On the other hand, the classical result $\omega(k)_\perp = 0$ indicates that for fluids the separation between UV and IR is not clear-cut. So we could consider the possibility that the gap is also controlled by the UV cut-off length $1/\Lambda$. In the extreme case, we could expect the gap to only depend on $\rho_m$ and $\Lambda$, in which case it would be $\Lambda^{D+2}/\rho_m$. If that were the case there would be nothing to talk about: no transverse degree of freedom would survive in the low energy EFT, which would then only feature the longitudinal modes, precisely like one observes in a superfluid at zero temperature. This second option, with its consequences, is what Landau considers to be the correct one in his 1941 paper. Focusing on the 2-dimensional case $D = 2$, we will show in this paper that neither option appears to be the correct one. What we will find is that there exist states with a sort of mixed dispersion relation $\omega(k) \sim \Lambda^2 k^2/\rho_m$. We will name vortons the corresponding quanta.

## 1.4 Outline

We here offer an overview of the content of the paper.

As we have seen, the incompressible fluid is a mechanical system whose dynamical coordinates span the $S$Diff group manifold. If one leaves aside the fact that $S$Diff is infinite dimensional, the incompressible fluid is but one instance in the vast class of mechanical systems describing motion on a group manifold $G$. As the simple case $G = SO(3)$ is just the ordinary mechanical rigid body, these systems generalize the notion of rigid body. It seems fair, in order to attack our problem, to first appreciate how this class of systems works, both classically and quantum mechanically. Section 2 is devoted to that, while in section 3 we shall formally adapt the results to the specific case of a 2D fluid flowing on a square torus $T^2$ where the relevant group is $S$Diff($T^2$). The concrete implementation of the formalism of section 3 at the quantum mechanical level requires to represent $S$Diff($T^2$) on a Hilbert space, which is not an obvious task. In section 4 we address this difficulty, by working on a finite dimensional surrogate of $S$Diff($T^2$). More precisely we make use of an old result establishing that $S$Diff($T^2$) equals a certain $N \to \infty$ limit of $SU(N)$. The truncation of $S$Diff($T^2$) to $SU(N)$, besides allowing calculability through the well known construction of representations of $SU(N)$, also automatically introduces a UV regulator in the form of a spacial lattice. Conceptually that is important in light of the discussion in the previous section: we can now concretely study the effects of a UV-cut off on the dynamics, the spectrum in particular. That is done in section 5, where we work out both kinematics and dynamics. As concerns the spectrum, the basic result is that representations that can be written as tensor products of adjoints are not gapped. The

basic building blocks, corresponding to the adjoint representation, satisfy a quadratic dispersion relation $\omega(k) \sim \Lambda^2 k^2 / \rho_m$ and can be viewed as quanta, which we name vortons, carrying a dipole of vorticity proportional, but orthogonal, to their momentum. All the other representations, for all that we could check, have energy gapped at the scale $\Lambda^4 / \rho_m$, in agreement with Landau's guess. Among these are vortex and antivortex states, which correspond to respectively the fundamental and antifundamental representations. Our construction, beside the vorton spectrum, predicts their interactions. In section 6 we illustrate how this works by computing the simplest instance of $2 \to 2$ vorton scattering and discuss the peculiar form of the result. There we also offer an alternative and purely field theoretic path to the vorton spectrum and Lagrangian.

While we think our results are concrete progress, there are still unsolved aspects. These concern in particular the multiplicity and the statistics of vortons. A summary and an assessment of our results are finally presented in sections 7 and 8.

## 2 The general rigid body

### 2.1 Classical rigid body

The system we want to study can be viewed as a generalization of the quantum mechanical rigid body that describes the rotational modes of molecules. Classically each configuration of the rigid body is fully specified by the $SO(3)$ rotation that relates a static laboratory frame to a frame that is fixed with respect to the body and rotating with it. The space of configurations of the rigid body then coincides with the $SO(3)$ group manifold. Stated that way, the notion of rigid body is readily generalized to a more abstract mechanical system whose configuration space is a general compact and connected Lie group $G$ [30, 31].

Systems whose configuration space is a whole Lie group can be viewed, in turn, as a special sub-class of systems whose configuration space is a coset $G/H$, with $H$ a subgroup of $G$. The generalized rigid body then simply corresponds to a coset $G/H$ where $H$ is the limiting subgroup consisting of just the identity element: $H = \{e\}$. As $G/H$ implements a (partially) non-linear realization of the symmetry $G$, systems with this configuration space are central in field theory to describe the low energy dynamics of spontaneous symmetry breaking. The formalism to construct the most general dynamics for such systems was first given in refs. [4, 5], but a more pedagogical presentation can be found for instance in ref. [32]. The generalized rigid body is a mechanical system, so it can be viewed as a field theory in $1 + 0$ dimensional space time. It is thus straightforward to adapt to it the methodology of the above references, as we shall now do.

To begin, we can parametrize the configuration space $G$ through any faithful unitary matrix representation of its abstract elements $g(\pi)$

$$U_{g(\pi)} \equiv U(\pi) = e^{\pi_A T^A}, \qquad A = 1, \dots, N_G, \tag{11}$$

where $N_G$ is the dimension of $G$, $T^A = -T_A^\dagger$ are anti-hermitian matrices representing a basis of its Lie algebra $\mathfrak{g}$ and $\pi_A$ are dynamical Lie parameters.[5] We shall assume the normalization $\mathrm{Tr}(T^A T_B^\dagger) = \delta_B^A$ and use a notation where indices are raised and lowered by Hermitian conjugation and by matrix inversion.

$G$ can be equivalently realized through its left action on $U$

$$G_L : \quad U(\pi) \to U_g U(\pi) \equiv U(f_g^L(\pi)), \qquad g \in G, \tag{12}$$

---

[5]In the field theoretic constructions the $\pi_A$ represent the soft modes dictated by Goldstone theorem.

which we label as $G_L$, and through its right action

$$G_R : U(\pi) \to U U_g^{-1} \equiv U(f_g^R(\pi)), \qquad g \in G, \tag{13}$$

which we label as $G_R$. The functions $f_g^L$ and $f_g^R$ offer then explicit realizations on the the $\pi$ coordinates of respectively $G_L$ and $G_R$. The combined action of $G_L \times G_R$ is then simply $U \to U_{g_L} U U_{g_R}^{-1}$. We will assume $G_L$ is exact, while $G_R$ will in general be explicitly broken.

According to the above definitions, we can also view $U(\pi)$ as parametrizing the coset $G_L \times G_R / G_D$ where $G_D$ is the diagonal $G$ subgroup, acting on $U(\pi)$ as

$$G_D : U(\pi) \to U_g U U_g^{-1}, \qquad g \in G, \tag{14}$$

so that $\pi_A$ transforms linearly under $G_D$.

We will use indices $\bar{a}$ and $a$ to label generators in the Lie Algebras of respectively $G_L$ and $G_R$, while $A, B, \dots$ label the adjoint of $G_D$, in particular the dynamical variables $\pi_A$. All these set of indices run from 1 to $N_G$.

The main object to build the dynamics is the Cartan form [33]

$$\Omega \equiv U^{-1} dU = T^a f_a^A d\pi_A, \tag{15}$$

which by construction takes values in the Lie algebra. Notice that $f_a^A \equiv f_a^A(\pi)$ are fully determined by the structure constants and can be viewed as a $\pi$-dependent $n$-bein. By construction $\Omega$ is a singlet of $G_L$ and an adjoint of $G_R$.

Indicating the time derivative of $U$ by $\dot{U}$, the angular *velocity* matrix $U^{-1}\dot{U}$ can be written according to eq. (15) as

$$U^{-1}\dot{U} = T^a f_a^A \dot{\pi}_A \equiv T^a d_a. \tag{16}$$

By the transformation properties of $\Omega$ the $d_a$'s are all $G_L$ singlets and form an adjoint of $G_R$. Considering the system classically, the most general $G_L$-invariant two derivative Lagrangian is then[6]

$$\mathcal{L} = \frac{1}{2} I^{ab} d_a d_b \equiv \frac{1}{2} g^{AB} \dot{\pi}_A \dot{\pi}_B, \tag{17}$$

where the parameters $I^{ab} = I^{ba}$ are the entries of a generalized inertia tensor and

$$g^{AB}(\pi) \equiv I^{ab} f_a^A f_b^B, \tag{18}$$

represents the most general $G_L$-invariant metric on the $G$ group manifold. Notice that $G_R$ is not a symmetry for generic $I^{ab}$, and, relatedly, $g^{AB}$ is not $G_R$-invariant. Only for $I^{ab} \propto \delta^{ab}$ is $G_R$ a symmetry. It is useful to visualize this state of things by viewing $U(\pi)$ as a matrix describing the relative orientation of the static laboratory frame with respect to the rotating body frame. In this view the action of $G_L$ and $G_R$ describe changes of respectively laboratory and body frame. The former, $G_L$, corresponds to a symmetry, while the fate of the latter, $G_R$, depends on the "shape" of the body via the inertia tensor $I^{ab}$.

The abstract group element $g(\pi)$ admits, in particular, the adjoint representation, whose elements can be written in terms of the generic representation $U(\pi)$ via

$$\Delta_a^{\bar{b}}(\pi) \equiv \text{Tr}(T_a^\dagger U^{-1} T^{\bar{b}} U). \tag{19}$$

In other words, the matrix $U(\pi)$ in eq. (11) coincides with $\Delta_a^{\bar{b}}(\pi)$ when the adjoint representation for $T^A$ is chosen.[7] Notice that $\Delta_a^{\bar{b}}$ is a real orthogonal matrix, whose inverse

---

[6]This is just the reduction to a mechanical system of the general result in ref. [32].

[7]The adjoint may or may not be a faithful representation. For the sake of the picture we offer in this paragraph, we assume the adjoint is faithful, like for the mechanical rigid body, where $G = SO(3) \sim SU(2)/Z_2$, or, more generally, for $SU(N)/Z_N$.

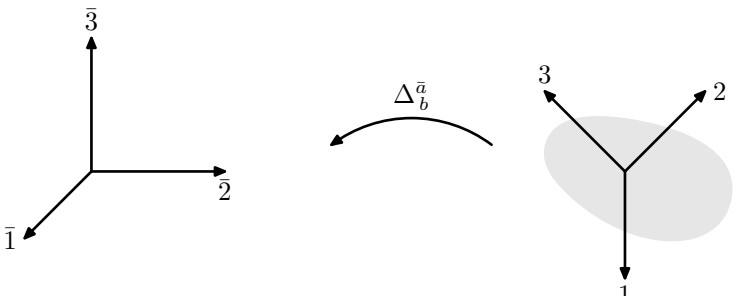

Figure 1: The configuration of the rigid body is described by the $SO(3)$ mapping between a still laboratory frame and a rotating body frame.

$(\Delta^{-1})^a_{\bar{b}} \equiv \Delta_{\bar{b}}{}^a$ is simply given by the transposed: $(\Delta^{-1})^a_{\bar{b}} = \Delta_{\bar{b}}{}^a$. Indicating the Lie parameters of $G_L$ and $G_R$ by respectively $\alpha_L$ and $\alpha_R$, the action of $G_L \times G_R$ on $\Delta_{a}^{\bar{b}}$ is thus

$$\Delta_a^{\bar{b}} \Rightarrow \Delta_{\bar{c}}^{\bar{b}}(\alpha_L)\Delta_d^{\bar{c}}(\Delta^{-1}(\alpha_R))^d_a \equiv \Delta_{\bar{c}}^{\bar{b}}(\alpha_L)\Delta_d^{\bar{c}}\Delta_a{}^d(\alpha_R). \tag{20}$$

The system (see Fig. 1) can then conveniently be represented as a rigid body rotating in the adjoint representation vector space, with $\Delta_a^{\bar{b}}$ the change of basis from the laboratory frame (labelled by indices $\bar{b}$) to the body frame (labelled by indices $a$). The moment of inertia $I^{ab}$ is referred to the latter frame. The $d_a$ measure the angular velocity with respect to the same frame.

Considering abstract group elements $g(\alpha_L)$ and $g(\alpha_R)$, with $\alpha_{L\bar{b}}$ and $\alpha_{Rb}$ infinitesimal Lie parameters, the infinitesimal action of $G_L$ and $G_R$ on $\pi$ is given by[8]

$$U(\alpha_L)U(\pi) \simeq \left(\mathbb{1} + \alpha_{L\bar{b}}T^{\bar{b}}\right)U(\pi) \simeq U(\pi - \alpha_{L\bar{b}}\delta_L^{\bar{b}}\pi) \quad \Rightarrow \quad \delta_L^{\bar{b}}\pi_A = -\Delta_a{}^{\bar{b}}f_A^a, \tag{21}$$

$$U(\pi)U(\alpha_R)^{-1} \simeq U(\pi)\left(\mathbb{1} - \alpha_{Rb}T^b\right) \simeq U(\pi - \alpha_{Rb}\delta_R^b\pi) \quad \Rightarrow \quad \delta_R^b\pi_A = f_A^b, \tag{22}$$

with $f_A^b$ the inverse of $f_a^B$, that is $f_A^a f_a^B = \delta_A^B$ and $f_C^a f_b^C = \delta_a^b$. The "Noether" charges for $G_L$ and $G_R$ can then be written as

$$L^{\bar{a}} \equiv \frac{\partial\mathcal{L}}{\partial\dot{\pi}_A} \times \delta_L^{\bar{a}}\pi_A = -I^{ab}d_b\Delta_a{}^{\bar{a}}, \tag{23}$$

$$R^a \equiv \frac{\partial\mathcal{L}}{\partial\dot{\pi}_A} \times \delta_R^a\pi_A = I^{ab}d_b. \tag{24}$$

Although, for generic $I^{ab}$, only the $L^{\bar{a}}$ are conserved, the Poisson brackets of $L^{\bar{a}}$ and $R^a$ realize nonetheless the Lie algebra of $G_L \times G_R$

$$\{L^a, L^b\} = f^{ab}{}_c L^c, \qquad \{R^a, R^b\} = f^{ab}{}_c R^c, \qquad \{L^a, R^b\} = 0, \tag{25}$$

with the structure constants defined by $[T^a, T^b] = f^{ab}{}_c T^c$. That follows because $\partial\mathcal{L}/\partial\dot{\pi}^A \equiv p^A$ is the canonical momentum and because $\delta_L\pi$, $\delta_R\pi$ are infinitesimal realizations of $G$. Eqs. (23,24) can also be written as (we recall $\Delta_{\bar{b}}{}^a \equiv (\Delta^{-1})_a^{\bar{b}}$)

$$R^a = -\Delta_{\bar{b}}{}^a L^{\bar{b}}, \qquad d_a = I_{ab}^{-1}R^b. \tag{26}$$

---

[8]Notice that $\delta_L^{\bar{b}}\pi_A$ defined with a minus sign offers the proper representation of $G$ on the space of functions of $\pi$ which reads $g : A(\pi) \rightarrow A(f_{g^{-1}}^L(\pi))$. The same comment applies to our definition of $\delta_R^b\pi_A$.

By the first equation, the right charges $R^a$, coincide, up to a sign, with the projection of the left charges $L^{\bar{a}}$ on the axes of the rigid body. As the $L^{\bar{a}}$ are conserved, the time dependence of the $R^a$ is directly controlled by the time evolution of the orientation of the body with respect to the $L^a$. Moreover, by eq. (17) and by the second identity in eq. (26), the Hamiltonian takes the form ($p^A = \partial \mathcal{L}/\partial \dot{\pi}_A$)

$$H = \frac{1}{2} g_{AB} p^A p^B = \frac{1}{2} I^{ab} d_a d_b = \frac{1}{2} I^{-1}_{ab} R^a R^b, \qquad g_{AB} \equiv (g^{-1})_{AB}. \tag{27}$$

The Hamiltonian is a quadratic form in the components of the charges projected onto the axes of the rigid body. By the conservation of $L^{\bar{a}}$ and by eq. (26), the time evolution of the configuration coordinates $\Delta_{\bar{b}}^{\ a}$ can equivalently well be described by the time evolution of $R^a$, which is in turn directly controlled by the Lie algebra

$$\dot{R}^b = \{H, R^a\} = I^{-1}_{ac} f^{ab}_{\ \ d} R^c R^d \equiv -f^{ab}_{\ \ d} \Omega_a R^d. \tag{28}$$

Here, according to the sign choice in eqs. (23,24), we identified $d_a$ with minus the angular velocity: $\Omega_a \equiv -d_a$. For $G = SO(3)$, eqs. (26,27,28) summarize the classic results in the mechanics of the rigid body.

The system we have been considering has $N_G$ pairs ($\pi_A$, $p^A$) of canonical variables. The structure of the Hamiltonian entails however a number of conservation laws, which constrain the flow to lie on a submanifold of the phase space. The crucial property, in that respect, is that $H$ is purely a function of the right charges $R^a$. Then, while for arbitrary $I^{ab}$ the $G_R$ symmetry may be fully broken, the $G_R$ Casimirs still have vanishing Poisson brackets with $H$. Indeed, by eq. (26), the $G_R$ Casimirs coincide with the $G_L$ ones. The number of Casimirs equals the dimension $C_G$ of the Cartan subalgebra of $G$, so that, under time evolution, the $R^a$ charges span a manifold of dimension $N_G - C_G$. One basic result in the classification of Lie algebras is that $N_G - C_G$ is an even number (see e.g. [34]). That is simply because operators that raise and lower the Cartan algebra eigenvalues come in pairs. Indeed by a result in the theory of Poisson manifolds, known as the splitting theorem [35], the resulting constrained flow of the $R^a$ does correspond to a Hamiltonian flow in a phase space consisting of $(N_G - C_G)/2$ pairs of canonical variables. For instance, in the case of $SO(3)$, fixing $R^2 = R_1^2 + R_2^2 + R_3^2$, one can explicitly check that by solving for $R_3$ in terms of $R_{1,2}$. One then reduces to a system of one canonical pair $(p, q)$, consisting of functions of $R_1$ and $R_2$ (and of the constant Casimir $R^2$).

## 2.2 Quantum mechanical rigid body

The quantum mechanical rigid body is constructed starting from the results in the previous section.[9] Indeed, see eq. (17), the system corresponds to a particle freely moving on a manifold with metric $g^{AB}$, whose quantization is straightforward. The Hilbert space $\mathcal{H}$ consists of the space of square integrable functions $\psi(\pi)$ on $G$. The group action on $\mathcal{H}$ is represented by $\psi(\pi) \to \psi(f^L_{g^{-1}}(\pi))$ and $\psi(\pi) \to \psi(f^R_{g^{-1}}(\pi))$ for respectively $G_L$ and $G_R$. Given two wave functions $\psi_1(\pi)$ and $\psi_2(\pi)$, the scalar product can then be defined by the unique (up to a constant) $G_L \times G_R$ invariant quadratic form

$$\langle \psi_1 || \psi_2 \rangle \equiv \int d\mu_\pi \, \psi_1(\pi)^* \psi_2(\pi), \tag{29}$$

with $d\mu_\pi$ is the Haar measure on $G$, which by eq. (18) can be written as

$$d\mu_\pi \equiv \frac{\sqrt{\det g}}{\sqrt{\det I}} \, d^{N_G}\pi = \det(f) \, d^{N_G}\pi \equiv f \, d^{N_G}\pi. \tag{30}$$

---

[9]This part, as well as the discussion on the gauging of the left symmetry in section 3, greatly benefitted from sharp criticism by Ben Gripaios.

Upon the identification $p^A = -i\partial/\partial\pi_A \equiv -i\partial^A$, the Hamiltonian operator is given by the $G_L$ invariant 2-derivative quadratic form

$$\langle\psi_1|H|\psi_2\rangle \equiv \frac{1}{2}\int d\mu_\pi\, g_{AB}\,\partial^A\psi_1(\pi)^*\partial^B\psi_2(\pi). \tag{31}$$

We stress that, while $H$ is generally not $G_R$ invariant and depends on $I^{ab}$ via the inverse metric $g_{AB}$, the scalar product of eq. (29) does not depend of $I^{ab}$ and is invariant under $G_L \times G_R$. By the canonical identification $R^a = \delta^a_R\pi_A(-i\partial^A)$, the Hamiltonian can equivalently be written as

$$\langle\psi_1|H|\psi_2\rangle \equiv \frac{1}{2}\int d\mu_\pi\, I^{-1}_{ab}\,[R^a\psi_1(\pi)]^*[R^b\psi_2(\pi)] \tag{32}$$

$$\equiv \frac{1}{2}\int d\mu_\pi\, I^{-1}_{ab}\,\psi_1(\pi)[R^aR^b\psi_2(\pi)], \tag{33}$$

where in the second line we used the hermiticity of $R^a$ which follows directly from the $G_R$ invariance of the Haar measure.

An important result in group theory, the Peter-Weyl theorem, states that a complete orthonormal basis $\{\Psi\}$ for wave-functions on the manifold of a group $G$ is given by the entries of all possible irreducible representations of $G$

$$\Psi_{r,\alpha_r,\beta_r}(\pi) \equiv D^r_{\alpha_r\beta_r}(g(\pi)), \tag{34}$$

with $r$ the label for the irreps and $\alpha_r, \beta_r = 1, \ldots, d_r$ the indices of the entries, with $d_r$ the dimensionality of the $r$-irrep. In other words, by a suitable normalization of the measure, one has

$$\langle s, \alpha_s\beta_s|r, \alpha_r, \beta_r\rangle \equiv \int d\mu_\pi\, \Psi_{s,\alpha_s,\beta_s}(\pi)^*\Psi_{r,\alpha_r,\beta_r}(\pi) = \delta_{r,s}\delta_{\alpha_s,\alpha_r}\delta_{\beta_s,\beta_r}, \tag{35}$$

whereas any $L_2$ function $\Psi(\pi)$ on $G$ can be uniquely decomposed as

$$\Psi(\pi) = \sum_{r,\alpha_r,\beta_r} c_{r,\alpha_r,\beta_r}\,\Psi_{r,\alpha_r,\beta_r}(\pi). \tag{36}$$

The Hilbert space of the rigid body, decomposes then as a direct sum

$$\mathcal{H} = \oplus_r \mathcal{H}_r, \tag{37}$$

with $\mathcal{H}_r$ a subspace of dimensionality $d_r^2$ generated by the $D^r_{\alpha_r\beta_r}(g(\pi))$ for $\alpha_r, \beta_r = 1, \ldots, d_r$. The $\mathcal{H}_r$ subspaces are invariant under the action of $G_L \times G_R$ as one has

$$D^r(g(\pi)) \to D^r(g_L^{-1}g(\pi)g_R) = D^r(g_L)^{-1}D^r(g(\pi))D^r(g_R). \tag{38}$$

$\mathcal{H}_r$ thus hosts the $(r, r)$ representation of $G_L \times G_R$. In particular, by eq. (38), $G_L$ and $G_R$ act respectively on the $\alpha_r$ and $\beta_r$ indices of the basis vectors $|r, \alpha_r, \beta_r\rangle$. Since the Hamiltonian is purely written in terms of the $G_R$ generators, the subspaces $\mathcal{H}_r$ are also invariant under the action of the Hamiltonian, which block decomposes as

$$H = \oplus_r H_r, \tag{39}$$

with $H_r$ a matrix of dimension $d_r^2 \times d_r^2$. The spectrum of the system is then found by diagonalizing each $H_r$ block separately. As $G_L$ is a symmetry, the eigenvalues of $H_r$ on $\mathcal{H}_r = (r, r)$ are $d_r$ times degenerate. For sufficiently general $I_{ab}$, $G_R$ is fully broken and there are no further

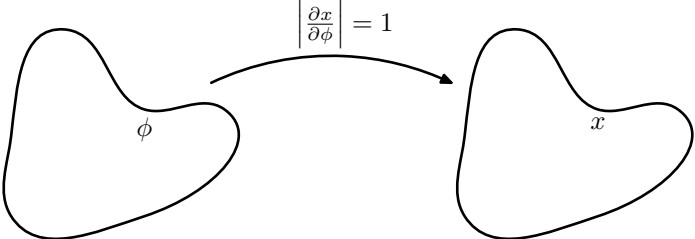

Figure 2: Sketch of the embedding of the fluid elements into physical space.

degeneracies, while in the limiting case $I_{ab} \propto \delta_{ab}$, $G_R$ is a symmetry and the eigenvalues in $\mathcal{H}_r = (r, r)$ are fully degenerate and proportional to the quadratic Casimir.

It is worth showing how the above general discussion incarnates in the case of the ordinary rigid body for which $G$ is taken to be either $SU(2)$ or $SO(3) = SU(2)/Z_2$ (see for comparison the discussion in ref. [36]) . For definiteness, let us consider $SU(2)$ first. Representing the generators by Pauli matrices, we can write

$$U = e^{i\pi_A \sigma^A} \equiv x_0 + i x_A \sigma^A \equiv x_\mu \Sigma^\mu \,, \tag{40}$$

where $A = 1, \dots, 3$ runs on the generators, while $\mu = 0, \dots, 3$ and $x_\mu = (x_0, x_A)$, $\Sigma^\mu = (\mathbb{1}, \sigma^A)$. The $x_\mu$ satisfy the constraint $x_\mu x_\mu \equiv x^2 = 1$ and provide an embedding of the $SU(2)$ group manifold in $\mathbb{R}^4$ showing its diffeomorphic equivalence to $S_3$. The above $2 \times 2$ matrix coincides with $D^{1/2}$, the $\ell = 1/2$ representation of $SU(2)$. According to the discussion around eq. (38), its entries, parametrized by the $x_\mu$, form a $(1/2, 1/2)$ under $SU(2)_L \times SU(2)_R$, that is a vector of the locally isomorphic $SO(4) \sim SU(2)_L \times SU(2)_R$. According to the Peter-Weyl theorem a complete basis of wave functions on $SU(2)$ is given by the set of all the entries of the $D^\ell$ representations of $SU(2)$ for all possible half-integer $\ell$. These transform as $(\ell, \ell)$ under $SU(2)_L \times SU(2)_R$ and are equivalently parametrized by trace-subtracting the monomials $x_{\mu_1} \dots x_{\mu_{2\ell}}$. As these are nothing but the spherical harmonics on $S_3$, for $SU(2)$ the Peter-Weyl theorem boils down to a rather standard result. The Hilbert space of the quantum rigid body thus decomposes in irreducible invariant subspaces under $SU(2)_L \times SU(2)_R$ as $\oplus_\ell (\ell, \ell)$. The $(\ell, \ell)$ blocks have dimensionality $(2\ell+1)^2$ and are invariant under time evolution as the Hamiltonian is a polynomial (a quadratic one) in the $SU(2)_R$ generators $R_a$. The energy levels from $(\ell, \ell)$ have each $2\ell + 1$ degeneracy, because of $SU(2)_L$ invariance. For a general inertia tensor $I_{ab}$, $SU(2)_R$ is fully broken and that is the only degeneracy. The case of $SO(3)$ is now simply obtained by noting that $SO(3) = SU(2)/\mathbb{Z}_2$ with $\mathbb{Z}_2 : x_\mu \to -x_\mu$. The complete set of eigenfunctions on $SO(3) \sim S_3/\mathbb{Z}_2$ is then given by the trace-subtracted $x_{\mu_1} \dots x_{\mu_{2\ell}}$ with integer $\ell$. The resulting Hilbert space is $\oplus_\ell (\ell, \ell)$ with integer $\ell$.

## 3 The perfect fluid in 2D

### 3.1 Classical fluid

The perfect incompressible fluid is formally constructed by considering two copies of the same space: $\mathbf{\Phi}$, the space of fluid elements, and $\mathbf{X}$, the physical space. The physical configurations are given by the set of volume preserving diffeomorphisms $\mathbf{\Phi} \to \mathbf{X}$. Our discussion could easily encompass spaces with any geometry, but to keep the notation simple we focus on spaces with flat geometry, like $\mathbb{R}^3$ or the torus. Parametrizing $\mathbf{\Phi}$ and $\mathbf{X}$ respectively with coordinates $\phi^a$ and $x^I$, the trajectories are given by the mappings $\phi^a \to x^I(\phi, t)$ subject to the condition

$dV(x) = dV(\phi)$ for the volume elements, or equivalently $\det(\partial x/\partial \phi) \equiv |\partial x/\partial \phi| = 1$ for the Jacobian, see Fig. 2. As $\mathbf{\Phi}$ and $\mathbf{X}$ are identical spaces, each configuration $\phi^a \to x^I(\phi, t)$ is an element of the volume preserving diffeomorphism group $S\text{Diff}(\mathbf{X})$ of $\mathbf{X}$ onto itself. This establishes a very close analogy with the generalized rigid body, which we will now elucidate and then exploit. A detailed discussion at the classical level is found in refs. [30, 37].

The invertibility of $x^I(\phi, t)$ offers two alternative perspectives on the flow. The *Lagrangian* perspective views the flow as the trajectory in $\mathbf{X}$ space of each point in $\mathbf{\Phi}$. The *Eulerian* perspective pictures the flow in terms of the time evolution of the fluid velocity at any given point in $\mathbf{X}$. The $x$'s and the $\phi$ are then respectively referred to as the Eulerian and the Lagrangian coordinates, or, equivalently, the physical and the comoving coordinates.

Given the configuration $x(\phi, t)$ and $g \in S\text{Diff}(\mathbf{X})$, we can define, as previously, the left and the right action as[10]

$$\text{left action} \quad x(\phi, t) \to x(g^{-1}(\phi), t), \tag{41}$$

$$\text{right action} \quad x(\phi, t) \to g(x(\phi, t)). \tag{42}$$

An infinitesimal $g \in S\text{Diff}(\mathbf{X})$ transformation is written as $g(y) \simeq y + f(y)$ with $f^I(y)$ a vector field satisfying $\partial_I f^I = 0$. The corresponding infinitesimal variations are

$$\text{left} \quad \delta_L x^I = -f^a(\phi)\partial_a x^I, \tag{43}$$

$$\text{right} \quad \delta_R x^I = f^I(x). \tag{44}$$

To study the dynamics we will focus on the simplest possible situation of a 2D fluid. It will also be convenient to compactify space on a square torus $T^2$ of radius $r$: all coordinates $(x^1, x^2$ and $\phi^1, \phi^2)$ then range between 0 and $2\pi r$. Working on $T^2$, it will also be useful to define ($\epsilon_{12} = -\epsilon_{21} = 1$, $\epsilon^{IJ}\epsilon_{KJ} = \delta^I_K$)

$$\tilde{x}_I \equiv \epsilon_{IJ}x^J, \qquad \tilde{\partial}^I \equiv \epsilon^{IJ}\partial_J, \qquad \tilde{\phi}_a \equiv \epsilon_{ab}\phi^b, \qquad \tilde{\partial}^a \equiv \epsilon^{ab}\partial_b, \tag{45}$$

by which the incompressibility condition $|\frac{\partial x}{\partial \phi}| = 1$ can be equivalently expressed as

$$\tilde{\partial}^a \tilde{x}_I = \partial_I \phi^a, \qquad \tilde{\partial}^I \tilde{\phi}_a = \partial_a x^I. \tag{46}$$

Indicating by $v^I(\phi, t) = dx^I(\phi, t)/dt$ the velocity field, the Lagrangian in the incompressible limit is simply [20] (from here on, we indicate the mass density $\rho_m$ as $\rho$)

$$\mathcal{L} = \int d^2\phi \, \frac{\rho}{2} \, v^2. \tag{47}$$

Notice that incompressibility constrains $v^I$ to be divergence free:

$$\partial_I \dot{x}^I = \partial_a \dot{x}^I \partial_I \phi^a = \partial_a \dot{x}^I \tilde{\partial}^a \tilde{x}_I = \frac{1}{2}\frac{d}{dt}\left|\frac{\partial x}{\partial \phi}\right| = 0. \tag{48}$$

Given $x^I(\phi, t)$ are constrained variables, the Hamiltonian description and canonical quantization are not immediately derived, at least at first sight. The standard approach is offered by Dirac's method, consisting in the replacement of the Poisson brackets (and of their quantum counterparts) with suitable Dirac brackets. However, like in eqs. (25,27,28), to describe the dynamics we won't really need the variable $x^I(\phi, t)$, but just the *R*-charges, their commutation

---

[10]Notice that, in the case of the rigid body, the left action is on the laboratory frame while here it is on the analogue of the rigid body frame, i.e. the fluid coordinates $\phi$. Because of that, in the mathematics literature, see [37], the naming of eqs. (41) and (42) is swapped. We chose to call *left* the action that corresponds to a real symmetry.

relations and the expression of the Hamiltonian. All of that can be robustly derived bypassing Dirac's procedure. The point is very simple and general and can be made by considering a generic dynamical system with variables $q_\alpha$ (with $\alpha$ discrete or continuous) subject to a set of constraints that do not involve time derivatives (like for our $x^I(\phi, t)$ or for the $U$ matrices of the previous section). The constraints can be solved, at least locally, in favor of a set of non redundant parameters $\pi_A$: $q_\alpha \equiv q_\alpha(\pi)$. For the systems we are considering, the role of the $\pi$ is played by the Lie parameters of the group. Now, a symmetry transformation on the non-redundant variables $\delta\pi_A$ will correspond to $\delta q_\alpha = \delta\pi_A(\partial q_\alpha/\partial\pi_A)$. In particular for a time translation we have

$$\dot{q}_\alpha = \dot{\pi}_A \frac{\partial q_\alpha}{\partial \pi_A} \Rightarrow \frac{\partial \dot{q}_\alpha}{\partial \dot{\pi}_A} = \frac{\partial q_\alpha}{\partial \pi_A}. \tag{49}$$

By this equation, the Lagrangian $\mathcal{L}(q, \dot{q})$, can be written in terms of $\pi_A$ and $\dot{\pi}_A$, while the Hamiltonian and conserved global charges are equally well written in terms of either the $\pi$'s or the $q$'s:

$$H = \frac{\partial \mathcal{L}}{\partial \dot{\pi}_A} \dot{\pi}_A - \mathcal{L} = \frac{\partial \mathcal{L}}{\partial \dot{q}_\alpha} \frac{\partial \dot{q}_\alpha}{\partial \dot{\pi}_A} \dot{\pi}_A - \mathcal{L} = \frac{\partial \mathcal{L}}{\partial \dot{q}_\alpha} \dot{q}_\alpha - \mathcal{L}, \tag{50}$$

$$Q = \frac{\partial \mathcal{L}}{\partial \dot{\pi}_A} \delta\pi_A = \frac{\partial \mathcal{L}}{\partial \dot{q}_\alpha} \frac{\partial q_\alpha}{\partial \pi_A} \delta\pi_A = \frac{\partial \mathcal{L}}{\partial \dot{q}_\alpha} \delta q_\alpha. \tag{51}$$

By employing the left ends of the above two equations, the Poisson brackets (or commutators) are then trivially determined by working with the unconstrained canonical variables, $\pi_A$ and $p_A \equiv \partial\mathcal{L}/\partial\dot{\pi}_A$. On the other hand, the right ends of the same equations offer the charges in terms of the more manageable, but constrained, $q_\alpha$ and $\dot{q}_\alpha$.

According to eqs. (43,44) the charges of $S\text{Diff}(T^2)_L \times S\text{Diff}(T^2)_R$ are then

$$L_f = -\int d^2\phi\, \rho\, \dot{x}^I f^a \partial_a x^I, \tag{52}$$

$$R_f = \int d^2\phi\, \rho\, \dot{x}^I f^I(x(\phi)) = \int d^2x\, \rho\, \dot{x}^I f^I(x), \tag{53}$$

where in the last equation we made use of the constraint $\left|\frac{\partial x}{\partial \phi}\right| = 1$ and switched to Eulerian coordinates. By the discussion in the previous paragraph, these charges must satisfy the algebra of the grou.[11] To spare formulae, we can directly write the result in terms of commutators in the quantum theory. They read

$$\left[L_f, L_h\right] = L_{[f,h]}, \qquad \left[R_f, R_h\right] = R_{[f,h]}, \qquad \left[L_f, R_h\right] = 0, \tag{54}$$

where $[f, h]$ is the vector field commutator

$$[f, h]^J = i\left(h^I \partial_I f^J - f^I \partial_I h^J\right). \tag{55}$$

In full analogy with eq. (26) we can express the right charges $R_f$ as a linear combination of the left ones with coefficients that depend on the physical configuration $x(\phi, t)$. Indeed one can first write the $R$ charges as

$$R_f = \int d^2\phi\, \rho\, \dot{x}^I f^I(x(\phi)) = \int d^2\phi\, \rho\, \dot{x}^I \partial_a x^I \partial_J \phi^a f^J(x(\phi)) \equiv \int d^2\phi\, \rho\, \dot{x}^I \hat{f}^a \partial_a x^I, \tag{56}$$

where $\hat{f}^a \equiv f^J \partial_J \phi^a$ can be viewed as a function of $\phi$ and $t$ determined by the configuration $x^I(\phi, t)$. Secondly, using eq.(46) one can show that $\partial_a \hat{f}^a = 0$. A comparison of eqs. (52) and

---

[11]The same result has been obtained by employing the Dirac bracket formalism [21].

(56), then shows that the latter equation offers the $R$-charges as linear combinations, with time dependent coefficients, of the $L$-charges. Eq. (56) is therefore the analogue of eq. (26).

We can further elucidate the relation between $L_f$ and $R_f$ by writing the general transverse vector fields in respectively $\phi$ and $x$ coordinates as ($L$ and $R$ subscripts used in an obvious way)

$$f_L^a(\phi) = f_{0L}^a - \tilde{\partial}^a f_L(\phi), \qquad f_R^I(x) = f_{0R}^I - \tilde{\partial}^I f_R(x), \qquad (57)$$

where $f_{0L}^a$, $f_{0R}^I$ are just constants parametrizing rigid translations, while $f_L$ and $f_R$ are single valued functions on $T^2$. By simple manipulations and using the identities of eq. (46) we can then write the charges as

$$L_{f_L} = \int d^2\phi\, \rho \left[ -\dot{x}^I \partial_a x^I f_{0L}^a + \epsilon^{JI} \partial_J \dot{x}^I f_L(\phi) \right], \qquad (58)$$

$$R_{f_R} = \int d^2 x\, \rho \left[ \dot{x}^I f_{0R}^I - \epsilon^{JI} \partial_J \dot{x}^I f_R(x) \right]. \qquad (59)$$

The two constants $f_{0L}^a$ and the function $f_L(\phi)$ label all the conserved charges. Among the right charges only those associated with $f_{0R}^I$ are exactly conserved, and these simply correspond to the total momentum. Notice however that the $R$-charges, and the total momentum in particular, are expressible as a time dependent linear combination of left charges. A complete basis of the conserved charges can thus be equivalently labelled by $f_{0R}^I$ (total momentum) and $f_L(\phi)$. Finally, it is interesting to choose a $\delta$-function basis for the Lie parameter functions: $f_L(\phi, \bar{\phi}) = \delta^2(\phi - \bar{\phi})$ and $f_R(x, \bar{x}) = \delta^2(x - \bar{x})$. That choice defines local charges associated to the vorticity $\Omega \equiv \partial \wedge v = \epsilon^{JI} \partial_J \dot{x}^I$. More precisely, and in an obvious notation, we can write

$$L(\phi, t) = \rho\, \Omega(x(\phi, t), t) \equiv \tilde{\Omega}(\phi, t), \qquad R(x, t) = -\rho\, \Omega(x, t). \qquad (60)$$

The conservation of left charges $\dot{L}(\phi, t) = 0$ then implies

$$\frac{d}{dt} \tilde{\Omega}(\phi, t) = 0, \qquad (61)$$

which in Eulerian coordinates reads

$$(\partial_t + v^I \partial_I) \Omega(x, t) = 0. \qquad (62)$$

This last equation expresses the well known convective conservation of vorticity. It is the fluid analogue of eq. (28). Even though vorticity is only convectively conserved, eq. (62) together with $\partial_I v^I = 0$ implies that the quantities

$$C_k \equiv \int d^2 x\, \Omega(x)^k, \qquad (63)$$

are conserved. The $C_k$ are nothing but the Casimirs of $S\text{Diff}(T^2)_R$. In the Hamiltonian approach (see below) their conservation simply follows from the fact that the Hamiltonian is a function of the $R$ charges. That is again in full analogy with the case of the rigid body.

## 3.2 Quantum mechanical fluid

Precisely like for the rigid body, we can now write the Hamiltonian in terms of the generators of $S\text{Diff}(T^2)_R$. For that purpose it is useful to work in momentum space, where a suitably normalized complete basis of infinitesimal $S\text{Diff}(T^2)_R$ transformations is given by (see eq. (57))

$$f_{0J}^I = \delta_J^I r, \qquad (64)$$

$$f_{\mathbf{n}}^I = -i\, \epsilon^{IJ} n_J r e^{i\mathbf{n}x/r} \equiv -i\, \tilde{n}^I r e^{i\mathbf{n}x/r}. \qquad (65)$$

Here $f_{0J}^I$, for $J = 1, 2$, is the vector field associated with rigid translations, $\mathbf{n} \equiv (n_1, n_2)$ is an integer valued wave vector and $\boldsymbol{x} \equiv (x^1, x^2)$ is the coordinate vector on $T^2$. From here on we stick to boldface type to indicate 2-vectors. Indeed $\mathbf{n} \in \mathbb{Z}^2 - (0, 0)$, as there is no $f_{\mathbf{n}}^I$ associated with $\mathbf{n} = 0$. The corresponding charges $R_J$ and $R_{\mathbf{n}}$, according to eq. (54), satisfy the algebra

$$[R_J, R_I] = 0, \qquad [R_J, R_{\mathbf{n}}] = n_J R_{\mathbf{n}}, \qquad [R_{\mathbf{n}}, R_{\mathbf{m}}] = i(\mathbf{n} \wedge \mathbf{m}) R_{\mathbf{n+m}}, \tag{66}$$

where $(\mathbf{n} \wedge \mathbf{m}) \equiv \epsilon^{IJ} n_I m_J$. According to eq. (53) we can write explicitly

$$R_J = \rho \, r \int d^2 x \, \dot{x}^J \equiv \rho \, r (2\pi r)^2 \bar{v}^J, \qquad R_{\mathbf{n}} = -i\rho \, r \int d^2 x \, (\dot{\boldsymbol{x}} \wedge \mathbf{n}) \, e^{i\mathbf{n}\boldsymbol{x}/r} \equiv -\rho \, r^2 \Omega_{\mathbf{n}}. \tag{67}$$

$R_J \equiv r P_J$ are proportional to the total momentum and to the zero mode of the velocity $\boldsymbol{v} \equiv \dot{\boldsymbol{x}}$. The $R_{\mathbf{n}}$ are instead proportional to the Fourier modes of the vorticity, $\Omega(x) \equiv \partial \wedge \boldsymbol{v}(x)$, which in turn are in one-to-one correspondence with the non-zero modes of the velocity. Again $R_{\mathbf{n}=0} = 0$, corresponding to vorticity being a total derivative with vanishing zero mode.

The velocity $v(x)^I$ is fully determined by its zero mode $\bar{v}^I$ and by the vorticity according to

$$v^I = \bar{v}^I - \epsilon^{IJ} \frac{\partial_J}{\Delta'} \Omega, \tag{68}$$

where $1/\Delta'$ is the inverse Laplacian on $T^2$, which is well defined on functions with vanishing zero mode. From this result, the Hamiltonian can be written in terms of the $R$ charges as

$$H = \frac{\rho}{2} \int d^2 x \, \boldsymbol{v}^2 = \frac{\rho}{2} \int d^2 x \left( \bar{v}^2 - \Omega \frac{1}{\Delta'} \Omega \right) = \frac{1}{2\rho} \frac{1}{(2\pi)^2 r^4} \left[ R_J R_J + \sum_{\mathbf{n} \neq 0} \frac{1}{\mathbf{n}^2} R_{\mathbf{n}}^\dagger R_{\mathbf{n}} \right]. \tag{69}$$

By eqs. (66) and (67) the time evolution of the vorticity is then

$$\dot{\Omega}_{\mathbf{n}} \equiv i[H, \Omega_{\mathbf{n}}] = \frac{i}{2} \frac{n_J}{r} (\bar{v}_J \Omega_{\mathbf{n}} + \Omega_{\mathbf{n}} \bar{v}_J) + \frac{1}{2(2\pi r)^2} \sum_{\mathbf{m} \neq 0} \frac{\mathbf{n} \wedge \mathbf{m}}{\mathbf{m}^2} \left( \Omega_{\mathbf{m}}^\dagger \Omega_{\mathbf{n+m}} + \Omega_{\mathbf{n+m}} \Omega_{\mathbf{m}}^\dagger \right). \tag{70}$$

A Fourier transform to position space and use of eq. (68) finally give, as expected, the quantum mechanical Euler equation

$$\dot{\Omega}(x) = -\frac{1}{2} \left( v^J(x) \partial_J \Omega(x) + \partial_J \Omega(x) v^J(x) \right). \tag{71}$$

Notice that, as eqs. (70) or (71) involve respectively infinite momentum sums or products of operators at coinciding points, we expect the need for a UV regulator at some stage.

Now, in order to construct the quantum theory, in analogy with the case of the rigid body, the first step would be to find the complete basis of the Hilbert space. As the group manifold is now infinite dimensional we must deal with functionals rather than just functions. Unfortunately we are not aware of an analogue of the Peter-Weyl theorem for such case. One natural way to proceed would then be to find an infinite sequence of finite groups that approximate $S\text{Diff}(T^2)$ arbitrarily well. As we show in the next section, that can indeed be done. Regardless of the details, such finite group approximations of the perfect fluid will be nothing but special cases of the generalized rigid body discussed in the previous section. The basis of the Hilbert space, see eq. (37), will thus consist of $d_r^2$-dimensional $(r, r)$ representations of the mutually commuting $L$ and $R$ charges.

The $(r, r)$ structure of the Hilbert space basis and the associated degeneracies invites some considerations. As the eulerian flow configuration $\boldsymbol{v}(x, t)$ is fully specified by the $R$ charges, the corresponding states in the $(r, r)$ block will have a perfect $d_r$-degeneracy associated with

the action of the $L$ algebra. Classically this corresponds to the fact that the eulerian variables, $\boldsymbol{v}(x,t)$, determine the Lagrangian coordinates $\phi_a$ only up to the action of $(SDiff)_L$. On the other hand, if we were to consider $\boldsymbol{v}(x,t)$ as the only physical variables, we could do away with the $L$ algebra and have a Hilbert space basis featuring just one copy of each $r$ irrep of the $R$ algebra. As the $r$ blocks have dimension $d_r$, as opposed to the dimension $d_r^2$ for the $(r,r)$ blocks, the latter construction would in practice reduce the number of dynamical degrees of freedom by a factor of two. For instance, the entropy of each block would be $\ln d_r$ instead of $2\ln d_r$. Indeed, as discussed at the end of section 2.1, a reduction by roughly a factor of 2 in the number of degrees of freedom is also operated, at least for a group $G$ of large dimension $d_G \gg 1$, by projecting the motion of the rigid body on the space of $R$ charges subject to the Casimir constraints. In that case the resulting Hamiltonian system consists of $(d_G - d_C)/2$ canonical pairs, which for a group of large dimension is roughly half the original number $d_G$ of canonical pairs.[12]

With the above comments in mind, we can then face the degeneracy in two ways. The first, **A**, is to accept it, implying there exist roughly twice as many dynamical variables as accounted for by $\boldsymbol{v}(x,t)$. This doubling does not seem to cause any physical inconsistency, besides the annoying feature that on eulerian variables the state will be in general described by a density matrix, rather than by a pure state. For instance, the state $\sum_i |\psi_i^L\rangle \otimes |\psi_i^R\rangle$ describes measurements of $\boldsymbol{v}(x,t)$ through the density matrix $\sum_{ij} \rho_{ij} |\psi_i^R\rangle \langle \psi_j^R|$, with $\rho_{ij} = \langle \psi_j^L | \psi_i^L \rangle$.

The second, **B**, is to take the eulerian variables $\boldsymbol{v}(x,t)$ as the only physical ones, and view the degeneracy as unphysical. In this case we can further think of two options. The standard first option, **B1**, is to simply gauge $SDiff(T^2)_L$, i.e. the exact symmetry responsible for the degeneracy. This just amounts to projecting on the subspace with vanishing $L$ charges. Unfortunately, as the states come in equivalent representations $(r,r)$ of the mutually commuting $L$ and $R$ charges, the only state surviving this projection is the total singlet state, where all the $R$ charges also vanish and the flow is trivial: all the degrees of freedom are projected out and $\boldsymbol{v}(x,t) \equiv 0$. What we end up with is therefore a fluid without any transverse mode. If we view our incompressible fluid as the EFT resulting upon integrating out the compressional modes (along the lines explained in the Introduction), the only residual flow variables would precisely be those compressional modes. The resulting system would then appear indistinguishable from a superfluid [18].

The less standard second option, **B2**, is instead to construct a Hilbert space that only represents the $R$-charges. In a sense this corresponds to renouncing the existence of a group manifold configuration space. This is loosely similar to having a rotator that is not an extended molecule, but just an internal spin degree of freedom.[13] Very much like in that case, however, there is now no clear rationale for which irreps of the $R$-algebra to admit in the Hilbert space.

In the rest of the paper we will only consider the $R$ algebra. Our results can then be suitably interpreted according to either hypothesis **A** or hypothesis **B2**. We leave the discrimination of these two options for future work, possibly considering concrete physical systems. As shown in the rest of the paper, the dynamics that emerges from our construction is structurally rich and new. In our mind this justifies setting aside, for the moment, the degeneracy issue.

# 4  Finite truncation of $SDiff(T^2)$

Our goal is now to parametrize the states and the dynamics by approximating $SDiff(T^2)$ by some finite-dimensional Lie group. As the hydrodynamic description is anyway expected to

---

[12]One should be careful when extending this naive counting to the limit where the group dimension becomes infinite.

[13]We thank Alberto Nicolis for this illuminating analogy.

break down at short distance, it is natural to expect there exists a finite dimensional construction that captures the long distance dynamics. In what follows we present such a truncation and study the resulting finite system.

The long distance dynamics is intuitively captured by $R_{\mathbf{n}}$ with small enough $|\mathbf{n}|$. However, if we limited $\mathbf{n}$ to any finite range, the commutation relation in eq. (66) would not close. The commutation relations must thus be modified for large enough $|\mathbf{n}|$, say $|\mathbf{n}| \gtrsim n_{UV}$. Indeed, as it was clarified in a series of papers [38–40] already in the 80's, the Lie algebra of $SU(N)$ with $N > O(n_{UV}^2)$ offers such consistent finite truncation of the $S\mathrm{Diff}(T^2)$ algebra.[14] This realization was made in the attempt to quantize the relativistic membrane, where a convenient partial gauge fixing happens to respect invariance under time-independent area-preserving diffeomorphisms [41–43], in close analogy with the fluid we are considering.[15] The finite truncation is made manifest for a particular choice of the generators of $SU(N)$, which was introduced by 't Hooft [48] and which we now describe. Assuming, for convenience, $N$ is odd, consider the two unitary $N \times N$ matrices $h$ and $g$

$$h^{\alpha}_{\ \beta} = \delta_{\alpha+1,\beta}, \qquad \text{and} \qquad g^{\alpha}_{\ \beta} = \omega^{\alpha} \delta_{\alpha,\beta}. \tag{72}$$

Here $\alpha$ and $\beta$ take the $N$ integer values running from $-\frac{N-1}{2}$ to $\frac{N-1}{2}$, including 0 ($N$ is odd), while $\omega \equiv e^{\frac{2\pi}{N}i}$ is a primitive $N$-th root of unity. We also define $\delta_{\alpha,\beta} = 1$ for $\alpha = \beta \mod N$, so that $h$ features 1's at one step above the diagonal and at the lower left corner. In matrix form we have

$$h = \begin{pmatrix} 0 & 1 & 0 & \dots & 0 \\ 0 & 0 & 1 & \dots & 0 \\ \vdots & \vdots & \vdots & \ddots & \vdots \\ 0 & 0 & 0 & \dots & 1 \\ 1 & 0 & 0 & \dots & 0 \end{pmatrix}, \qquad g = \begin{pmatrix} \omega^{-\frac{N-1}{2}} & 0 & 0 & \dots & 0 \\ 0 & \omega^{-\frac{N-3}{2}} & 0 & \dots & 0 \\ 0 & 0 & \omega^{-\frac{N-5}{2}} & \dots & 0 \\ \vdots & \vdots & \vdots & \ddots & \vdots \\ 0 & 0 & 0 & \dots & \omega^{\frac{N-1}{2}} \end{pmatrix}, \tag{73}$$

and, as one can easily check,

$$g^N = h^N = 1, \qquad hg = \omega\, gh. \tag{74}$$

Multiplying different powers of $h$ and $g$ one can construct $N^2$ unitary matrices, the 't Hooft matrices:

$$J_{\mathbf{n}} \equiv \omega^{\frac{1}{2}n^1 n^2} g^{n^1} h^{n^2}, \tag{75}$$

where the components of the 2-vector index $\mathbf{n} \equiv (n^1, n^2)$ take values in the same range as the matrix indices: $-\frac{N-1}{2} \leq n^I \leq \frac{N-1}{2}$. All these matrices are linearly independent and traceless,

---

[14]This construction extends to other 2D surfaces [41] such as the Klein bottle or the projective plane, where the associated area-preserving transformation group is also viewed as the large $N$ limit of a classical finite-dimensional Lie group. In the end, as we shall explain, in the large $N$ limit the actual global shape of the 2D surface will not matter for the local fluid dynamics. We will thus content ourselves with $T^2$.

[15]We learned of the $S\mathrm{Diff}(T^2) \sim SU(N \to \infty)$ equivalence from George Savvidy, right at the beginning of our study. Only later did we discover that its potential relevance in the study of 2D fluids had already been considered several times in the literature. In particular, refs. [44,45] applied it in a classical statistical mechanics context in order to reduce to a finite set of variables. Another possible connection came directly from the study of the membrane where the dynamics was formulated in terms of a perfect fluid [46], though apparently one undergoing a pure gradient flow, which is not the case we are considering. More recently the $SU(N)$ rigid body and its relation with the perfect 2D fluid was investigated in the context of quantum complexity in ref. [47]. Somehow all these other studies have, in the end, little concrete overlap with ours. As we will have the opportunity to explain below, the main reason for that resides in the effective field theory perspective underlying our approach. One main consequence of that, and a centerpiece of our construction, is the necessity for a UV regulation of the Hamiltonian. It would however be interesting to go back and investigate the possible relevance of our field theoretic construction to those previous studies.

apart from $J_{(0,0)}$, which is the identity matrix. The choice of the numerical factor in front ensures that $J_\mathbf{n}^\dagger = J_{-\mathbf{n}}$. Therefore, the $N^2 - 1$ matrices $J_\mathbf{n} + J_{-\mathbf{n}}$ and $i(J_\mathbf{n} - J_{-\mathbf{n}})$ with $\mathbf{n} \neq \mathbf{0}$ form a complete set of traceless Hermitian $N \times N$ matrices, i.e. a basis for the $SU(N)$ algebra.

The commutator of two 't Hooft matrices reads

$$[J_\mathbf{n}, J_\mathbf{m}] = -2i \sin\left(\frac{\pi}{N}(\mathbf{n} \wedge \mathbf{m})\right) J_{\mathbf{n+m}}. \tag{76}$$

Notice that $J_\mathbf{n}$ is periodic up to a sign when any entry of $\mathbf{n}$ is shifted by $N$, or more precisely

$$J_{(n_1+N,n_2)} = (-1)^{n_2} J_{(n_1,n_2)}, \qquad J_{(n_1,n_2+N)} = (-1)^{n_1} J_{(n_1,n_2)}. \tag{77}$$

Eq. (76) is then consistent also for $n^I + m^I$ outside the domain $\left[\frac{N-1}{2}, \frac{N-1}{2}\right]$.

When $|\mathbf{n}|$ and $|\mathbf{m}|$ are smaller than $\sqrt{N}$, the sine in eq. (76) can be approximated by its argument so that the result is proportional to that of $S\text{Diff}(T^2)$, see eq. (66). To match more precisely, we introduce a set of rescaled $SU(N)$ generators

$$\tilde{R}_\mathbf{n} \equiv -\frac{N}{2\pi} J_\mathbf{n}, \tag{78}$$

for which the commutation relation reads

$$\left[\tilde{R}_\mathbf{n}, \tilde{R}_\mathbf{m}\right] = i\frac{N}{\pi} \sin\left(\frac{\pi}{N}(\mathbf{n} \wedge \mathbf{m})\right) \tilde{R}_{\mathbf{n+m}}. \tag{79}$$

Comparing to eq. (66) we see that the commutation relations of $SU(N)$ and $S\text{Diff}(T^2)$ coincide for $|\mathbf{n}| \ll \sqrt{N}$. The role of $n_{UV}$ is thus played by $\sqrt{N}$. For $|\mathbf{n}| \ll \sqrt{N}$ it is thus natural to identify the $SU(N)$ generators $\tilde{R}_\mathbf{n}$ with the $S\text{Diff}(T^2)$ generators $R_\mathbf{n}$. This provides an embedding of a truncated $S\text{Diff}(T^2)$ algebra into the $SU(N)$ algebra. Notice that the set $\{\tilde{R}_\mathbf{n}\}$ with $|\mathbf{n}| \lesssim \sqrt{N}$, which includes the truncated $S\text{Diff}(T^2)$, consists of $\sim O(N)$ generators. For large $N$ this is but a tiny fraction of the $N^2 - 1$ generators of $SU(N)$. The fact that the commutation relations of eq. (66) and eq. (79) approximately coincide only for a (small) subset of the generators cannot be overlooked, and will be crucial in our construction of the effective Hamiltonian in the next section. To our knowledge, this fact was not not emphasized and did not play a role in the original literature discussing the finite truncation of $S\text{Diff}$, whether in the context of classical fluid or in that of relativistic membranes.

The $R_\mathbf{n}$ actually span a subalgebra $S\text{Diff}(T^2)' \subset S\text{Diff}(T^2)$. The full algebra also includes the two translation generators $R_1$ and $R_2$. Strictly speaking then, what we have just shown is that $SU(N)$ offers a truncation of the subalgebra $S\text{Diff}(T^2)'$. We will comment in a moment on the fate of translations in the $SU(N)$ modelling of the fluid. The absence of the analogues of the $R_I$ represents at first sight an obstacle to the proper description of the fluid long distance dynamics, see eq. (69). However we will explain in what sense that is not the case. In the rest of the paper we will keep indicating by $S\text{Diff}(T^2)$ the $R_\mathbf{n}$ algebra, though it should be understood that we mean indeed $S\text{Diff}(T^2)'$.

The coincidence of the $SU(N)$ and $S\text{Diff}(T^2)$ algebras for $|\mathbf{n}| \lesssim \sqrt{N}$, suggests that the $SU(N)$ quantum rigid body —a problem we have shown how to treat— offers a UV completion of incompressible quantum-hydrodynamics on $T^2$. As the Euler equation is local, we expect there should exist a local effective description that reduces to hydrodynamics at long distance, while the short distance degrees of freedom effectively decouple. Indeed, following these suggestions, we will make an appropriate choice for the Hamiltonian of the $SU(N)$ rigid body such that the low $|\mathbf{n}|$ degrees of freedom satisfy a regulated form of the quantum Euler equation of eqs. (70) and (71). It will therefore be natural to interpret the resulting long distance dynamics as a quantum incompressible fluid. Moreover, as the quantum Euler equation coincides, modulo commutators, with its classical counterpart, classical hydrodynamics

should also emerge in suitable "excited states", where the semiclassical approximation applies and where commutators can be neglected in first approximation.

Notice we here focused only on the $R$ charges, replacing in practice the infinite dimensional group $S\text{Diff}(T^2)_R$ with the finite dimensional $SU(N)_R$. We will later discuss the (possible) role of $S\text{Diff}(T^2)_L$ and $SU(N)_L$.

To close this section and before studying the dynamics, we should discuss the change in the "kinematics" entailed by the truncation $S\text{Diff}(T^2) \to SU(N)$. The generators of $S\text{Diff}(T^2)$, the $R_{\mathbf{n}}$, are labelled by a discrete momentum $\mathbf{n} \in \mathbb{Z}^2 - (0,0)$. The *infinity and discreteness* of this set is the reflection of respectively the continuity and the compactness of the manifold, $T^2$, upon which the fluid flows. For the $SU(N)$ generators $\tilde{R}_{\mathbf{n}}$, $\mathbf{n}$ takes instead $N^2 - 1$ values, a finite number. Technically we can view this finite set as $\mathbb{Z}^2 - (0,0)$ modded by the shift $\mathbf{n} \to \mathbf{n} + N\mathbf{m}$ with $\mathbf{m} \in \mathbb{Z}^2$. This set is just the discrete $N \times N$ torus, with the origin $(0,0)$ removed. It is natural to interpret also this $\mathbf{n}$ as a momentum variable.[16] As the discrete $N \times N$ torus is dual to itself under Fourier transform, the truncation to $SU(N)$ is therefore equivalent to replacing $T^2$ with an $N \times N$ toroidal lattice. Matching the length of the latticized torus to that ($2\pi r$) of the original one, fixes the lattice spacing $a$ to equal $\frac{2\pi r}{N}$. Correspondingly, the Fourier label $\mathbf{n}$ corresponds to momentum $\mathbf{p} = \mathbf{n}/r$. As the $S\text{Diff}(T^2)$ and $SU(N)$ algebra coincide for $|\mathbf{n}| \lesssim \sqrt{N}$, the momentum $\Lambda \equiv \sqrt{N}/r$ is naturally identified with the UV cut-off of the fluid description. In position space this corresponds to a breakdown of hydrodynamics at distances shorter than[17]

$$2\pi/\Lambda = 2\pi r/\sqrt{N} \equiv a_1. \tag{80}$$

The three relevant length scales then satisfy

$$a = \frac{a_1}{\sqrt{N}}, \qquad 2\pi r = a_1\sqrt{N} \qquad \Rightarrow \qquad a_1^2 = 2\pi r \times a, \tag{81}$$

so that, for large $N$, $a \ll a_1 \ll 2\pi r$. Moreover, taking the $N \to \infty$ limit with the physical UV cut-off $a_1$ fixed, corresponds to taking the continuum limit $a \to 0$ and the infinite volume limit $2\pi r \to \infty$ at the same time.

By the above discussion, the sites of the dual spacial $N \times N$ lattice can be parametrized as $\mathbf{x} = a\,\boldsymbol{\alpha}$ with $\boldsymbol{\alpha}$ an integer 2-vector. According to eq. (67) we can then define the UV regulated spacial vorticy as

$$\tilde{\Omega}(\mathbf{x}) = -\frac{1}{(2\pi)^2 \rho r^4} \sum_{\mathbf{n}}{}' e^{-i\frac{\mathbf{n}\mathbf{x}}{r}} \tilde{R}_{\mathbf{n}} \equiv \frac{N}{(2\pi)^3 \rho r^4} \sum_{\mathbf{n}}{}' \omega^{-\mathbf{n}\boldsymbol{\alpha}} J_{\mathbf{n}}, \tag{82}$$

where here and henceforth the prime indicates a sum running over the discrete $N \times N$ torus: $-\frac{1}{2}(N-1) \le n^I \le \frac{1}{2}(N-1)$, with $\mathbf{n} = \mathbf{0}$ excluded.

Since we have reduced space to a lattice, infinitesimal translations are no longer a symmetry. This corresponds to our previous remark that no generator of the $SU(N)$ algebra at finite $N$ plays the role of the two translation generators $R_J = rP_J$ of eq. (67). However, and somewhat expectedly, there exist instead two finite $SU(N)$ group elements corresponding to finite translations by one lattice site in each direction. These are

$$T_1 = h^{-1}, \quad \text{and} \quad T_2 = g. \tag{83}$$

Acting on the vorticity operators, we have indeed

$$T_1 R_{\mathbf{n}} T_1^{-1} = \omega^{-n^1} R_{\mathbf{n}} \equiv e^{-i\frac{n^1}{r}a} R_{\mathbf{n}}, \qquad\qquad T_2 R_{\mathbf{n}} T_2^{-1} = \omega^{-n^2} R_{\mathbf{n}} \equiv e^{-i\frac{n^2}{r}a} R_{\mathbf{n}}, \tag{84}$$

---

[16]We recall that the removal of $(0,0)$ is associated with the vanishing of the zero mode of the vorticity.

[17]In an ordinary fluid such scale would coincide with the mean free path of the particles that compose it.

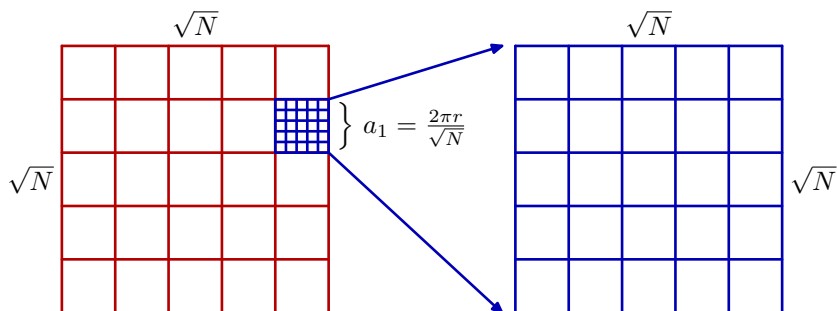

Figure 3: The $N \times N$ finer lattice of size $a$ is decomposed into $\sqrt{N} \times \sqrt{N}$ coarser cells with lattice size $a_1$.

showing that $R_{\mathbf{n}}$ transforms like an operator with momentum $p^I = \frac{n^I}{r}$ under a translation of length $a$.[18]

Note that elementary translations commute only up to a phase

$$T_2^{-1} T_1^{-1} T_2 \, T_1 = \omega \,. \tag{85}$$

$T_2^{-1} T_1^{-1} T_2 \, T_1$ is indeed the generator (i.e. the *smallest* element $\neq e$) of the $Z_N$ center of $SU(N)$, which, in the fundamental representation, consists of the matrices $\omega^n \times \mathbb{1}$, for $n = 0, \dots, N-1$. The center $Z_N$ is however trivially realized, and only in that case, in representations that can be written as a tensor product of a multiple of $N$ of fundamentals or, equivalently, as a tensor product of adjoints. Therefore only in the latter subset of representations are translations commuting. In particular, translations do not commute in the fundamental representation, where eq. (85) is computed.

For states where $Z_N$ is non-trivially realized, the fluid appears to live on a non-commutative torus. As we shall better see below, that is in full analogy with motion in a homogeneous magnetic field. In fact, $T_2^{-1} T_1^{-1} T_2 \, T_1 \in Z_N$ implies that, at the classical level, the truncation of $S\mathrm{Diff}(T^2)$ is actually the projective unitary group $PSU(N) = SU(N)/Z_N$, rather than the full $SU(N)$. In $PSU(N)$ two elementary translations do commute. Quantum mechanically, however, the states of the system can in principle transform in projective representations of $PSU(N)$. These coincide with ordinary representations of $SU(N)$. The fluid states that transform in projective representations of $PSU(N)$ are fully analogous to the spinors of the rotation group. In what follows we will indicate the symmetry group as $SU(N)$, though what we mean is, equivalently, $PSU(N)$ with projective representations.

While translations $T_J$ by one lattice site do not commute, eq. (85) also implies that translations by $\sqrt{N}$ spaces, $T_J^{\sqrt{N}}$ do commute, given $\omega^N = 1$. Of course in order for this statement to make sense we should assume $\sqrt{N}$ is also an integer, which we will in the following. $T_J^{\sqrt{N}}$ represent translations over a distance $\sqrt{N}a = a_1$, which is precisely the length scale we have identified as the ultimate possible short distance cut-off of hydrodynamics. Thus, over the length scales where hydrodynamics applies, translations are realized in the standard commuting way. In view of the above, and as illustrated in Fig. 3, we can picture our $N \times N$ lattice as a coarse $\sqrt{N} \times \sqrt{N}$ one, with lattice size $a_1$, where each cell consists in turn of $N = \sqrt{N} \times \sqrt{N}$ smaller cells with size $a$ and belonging to the finer lattice. Hydrodynamics emerges only at lengths larger than the site separation of the coarse lattice.

---

[18]One could naively think of defining momentum operators by taking the logarithm of translations: $\tilde{P}_I \equiv \frac{i}{a} \ln(T_I)$. Such $\tilde{P}_I$ are however elements of the algebra, and can be expressed as linear combinations of $J_{\mathbf{n}}$. They thus do not satisfy the required commutation relations $[\tilde{P}_J, \tilde{R}_{\mathbf{n}}] = i \frac{n^J}{r} \tilde{R}_{\mathbf{n}}$ necessary to interpret them as momentum operators. That is not surprising, given on a lattice only discrete translations make sense.

We are now ready to evaluate the consequences of the absence of the analogue of $R_I \equiv r P_I$ in the $SU(N)$ regulated fluid. On physical grounds, we expect the fluid description to be valid only at distances larger than a finite cut-off length. According to our discussion, this length should necessarily be larger or comparable to $a_1$. But as $2\pi r = a_1 \sqrt{N}$, the $N \to \infty$ limit, where ideally $SU(N) \to S\text{Diff}(T^2)$, also implies $r \to \infty$. The contribution of $R_I \equiv r P_I$ to the Hamiltonian of eq. (69) is then

$$\frac{P_I P_I}{2\rho(2\pi r)^2} \equiv \frac{P_I P_I}{2M_{tot}} \sim \frac{1}{N}, \tag{86}$$

and vanishes for $N \to \infty$ over states with finite total momentum. This is intuitively obvious: this contribution corresponds to the rigid motion of the whole fluid. At infinite volume the total mass of the fluid is infinite, so that the corresponding energy, eq. (86), and velocity $\bar{v}_I = P_I / M_{tot}$ vanish for finite $P_I$. We conclude that the only price to pay for the replacement of $S\text{Diff}(T^2)$ with $SU(N)$ is that our variables won't describe configurations with finite global velocity $\bar{v}_I$. This is not a real problem as these configurations can be recovered by performing a Lorentz (or Galilean) transformation.

## 5 Hilbert space and dynamics

By truncating $S\text{Diff}(T^2)$ to $SU(N)$, the quantum theory of a perfect fluid can be canonically constructed as described in section 2.2. The basis of the Hilbert space consists then of the states $|r, \alpha_r, \beta_r\rangle$, where $r$ labels the complete set of irreducible representation of $SU(N)$, while $\alpha_r = 1, \ldots, d_r$ and $\beta_r = 1, \ldots, d_r$ run on the $d_r$ basis states of each $r$ irrep. As $SU(N)_L$ and $SU(N)_R$ act respectively on $\alpha_r$ and $\beta_r$, the $SU(N)_L$ labels $\alpha_r$ are pure spectators in the computation of matrix elements of the velocity operator $v(x, t)$ and of the Hamiltonian, which purely depend on the generators of $SU(N)_R$. As discussed at the end of section 3, we could in principle do away with $SU(N)_L$ and consider a Hilbert space where only $SU(N)_R$ is represented. In so doing we would loose a path integral (or semiclassical) description of the system and, correspondingly, there would be no obvious rule establishing which representation $r$ must appear in the Hilbert space. The dynamics of $v(x, t)$ would however look the same. As already anticipated, we will not need to commit to one or the other approach. It will suffice to characterize the states purely by their $SU(N)_R$ quantum numbers. In practice, that means we will consider states $|r, \beta_r\rangle$, $\beta_r = 1, \ldots, d_r$ with $r$ *some* irrep of $SU(N)$.

In what follows we will study the physical properties of the basic $SU(N)$ representations. Our analysis is not mathematically comprehensive, in that we did not explicitly study or classify all the representations. However we believe it suffices to provide the physical picture for arbitrary states. Section 5.1 focuses on the kinematic properties of the states: by studying the vorticity matrix elements we will unveil their position space features. The results obtained there will provide physical intuition for the analysis in section 5.2, where we will introduce the Hamiltonian and study the spectrum.

### 5.1 Kinematics

We will here study the matrix elements of vorticity for some basic representations: the singlet, the fundamental, the antifundamental and the adjoint.

Let us consider the singlet representation first. There is obviously nothing to compute here: all the charges vanish and with them all the correlators of vorticity. The corresponding state, for any choice of a positive definite Hamiltonian quadratic in the charges, is also obviously the ground state. This simply generalizes to $SU(N)$ the known result for the ordinary $SO(3)$ rigid

body, for which the ground state wave function is a constant over the group manifold and all components of the angular momentum vanish. As we stated, here we are not considering the internal coordinates, corresponding to the $\phi^a$ fluid element labels and associated with the left action of the group: the ground state is just characterized by the vanishing of all the charges, hence of all velocity correlators. This is to be contrasted with ref. [28] where, at low energy and momenta, various 2-, 3- and 4-point functions were computed on the vacuum finding well defined and non-vanishing results.[19]

### 5.1.1 Fundamental and anti-fundamental representation

Consider now instead the more interesting case of the fundamental representation. We can label the basis states $|\alpha\rangle$ by an integer $\alpha \in [-\frac{N-1}{2}, \frac{N-1}{2}]$ and choose as generators the 't Hooft matrices of eq. (75): $\langle \alpha | J_{\mathbf{n}} | \beta \rangle = (J_{\mathbf{n}})^\alpha_\beta$. In order to offer a position space interpretation of the states, we must consider the action of the elementary translations $T_1$ and $T_2$. By the results of the previous section we have

$$T_1 |\alpha\rangle = (h^{-1})^\beta_\alpha |\beta\rangle = |\alpha + 1\rangle \,, \qquad T_2 |\alpha\rangle = g^\beta_\alpha |\beta\rangle = \omega^\alpha |\alpha\rangle \,. \tag{87}$$

The basis states $|\alpha\rangle$ are eigenstates of momentum in direction 2 with eigenvalue $p_2 = -\frac{\alpha}{r}$. At the same time, the first equation is compatible with these states being localized at position $x_1 = \alpha a$ in direction 1. In order to check that is indeed the case we consider the expectation value of vorticity at position $\mathbf{x} = \boldsymbol{\beta} a$ with $\boldsymbol{\beta} \in [-\frac{N-1}{2}, \frac{N-1}{2}]^2$. We find

$$\langle \alpha | \Omega(\mathbf{x}) |\alpha\rangle = \frac{N}{(2\pi)^3 \rho r^4} \sum_{\mathbf{n}} \omega^{-\mathbf{n}\boldsymbol{\beta}} \langle \alpha | J_{\mathbf{n}} |\alpha\rangle \tag{88}$$

$$= \frac{N}{(2\pi)^3 \rho r^4} \left( N \delta_{\beta_1, \alpha} - 1 \right) \tag{89}$$

$$\rightarrow \frac{\Lambda^2}{2\pi\rho} \frac{1}{2\pi r} \left[ \delta(x_1 - \alpha a) - \frac{1}{2\pi r} \right] \,, \tag{90}$$

where in the last line we took the continuum limit. The result corresponds to a vortex line localized at $x_1 = \alpha a$ with a compensating homogeneous vorticity density ensuring the vorticity integrates to zero over the full volume. We thus conclude that the basis states $|\alpha\rangle$ are eigenstates of momentum in direction 2 with eigenvalue $p_2 = -\frac{\alpha}{r}$ and are at the same time localized at position $x_1 = \alpha a$ in direction 1.

Alternatively, one can rotate to a basis of $T_1$ eigenstates via a discrete Fourier transform in $\alpha$,

$$|\tilde{\alpha}\rangle = \frac{1}{\sqrt{N}} \sum_\alpha \omega^{\alpha \cdot \tilde{\alpha}} |\alpha\rangle \equiv O^\alpha_{\tilde{\alpha}} |\alpha\rangle \,. \tag{91}$$

From their transformation under elementary translations,

$$T_1 |\tilde{\alpha}\rangle = \frac{1}{\sqrt{N}} \sum_\alpha \omega^{\alpha \cdot \tilde{\alpha}} |\alpha + 1\rangle = \omega^{-\tilde{\alpha}} |\tilde{\alpha}\rangle \,, \qquad T_2 |\tilde{\alpha}\rangle = \frac{1}{\sqrt{N}} \sum_\alpha \omega^{\alpha \cdot (\tilde{\alpha}+1)} |\alpha\rangle = |\tilde{\alpha} + 1\rangle \,, \tag{92}$$

---

[19]To be more precise ref. [28] considers the fluid in the general compressible regime, where the longitudinal mode is not decoupled. That makes the comparison less evident. On the other hand, the 2-point function for the transverse modes is not expected to be affected by the presence of the longitudinal mode, so that the non-vanishing result found in ref. [28] is in stark disagreement with our results. Now, as it turns out, the transverse correlator is found to be local, corresponding to the absence of propagating vorticity modes. It would then perhaps be interesting to see if this locality persists for all higher point functions including loop corrections. If that were the case, one could try and interpret the results of ref. [28] as the correlators in the singlet representation (where the vorticity dynamics is completely trivial) modulo local contact terms. We have not tried pursuing that route, as it would in any case not tell us anything about the more interesting dynamics of non-trivial representations.

we conclude $|\tilde{\alpha}\rangle$ has momentum $p_1 = \frac{\tilde{\alpha}}{r}$ and coordinate $x_2 = \tilde{\alpha}\, a$.[20] We thus see that, in both bases, the coordinate in one direction plays the role of momentum in the other. Therefore the states of the fundamental representation cannot be localised in both directions, at least not up to a fundamental lattice size. From the point of view of counting, that is obvious: the fundamental lattice has $N^2$ points, while the fundamental has only $N$ states. The non-commutativity of the two momentum components, and the resulting non-commutativity of the two position components, is quite the same encountered in the case of a particle with charge $e$ moving in a homogeneous magnetic field $B$.[21] In that case, states can be localized in both $x_1$ and $x_2$ only over an area $\sim 2\pi/eB$, which defines both the quantum unit of magnetic flux and the commuting finite translations. Similarly, see section 4, in our case translations by $\sqrt{N}$ fundamental sites $T_1^{\sqrt{N}}$ and $T_2^{\sqrt{N}}$ commute, and can be simultaneously diagonalized. There must therefore exists a basis of eigenstates of $T_J^{\sqrt{N}}$, which correspond to states localized on the coarse lattice cell of area $(a\sqrt{N})^2 = a_1^2$. The counting of states also supports this expectation, as the torus decomposes in precisely $N$ coarse cells.

In order to construct the localized states, we split the fundamental representation index $\alpha \in [-\frac{N-1}{2}, \frac{N-1}{2}]$ according to $\alpha = \alpha_1\sqrt{N} + \alpha_2$, where $\alpha_{1,2} \in [-\frac{\sqrt{N}-1}{2}, \frac{\sqrt{N}-1}{2}]$. States with definite values of both momenta mod $\sqrt{N}$ are then given by a Fourier transform in $\alpha_1$:

$$|\mathbf{n}\rangle_\square \equiv \frac{1}{N^{1/4}} \sum_{\alpha_1}^{\sim} \omega^{\sqrt{N}\alpha_1 n_1 - \frac{1}{2}n_1 n_2} \left|\alpha_1\sqrt{N} - n_2\right\rangle . \tag{93}$$

Notice that, unlike for the adjoint, here $\mathbf{n} \equiv (n_1, n_2)$ belongs to the coarse lattice with the origin $\mathbf{n} = 0$ included: $\mathbf{n} \in [-\frac{\sqrt{N}-1}{2}, \frac{\sqrt{N}-1}{2}]^2$. We have added a $\sim$ on the summation symbol to account for that. In order to avoid confusion with the momentum indices of the adjoint, which run instead on the fundamental lattice, we have added the subscript $\square$ to stress we are considering the fundamental. The $|\mathbf{n}\rangle_\square$ are eigenvectors of both translations $T_J^{\sqrt{N}}$ with eigenvalues $\omega^{-\sqrt{N}n_J} = e^{-i\frac{n_J}{r}a_1}$ and provide an orthonormal basis:

$$\left\langle\mathbf{n}'\middle|\mathbf{n}\right\rangle_\square = \delta_{n_1}^{n_1'} \delta_{n_2}^{n_2'} . \tag{94}$$

Notice that the extra phase factor $\omega^{-\frac{1}{2}n_1 n_2}$ in eq. (93) was introduced so as to ensure that $O$, defined in eq. (91), properly realises $\pi/2$ rotations:

$$\left\langle\mathbf{n}'\middle| O \middle|\mathbf{n}\right\rangle_\square = \delta_{-n_2}^{n_1'} \delta_{n_1}^{n_2'} . \tag{95}$$

The position eigenstates are now obtained by performing a Fourier transform mod $\sqrt{N}$ on both momentum labels:

$$|\boldsymbol{\alpha}\rangle_\square \equiv \frac{1}{N^{1/2}} \sum_{\mathbf{n}}^{\sim} \omega^{-\sqrt{N}(\alpha_1 n_1 + \alpha_2 n_2)} |\mathbf{n}\rangle_\square . \tag{96}$$

---

[20]Indeed, given $O\,g\,O^{-1} = h$ and $O\,h\,O^{-1} = g^{-1}$, and also given $\left(O^2\right)^\alpha_\beta = \delta^\alpha_{-\beta}$ is a reflection, the unitary matrix $O$ realises a $\pi/2$ rotation.

[21]What we are encountering here is more than an analogy. Indeed the relevance of $SU(N)$ and, more generally, of quantum modified volume preserving diffeomorphisms have been known for quite some time to play a central role in the description of quantum Hall systems as well as of rotating superfluids (which in $D = 2+1$ are equivalently described by charged point particles in an external magnetic field). Refs. [49–51] represent a far from exhaustive list of relevant previous work. Part of our results are then not entirely new, but, as far as we can see, the perspective (in particular the link between the quantum perfect fluid and the rigid body) and the main results on the spectrum and the dynamics are new. We are now motivated to better explore the link between our construction and previous ones, especially those concerning quantum Hall systems [52]. That is also in order to understand if our results apply to systems that can be engineered in the lab.

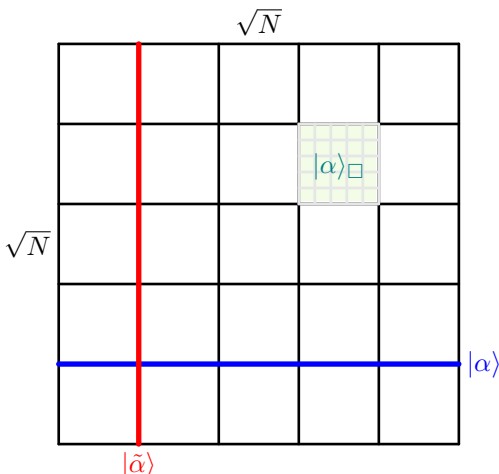

Figure 4: Representation of the localization of the different fundamental states whose transformations under translations are described in eqs. (87,92,97).

Again $\boldsymbol{\alpha} \equiv (\alpha_1, \alpha_2)$ are integer coordinates on the coarse lattice $[-\frac{\sqrt{N}-1}{2}, \frac{\sqrt{N}-1}{2}]^2$, corresponding to space coordinates $\boldsymbol{x} = \boldsymbol{\alpha} a_1$. These states have the same $\delta$-function normalization as in eq. (94). Moreover the action of translations indicates they are localized on the coarse lattice at $(\alpha_1, \alpha_2)$:

$$T_1^{\sqrt{N}} |(\alpha_1, \alpha_2)\rangle_\square = |(\alpha_1 + 1, \alpha_2)\rangle_\square \,, \qquad T_2^{\sqrt{N}} |(\alpha_1, \alpha_2)\rangle_\square = |(\alpha_1, \alpha_2 + 1)\rangle_\square \,. \qquad (97)$$

Fig. (4) illustrates the position space localization properties of the different bases of the fundamental representation. To confirm the interpretation of $|\boldsymbol{\alpha}\rangle_\square$ as localized states we again consider the expectation value of the vorticity. Without loss of generality we can focus on $|\boldsymbol{\alpha} = 0\rangle_\square$. For the vorticity at $\boldsymbol{x} = \boldsymbol{\beta} a$ we can then write ($\boldsymbol{\beta}$ labels operators in the adjoint and thus takes values on the fine lattice)

$$\langle \boldsymbol{\alpha} = 0 | \Omega(\boldsymbol{x}) | \boldsymbol{\alpha} = 0 \rangle_\square = \frac{N}{(2\pi)^3 \rho r^4} \sum_{\mathbf{n}}{}' \omega^{-\mathbf{n}\boldsymbol{\beta}} \langle \boldsymbol{\alpha} = 0 | J_{\mathbf{n}} | \boldsymbol{\alpha} = 0 \rangle_\square \qquad (98)$$

$$= \frac{N}{(2\pi)^3 \rho r^4} [F(\boldsymbol{\beta}, N) - 1]$$

$$\to \frac{\Lambda^2}{2\pi\rho} \left[ \delta^2(\boldsymbol{x}) - \frac{1}{(2\pi r)^2} \right],$$

where $F(\boldsymbol{\beta}, N)$ is an expression involving multiple sums over trigonometric functions and where in the last step we have taken the continuum limit.[22] The derivation of the last step and its meaning are explained as follows. $F(\boldsymbol{\beta}, N)$ is easily seen to satisfy the sum rule

$$\sum_{\boldsymbol{\beta}} F(\boldsymbol{\beta}, N) = N^2 \,. \qquad (99)$$

Furthermore, by a numerical study,[23] we could conclude that for large $N$ its behaviour is roughly

$$F(\boldsymbol{\beta}, N) \sim N f(\boldsymbol{\beta}/\sqrt{N}), \qquad (100)$$

---

[22]In the limit $N \to \infty$ with $\Lambda$ fixed we have $r \to \infty$ so that the second term in square brackets should be dropped. We kept it to maintain the information that the integral of $\Omega$ vanishes at any $N$.

[23]We explicitly computed $F$ using *Mathematica* up to $N = 15^2$. There sure must exist tricks to compute/estimate $F$ analytically, but we think our numerical study is sufficient.

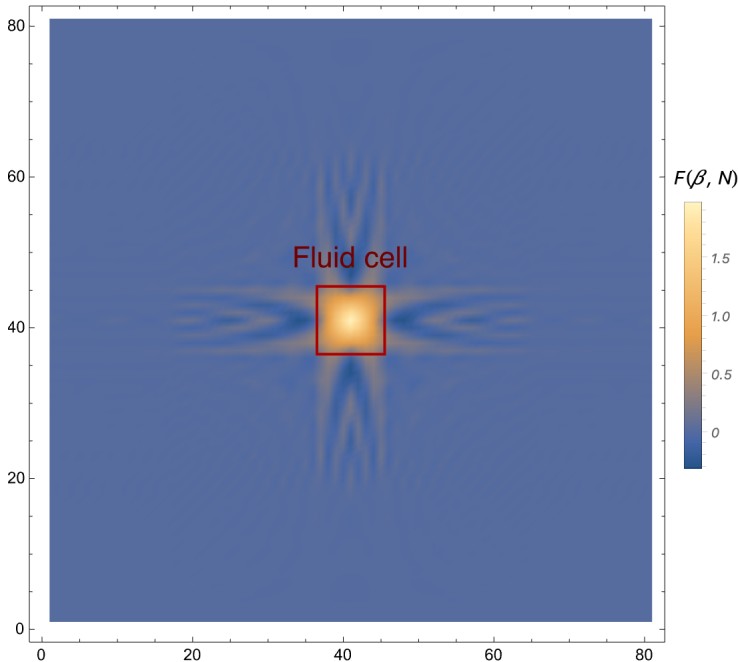

Figure 5: Numerical estimate of $F(\beta, N)$ defining the vorticity expectation value of localized fundamental states. We have $N = 81$, with a coarse lattice composed of $9 \times 9$ fluid cells.

with $f(\gamma) = O(1) > 0$ for $|\gamma| \lesssim O(1)$ and $f(\gamma) \to 0$ rapidly for $|\gamma| > O(1)$. On scales larger than the coarse lattice cell we can thus approximate

$$F(\beta, N) \sim N^2 \delta^2(\beta) = (Na)^2 \delta^2(x) = (2\pi r)^2 \delta^2(x), \tag{101}$$

from which the last line in eq. (98) follows. Fig. 5 shows the plot of $F(\beta, N)$ for $N = 81$.

To sum up, the states of the fundamental representation can be viewed as vortices localised on the cells of the coarse lattice, immersed in a compensating uniform background with negative vorticity. Expectedly, the localization area, equalling $\sqrt{N} \times \sqrt{N} = N$ fundamental lattice cells, is the same as for the $|\alpha\rangle$ states. The latter are indeed fully localized on the fundamental lattice in one direction and fully delocalized in the other, corresponding to $1 \times N = N$ fundamental cells. On the coarse lattice we can thus picture a vortex as consisting of one quantum of vorticity $(\frac{\Lambda^2}{2\pi\rho})$ on a single cell superimposed to a compensating homogeneous background carrying $-1/N$ of the fundamental quantum on all the cells.[24]

The velocity field corresponding to eq. (98) is a circular flow around the location of the vortex. For a vortex at located at $(0, 0)$, the velocity at $x$ in the range $a_1 \ll |x| \ll 2\pi r$ is well approximated by

$$v = \frac{\Lambda^2}{4\pi^2\rho|x|} = \frac{1}{\rho a_1^2 |x|}, \tag{102}$$

and vanishes exactly for either $|x_1| = \pi r$ or $|x_2| = \pi r$ as a result of the compensating homogeneous negative vorticity. The maximal value $\sim \frac{1}{\rho a_1^3}$ is attained at the edge of the fluid element cell, at $|x| \sim a_1$.

---

[24]Another system containing quantized vortices is the superfluid. For comparison, in a weakly coupled non-relativistic BEC superfluid, the vorticity of a quantum vortex equals $\frac{2\pi}{m}$ where $m$ is the mass of the condensed boson. It matches our result $\frac{\Lambda^2}{2\pi\rho}$ with $m$ replaced by the mass of the elementary fluid cell $\rho a_1^2$.

The vorticity expectation value matches the classical picture for the fluid flow around a vortex, though these states are far from being semiclassical. One can see that by calculating higher point vorticity correlators.[25] To make the point it is sufficient to consider the simplest ones focusing on the $|\alpha\rangle$ states. For instance for $\langle\alpha|\Omega^2(\boldsymbol{x})|\alpha\rangle$, in the continuum limit we find

$$\langle\alpha|\Omega^2(\boldsymbol{x})|\alpha\rangle = 2\left(\frac{\Lambda^4}{(2\pi)^3\rho}\right)^2 \Pi\left(\frac{x_1-\alpha a}{\pi r}\right), \tag{103}$$

where $\Pi(t)$ is the periodic rectangle function defined by $\Pi(t+2) = \Pi(t)$ and by $\Pi(t) \equiv \theta(t+1/2)\theta(1/2-t)$ for $t\in[-1,1]$. Comparing this result to eq. (88), we see how in the large volume limit $\langle\Omega(\boldsymbol{x})^2\rangle \gg (\langle\Omega(\boldsymbol{x})\rangle)^2$, indicating the fully quantum nature of the flow in the single vortex states. In fact $\langle\Omega(\boldsymbol{x})\rangle \to 0$ at infinite volume, while $\langle\Omega(\boldsymbol{x})^2\rangle$ is finite and purely determined by $\Lambda$ and $\rho$ through dimensional analysis. The value of $\langle\Omega(\boldsymbol{x})^2\rangle$ is UV dominated at momenta $\gg \Lambda$, which is the regime where our system cannot be properly considered a fluid. A more proper observable to consider is then vorticity smeared over distances larger than the fluid cut-off. For instance considering

$$\bar{\Omega}(\boldsymbol{x},L) \equiv \frac{1}{(L+1)^2}\sum_{\beta_1=-L/2}^{\beta_1=L/2}\sum_{\beta_2=-L/2}^{\beta_2=L/2}\Omega(\boldsymbol{x}+a\boldsymbol{\beta}), \tag{104}$$

with $L \gtrsim \sqrt{N}$, one finds

$$\frac{\langle\bar{\Omega}^2\rangle - (\langle\bar{\Omega}\rangle)^2}{\langle\bar{\Omega}^2\rangle} = O(1). \tag{105}$$

This results shows the quantum nature of the flow even at physical scales.

Since every $SU(N)$ representation can be constructed as a tensor product of fundamentals, every state of the quantum fluid can be viewed as a direct product of the localized vortex states $|\boldsymbol{\alpha}\rangle_\square$. This fact offers a direct physical picture for the general representations. For instance, the states of the anti-fundamental representation can be viewed as totally antisymmetric products of $N-1$ vortices. Because of the total antisymmetry, the constituent vortices have to be localised on $N-1$ different coarse lattice cells leaving a single unoccupied cell. As the vorticity is given by the sum of the individual vorticities of the $N-1$ vortices, it will result in one negative quantum of vorticity at the 'empty' cell, superimposed to a homogeneous positive $1/N$ background. That is minus the vorticity of the fundamental representation. Antifundamental states thus correspond to elementary anti-vortices.

This last result can also be directly derived by considering that the generators $\bar{J}_{\mathbf{n}}$ in the anti-fundamental are related to those in the fundamental by $\bar{J}_{\mathbf{n}} = -J_{\mathbf{n}}^T$. Defining the action of parity on a vector as $(v_1,v_2) = \boldsymbol{v} \to \boldsymbol{v}_P = (v_1,-v_2)$, by eqs. (74,75) we have $\bar{J}_{\mathbf{n}} = -J_{\mathbf{n}_P}$. Using this result one can then define in the Hilbert space a parity operator $P$ satisfying

$$P^2 = 1, \qquad |\boldsymbol{\alpha}_P\rangle_{\bar{\square}} = P|\boldsymbol{\alpha}\rangle_\square, \qquad P\Omega(\boldsymbol{x})P = -\Omega(\boldsymbol{x}_P), \tag{106}$$

so that

$$\langle\alpha|\Omega(\boldsymbol{x})|\alpha\rangle_{\bar{\square}} = -\langle\alpha_P|\Omega(\boldsymbol{x}_P)|\alpha_P\rangle_\square = -\langle\alpha|\Omega(\boldsymbol{x})|\alpha\rangle_\square, \tag{107}$$

where in the last step we used that eq. (98) is invariant under $\boldsymbol{x} \to \boldsymbol{x}_P$. By applying this results to tensor products of fundamentals or anti-fundamentals we conclude that conjugated representations have equal and opposite vorticity expectation values.

---

[25]Curiously, the generating functionals $Z[f(\boldsymbol{x});\psi] = \langle\psi|e^{i\sum_{\mathbf{m}}f_{\mathbf{m}}J_{\mathbf{m}}}|\psi\rangle \equiv \langle\psi|e^{i\sum_{\boldsymbol{x}}f(\boldsymbol{x})J(\boldsymbol{x})}|\psi\rangle$ that would produce all the correlation functions of vorticity in states $|\psi\rangle$ are matrix elements of an $SU(N)$ operator as a function of group parameters $f(\boldsymbol{x})$. They are given by the $SU(N)$ generalization of spherical harmonics.

### 5.1.2 Adjoint representation

As will become clear below, among all representations, a central role is played by the self-conjugated ones, which can be written as tensor products of an equal number of fundamentals and anti-fundamentals. The smallest non-trivial representation in this class is the adjoint. Indeed, as all self-conjugated representations can be written as tensor products of adjoints, the adjoint is a crucial building block.

In order to gain insight into the physical properties of the adjoint, it is useful to view it alternatively either as the Lie algebra itself or as the tensor product $\square \otimes \overline{\square}$.

Let us consider the first perspective. We can label the $N^2 - 1$ basis states by the charges themselves: $J_{\mathbf{n}} \rightarrow |\mathbf{n}\rangle$. Choosing the normalization $\langle \mathbf{m} | \mathbf{n} \rangle = \delta_{\mathbf{m},\mathbf{n}}$, the matrix elements are then simply given by the structure constants:

$$\langle \mathbf{m} | J_{\mathbf{k}} | \mathbf{n} \rangle = -2i \sin\left(\frac{\pi}{N}(\mathbf{k} \wedge \mathbf{n})\right) \delta_{\mathbf{m},\mathbf{k}+\mathbf{n}}. \tag{108}$$

The basis states are also obviously eigenstates of translations with momentum $\mathbf{n}/r$. The adjoint representation renders clear what was somehow to be expected by replacing $S\mathrm{Diff}(T^2)$ with $SU(N)$: in general $S\mathrm{Diff}(T^2)$ is well approximated by $SU(N)$ only on a subset of a given representation of $SU(N)$. In the case of the adjoint, the subset is clearly given by the states $|\mathbf{n}\rangle$ with $|\mathbf{n}| \lesssim \sqrt{N}$. States labelled by $|\mathbf{n}| \gtrsim \sqrt{N}$ should be then interpreted as spurious states describing physics in a domain outside the universality class of the quantum perfect fluid. It will however be useful to contemplate for a moment the properties of these other states. One should also wonder about the need for an analogue truncation on the states of $\square$ and $\overline{\square}$. The study of the adjoint will automatically address that question.

Consider now the second perspective: $\mathbf{Adj} \oplus \mathbf{1} = \square \otimes \overline{\square}$. Using the coarse lattice basis of eq. (96) for $\square$ and $\overline{\square}$ and applying standard group theory we can immediately write

$$\mathbf{1}: \qquad |\bullet\rangle = \frac{1}{\sqrt{N}} \sum_{\alpha} |\alpha\rangle_{\square} \otimes |\alpha\rangle_{\overline{\square}}, \tag{109}$$

$$\mathbf{Adj}: \qquad |\mathbf{n}\rangle = \frac{1}{\sqrt{N}} \sum_{\alpha\beta} |\alpha\rangle_{\square} \otimes |\beta\rangle_{\overline{\square}} \langle \alpha | J_{\mathbf{n}} | \beta \rangle_{\square}. \tag{110}$$

We picked here a specific basis for the adjoint, but in general its states are obtained by contracting $|\alpha\rangle_{\square} \otimes |\beta\rangle_{\overline{\square}}$ with a traceless coefficient function $f^{\alpha}_{\beta}$: $\sum_{\alpha} f^{\alpha}_{\alpha} = 0$.

According to eq. (109), we can picture the trivial configuration as a homogeneous and maximally entangled superposition of states with a vortex and an antivortex placed on each cell of the coarse lattice, so as to make vorticity vanish exactly. According to eq. (110), states in the adjoint are instead given by traceless superpositions. Notice, however, that in the infinite volume limit we can choose smooth traceless functions $f(\alpha, \beta)$ that approximate the identity coefficient $\delta_{\alpha,\beta}$ over any local but arbitrarily large region. This hints that low momentum states belonging to the adjoint physically approximate the vacuum configuration. As we shall see in the next subsection, that is indeed the case when considering energy levels.

When considering, as we argued, that only for $|\mathbf{n}| \lesssim \sqrt{N}$ do adjoint states $|\mathbf{n}\rangle$ represent a consistent truncation of $S\mathrm{Diff}(T^2)$, eq. (110) has even more dramatic consequences. Indeed by the constraint $|\mathbf{n}| \lesssim \sqrt{N}$ only $O(N)$ of the $N^2 - 1$ states in the right-hand side of eq. (110) should be retained: if the vast majority of the states in their tensor product are unphysical we are led to conclude that also $\square$ and $\overline{\square}$ do not represent a consistent truncation of $S\mathrm{Diff}(T^2)$ and should therefore be discarded. As we shall now show, the study of the adjoint vorticity profile confirms this conclusion.

To study vorticity in the adjoint representation it is useful to consider wave packets. First

of all it is worth appreciating that the standard definition of the position eigenstates,

$$|\boldsymbol{x}\rangle \equiv \frac{1}{Na}{\sum_{\mathbf{n}}}' \omega^{-\mathbf{n}\cdot\boldsymbol{\alpha}} |\mathbf{n}\rangle\,, \qquad \boldsymbol{x} \equiv a\boldsymbol{\alpha}\,, \tag{111}$$

leads in the $N \to \infty$ limit to the expected normalization (again $\boldsymbol{x} \equiv a\boldsymbol{\alpha}$ and $\mathbf{y} \equiv a\boldsymbol{\beta}$):

$$\langle \boldsymbol{x}|\mathbf{y}\rangle = \frac{1}{a^2}\delta_{\boldsymbol{\alpha},\boldsymbol{\beta}} - \frac{1}{(Na)^2} \to \delta^2(\boldsymbol{x}-\mathbf{y})\,. \tag{112}$$

As for $N \to \infty$ things work as expected, we can use standard intuition. A state localized around the origin in position space and around momentum $\mathbf{p} \equiv \mathbf{m}/r$ in momentum space, can then be constructed using a gaussian wave packet

$$|\Psi\rangle \equiv {\sum_{\mathbf{n}}}' \psi(\mathbf{n}) |\mathbf{n}\rangle\,, \qquad \hat{\psi}(\mathbf{n}) = \frac{c}{\eta\sqrt{2\pi}} e^{-\frac{(\mathbf{n}-\mathbf{m})^2}{4\eta^2}}\,, \tag{113}$$

where $c$ is a normalization coefficient which, given $\langle\Psi|\Psi\rangle = 1$, tends to 1 when $N \to \infty$. For the state to be localized around $\mathbf{m}$ we also must impose $\eta \lesssim |\mathbf{m}|$. In position space the state will be localized over $\sim N/\eta$ sites of the fine lattice. As we are interested in localizations over distances $\gtrsim a_1$ and $\ll r$, we then further require $1 \ll \eta \lesssim \sqrt{N}$. Putting all these requests together, we then impose $1 \ll \eta \lesssim \min(|\mathbf{m}|, \sqrt{N})$. Notice finally that, for $|\mathbf{m}| \ll N$, the momenta that dominate the wave-packet consequently satisfy $1 \ll |\mathbf{n}| \ll N$. This implies we can take the continuum limit both in position and momentum space and safely approximate sums with integrals. For the vorticity expectation value we then find

$$\langle\Psi|\Omega(\boldsymbol{x})|\Psi\rangle \simeq \frac{\Lambda^2}{4\pi^2\rho\sigma^2}\left[e^{-\frac{1}{2\sigma^2}(\boldsymbol{x}+\frac{\pi\tilde{\mathbf{p}}}{\Lambda^2})^2} - e^{-\frac{1}{2\sigma^2}(\boldsymbol{x}-\frac{\pi\tilde{\mathbf{p}}}{\Lambda^2})^2}\right]\,, \tag{114}$$

where $\sigma = 2\pi r/\eta$ and, as before, $\tilde{p}^I = \epsilon^{IJ}p^J$. This corresponds to a configuration with a vortex and anti-vortex of unit vorticity $\Lambda^2/2\pi\rho$ centered respectively at $\mp\pi\tilde{\mathbf{p}}/\Lambda^2$. This result has crucial consequences on the physical interpretation of the states.

For $|\mathbf{p}| \gtrsim \Lambda$ the separation between vortex and antivortex is larger than the cut-off length $a_1 = 1/\Lambda$, which makes them individually resolvable within the hydrodynamic description. On the other hand, as we argued, adjoint states with momentum $|\mathbf{p}| \gtrsim \Lambda$ are outside the range where $S\mathrm{Diff}(T^2)$ is well approximated by $SU(N)$ and must therefore be discarded when zooming on the perfect fluid universality class. We should then conclude that vortices and antivortices, that is $\square$ and $\overline{\square}$, should also not be part of the long distance description. Indeed the study of the dynamics in the next subsection will offer an alternative motivation for discarding them: these states will turn out to be gapped. Notice, finally, that the fact that vortices should be discarded on physical grounds does not make the detailed discussion of their properties useless. Vortices are still relevant as group theoretic building blocks.

Consider now instead the regime $|\mathbf{p}| \ll \Lambda$. Here vortex and antivortex are separated by a distance smaller that the cut-off length $a_1 = 1/\Lambda$ and cannot be individually resolved. The only resolvable feature of vorticity in this regime is its dipole

$$d^I = \int d^2x\, x^I\, \langle\Psi|\Omega(\boldsymbol{x})|\Psi\rangle = -\frac{\tilde{p}_I}{\rho}\,. \tag{115}$$

Even this dipole vanishes for $|\mathbf{p}| \to 0$, which is possible when $N \to \infty$. This confirms, as argued above, that adjoint states approximate the trivial configuration as their momentum goes to zero. It is also useful to write the vorticity expectation value in terms of the position space wave function

$$\psi(\boldsymbol{x}) \equiv \langle\boldsymbol{x}|\Psi\rangle \simeq \frac{1}{\sqrt{2\pi\sigma^2}} e^{-\frac{\boldsymbol{x}^2}{4\sigma^2}+i\mathbf{p}\cdot\boldsymbol{x}}\,. \tag{116}$$

Expanding eq. (114) at first order in **p** we can then write

$$\langle\Psi|\,\Omega(\boldsymbol{x})\,|\Psi\rangle = \frac{i\,\partial\,\psi\wedge\partial\,\psi^*}{\rho}\,. \tag{117}$$

This last result purely relies on the $S$Diff algebra. To check that, it is instructive to apply the results of section 3. All the elements in eq. (117) (the bra, the ket and the operator) belong to the adjoint of $S$Diff. Thus each of them corresponds to a charge $R_f$ defined according to eq. (53) with $f^I \equiv -\tilde{\partial}^I f$. In particular $|\Psi\rangle$ corresponds to $R_\psi$ with $\psi^I \equiv -\tilde{\partial}^I\psi$, $\Omega(\boldsymbol{x})$ correspond to $R_{\delta_x}$ with $\delta_x^I(\mathbf{y}) = \tilde{\partial}_y^I\delta^2(\mathbf{y}-\boldsymbol{x})/\rho$ and finally $\langle\Psi|$ corresponds to $R_{\psi^*} = R_\psi^\dagger$. The state $|\Phi_x\rangle \equiv \Omega(\boldsymbol{x})\,|\Psi\rangle$ is then associated with the adjoint element $[R_{\delta_x}, R_\psi]$ and by eqs. (54) and (55) its wavefunction $\varphi_x(\mathbf{y})$ is given by (all **y**-derivatives)

$$[\delta_x, \psi]^I(\mathbf{y}) = -\tilde{\partial}^I(i\tilde{\partial}^J\psi(\mathbf{y})\partial_J\delta^2(\mathbf{y}-\boldsymbol{x})/\rho) \equiv -\tilde{\partial}^I\varphi_x(y)\,. \tag{118}$$

We can then write

$$\langle\Psi|\,\Omega(\boldsymbol{x})\,|\Psi\rangle = \langle\Psi|\Phi_x\rangle = \frac{i}{\rho}\int d^2\mathbf{y}\,\psi^*(\mathbf{y})\partial_y\delta^2(\mathbf{y}-\boldsymbol{x})\wedge\partial\,\psi(\mathbf{y})\,, \tag{119}$$

which upon integration by parts gives back eq. (117).

What is the main message of this study? We knew right at the beginning that only a relatively tiny subset of the generators of $SU(N)$ approximates the low momentum generators of $S$Diff$(T^2)$. Specifically that is roughly $N$ out of $\sim N^2$ generators. In view of that, when considering a generic representation of $SU(N)$ we should similarly have expected that only a small portion should be retained as a sensible realization of the $S$Diff$(T^2)$ algebra. We have now seen directly that is indeed the case. The study of the adjoint indicates that states with resolvable vortices and anti-vortices, in particular the fundamental and anti-fundamental, should be dismissed. The symmetric and antisymmetric products of two fundamentals are also in the same class: in the product state, vortices are either separated or overlapped to produce another vortex with twice the vorticity. One can similarly argue as concerns the product of any number $< N$ of fundamentals. Following this indication, the only states we can retain must consist of unresolvable vortex anti-vortex pairs and belong to representations that can be written as tensor products of adjoints. This subclass of $SU(N)$ representations indeed corresponds to the Young tableaux whose number of boxes equals a multiple of $N$, i.e. that can be written as a tensor product of a multiple of $N$ fundamentals. These trivially realize the $Z_N$ center of $SU(N)$ and provide a faithful representation of $PSU(N) = SU(N)/Z_N$. In that sense our procedure should now more properly be viewed as a truncation of $S$Diff$(T^2)$ to $PSU(N)$. The adjoint states with momentum $|\mathbf{p}| \lesssim \Lambda$ can now be viewed as the group theoretic building blocks of all physical states. In momentum space these states are counted to be $O(N)$. The wave packets constructed in this section give, for $\sigma \sim a_1$, states localized on the coarse lattice and a compatible counting. These building blocks carry a dipole of vorticity orthogonal to their momentum, see eq. (115). The corresponding flow becomes thus indistinguishable from the trivial one at zero momentum, as it is expected for the hydrodynamic modes of a finite density system. All this indicates these states, which we will henceforth call *vortons,* are also the basic quanta of the dynamics. The rest of the paper will confirm that is indeed the case.

## 5.2 Hamiltonian and spectrum

We are finally ready to construct the dynamics of the $SU(N)$ regulated quantum fluid. The basic idea is that the dynamics of the $SU(N)$ charges $\tilde{R}_\mathbf{n}$ approximate the dynamics of the perfect fluid charges $R_\mathbf{n}$ in the low-momentum regime $|\mathbf{n}| \lesssim \sqrt{N}$ where their commutation

relations coincide. To realize this idea we start by considering a Hamiltonian quadratic in the $\tilde{R}_{\mathbf{n}}$ and satisfying translation invariance, which we can write most generally as

$$H = \frac{1}{2\rho(2\pi)^2 r^4} {\sum_{\mathbf{n}}}' \frac{F(\mathbf{n})}{\mathbf{n}^2} \tilde{R}_{\mathbf{n}} \tilde{R}_{-\mathbf{n}}. \tag{120}$$

$F(\mathbf{n})$ is a regulator function which can be viewed as defining the moment of inertia of the $SU(N)$ rigid body: more precisely we have the inverse proportionality $I_{\mathbf{n},\mathbf{n}} \propto \mathbf{n}^2/F(\mathbf{n})$. The equation of motion then takes the form ($\tilde{\Omega}_{\mathbf{n}} \equiv -\tilde{R}_{\mathbf{n}}/(\rho r^2)$)

$$\dot{\tilde{\Omega}}_{\mathbf{n}} = \frac{1}{2(2\pi r)^2} {\sum_{\mathbf{m}}}' \frac{F(\mathbf{m})}{\mathbf{m}^2} \frac{N}{\pi} \sin\left(\frac{\pi}{N}(\mathbf{n} \wedge \mathbf{m})\right) \left(\tilde{\Omega}_{-\mathbf{m}}\tilde{\Omega}_{\mathbf{n}+\mathbf{m}} + \tilde{\Omega}_{\mathbf{n}+\mathbf{m}}\tilde{\Omega}_{-\mathbf{m}}\right). \tag{121}$$

This matches the long distance part of the Euler equation, eq. (70) at $\bar{v}_I = 0$, provided $F(\mathbf{m}) = 1$ for $|\mathbf{m}| \lesssim \sqrt{N}$ and $F(\mathbf{m}) \to 0$ for $|\mathbf{m}| \gtrsim \sqrt{N}$. The second request guarantees that the charges $\tilde{R}_{\mathbf{m}}$, with $|\mathbf{m}| \gtrsim \sqrt{N}$ decouple from the macroscopic dynamics. In the rigid body analogy, this choice corresponds to assigning an infinite moment of inertia to the UV modes. This separates our work from previous studies that have considered the relation between the $SU(N)$ rigid body and the perfect 2D fluid. We take an effective field theory perspective and, as a consequence, we are drawn to this UV regulation of the Hamiltonian. Moreover, even though at finite $N$ rotational symmetry is broken both microscopically, by the fine lattice, and globally, by the torus geometry, we would like to preserve it in the limit $N \to \infty$ with $a_1$ fixed. This can be simply achieved by choosing $F(\mathbf{n}) \equiv G(|\mathbf{n}|)$. We will thus consider systems defined by a regulator function $F(\mathbf{n})$ satisfying

$$F(\mathbf{n}) \equiv G(|\mathbf{n}|), \qquad F(\mathbf{n}) = 1, \text{ for } |\mathbf{n}| \ll \sqrt{N}, \qquad F(\mathbf{n}) \to 0, \text{ for } |\mathbf{n}| \gtrsim \sqrt{N}. \tag{122}$$

A possible choice is

$$F(\mathbf{n}) = \left(\frac{N}{\mathbf{n}^2 + N}\right)^p, \quad p > 0. \tag{123}$$

These requests define a family of UV completions of quantum hydrodynamics. We shall also illustrate what changes by relaxing some of these constraints and show that the crucial aspects of IR physics are shared by a broader class of Hamiltonians.

In order to make the scaling at large $N$ explicit, it is convenient to rewrite the Hamiltonian in terms of the canonically normalized $SU(N)$ generators $J_{\mathbf{n}}$. By eqs. (78) and (80) we can then write

$$H = \frac{\Lambda^4}{32\pi^4\rho} {\sum_{\mathbf{n}}}' \frac{F(\mathbf{n})}{\mathbf{n}^2} J_{\mathbf{n}} J_{-\mathbf{n}}. \tag{124}$$

The prefactor $\frac{\Lambda^4}{\rho}$ coincides, as indeed dictated by dimensional analysis, with Landau's guess [29,53] for the gap of the transverse modes. Indicating by $m = \rho a_1^2$ the mass of a fluid element, i.e. the mass in a coarse lattice cell, we can associate the energy $\frac{\Lambda^4}{4\pi^2\rho} = \Lambda^2/m$ to the motion of a fluid element with cut-off scale momentum.

Let us now study the spectrum of the Hamiltonian of eq. (124) focusing on the simplest $SU(N)$ representations. The very simplest one, the singlet, obviously has zero energy and provides the ground state of the system. Consider next the fundamental. The generators, as discussed in subsection 5.1.1, are the 't Hooft matrices of eq. (75). Since $J_{-\mathbf{n}}J_{\mathbf{n}} = \mathbb{1}$, the Hamiltonian is here proportional to the identity operator, all the $N$ states are degenerate: $H|\alpha\rangle = E_{\square}|\alpha\rangle$, with

$$E_{\square} = \frac{\Lambda^4}{32\pi^4\rho} {\sum_{\mathbf{n}}}' \frac{F(\mathbf{n})}{\mathbf{n}^2} \simeq \frac{\Lambda^4}{16\pi^3\rho} \ln(\sqrt{N}). \tag{125}$$

Here, in the last step, we used that $F(\mathbf{n})$ cuts off the logarithmic divergent sum, roughly at $|\mathbf{n}| \sim \sqrt{N}$. Of course while the log term is robust for large $N$, the finite $O(1)$ contributions depend on the detailed choice of $F$. In terms of the physical length scales we can equivalently write

$$E_\square = \frac{\Lambda^4}{16\pi^3\rho}\ln(\Lambda r). \tag{126}$$

We see that the overall scale of the energy matches the gap predicted by Landau. However, the result is enhanced by a logarithmic factor that depends on the size of the system. This result does not come unexpected, as we have already seen that the fundamental representation describes localized vortices: a logarithmic dependence of the energy on the total area is typical of vortex flows where the velocity falls off with the inverse of the distance, see eq. (102). Indeed inserting eq. (102) in the classical energy density $\rho v^2/2$ we find

$$E_{vortex} \simeq \frac{1}{2}\rho \int_{a_1}^{2\pi r} d^2x\, \mathbf{v}^2 \simeq \frac{\Lambda^4}{16\pi^3\rho}\ln(2\pi r/a_1), \tag{127}$$

in perfect agreement with eq. (126).

As the sum in eq. (125) depends logarithmically on the number of modes over which $F(\mathbf{n})$ is approximately constant, the result does not change qualitatively when a different choice for $F(\mathbf{n})$ is made. We are of course excluding extreme choices like, for instance, $F(\mathbf{n})$ growing in the UV. For example, if one extends $F(\mathbf{n})$ to be constant for all $\mathbf{n}$, the final answer will change by a factor of two. Note that, as the states in the fundamental are exactly degenerate, the dynamics of this sector is trivial: any superposition is also an energy eigenstate and all observables have time-independent expectation values.

The above discussion carries out unchanged to the antifundamental, given its generators satisfy $\bar{J}_\mathbf{n} = -J_\mathbf{n}^T$. Thus we have again $\bar{J}_{-\mathbf{n}}\bar{J}_\mathbf{n} = 1 \ \forall \mathbf{n}$ so that $E_{\bar{\square}} = E_\square$. Moreover, since all other unitary irreducible representations can be obtained by considering tensor products of the defining representation, the same argument shows that the energy spectra of any pair of conjugate representations coincide.

The fact that the fundamental representation is gapped at the energy cut off $\Lambda^4/\rho$ matches our conclusion in section 5.1.2 that the fundamental of $SU(N)$ should be discarded when focusing on $S\mathrm{Diff}(T^2)$. The validity of the simple semiclassical estimate of its energy in eq. (127) suggests that in fact all states with resolvable vortices will be gapped. We recall that this class of states fully includes the representations that cannot be written as the tensor product of the same number of $\square$ and $\bar{\square}$. We do not have a mathematical proof of that, but, as illustrated in the appendix, all the explicit cases we studied numerically confirm this expectation. Again this matches our conclusion that the only states that should be kept consist of non-resolvable vortex-antivortex pairs. The smallest representation containing these is the adjoint. Somewhat expectedly, one finds that its spectrum is indeed not gapped, as we now show.

### 5.2.1 The spectrum of the adjoint: Vorton states

As we already discussed, the adjoints states $|\mathbf{n}\rangle$ are labelled by the fine lattice wave vectors while, as shown in eq. (108), the action of the charges is given by the structure constant themselves. It is then straightforward to see that the Hamiltonian of eq. (124) is diagonal in the $\{|\mathbf{n}\rangle\}$ basis

$$\langle\mathbf{m}|H|\mathbf{n}\rangle = E_\mathbf{n}\delta_{\mathbf{m},\mathbf{n}}, \tag{128}$$

with eigenvalues given by

$$E_\mathbf{m} = \frac{\Lambda^4}{8\pi^4\rho}\sum_{\mathbf{n}\neq\mathbf{0}}F(\mathbf{n})\frac{\sin^2\left(\frac{\pi}{N}(\mathbf{m}\wedge\mathbf{n})\right)}{\mathbf{n}^2}. \tag{129}$$

In contrast to the case of the fundamental representation, the spectrum now substantially depends on the choice of $F(\mathbf{n})$.

Let us consider first a sharp cut-off function satisfying eq. (122): $F(\mathbf{n}) = 1$ for $|n^I| < \frac{1}{2}(\sqrt{N}-1)$ and $F(\mathbf{n}) = 0$ for $|n^I| > \frac{1}{2}(\sqrt{N}-1)$. We drop for ease of computation the request $F(\mathbf{n}) = F(|\mathbf{n}|)$. The behaviour of the sum in eq. (129) depends on whether $|\mathbf{m}|$ is below or above $\sqrt{N}$ (i.e. whether $|\mathbf{p}|$ is below or above the cut-off scale $\Lambda$). For $|\mathbf{m}| \ll \sqrt{N}$ the sine has an argument smaller than $O(1)$ and can be expanded in powers of $\frac{m^I}{\sqrt{N}} = \frac{p^I}{\Lambda}$. In this regime the dispersion relation then reads

$$E_{\mathbf{m}} \simeq \frac{\Lambda^4}{8\pi^4\rho} \sum_{\mathbf{n}\neq 0}^{|n^I|\leq\frac{1}{2}(\sqrt{N}-1)} \frac{\pi^2}{N^2} \frac{(\mathbf{m}\wedge\mathbf{n})^2}{\mathbf{n}^2} = \frac{\Lambda^2}{16\pi^2\rho}\left(\frac{\mathbf{m}^2}{r^2}\right) = \frac{p^2}{4\rho a_1^2}, \tag{130}$$

corresponding to the non-relativistic kinetic energy of a particle with mass $M = 2\rho a_1^2$. That is twice the mass of a fluid element. Notice however that, given the sum in eq. (130) diverges quadratically, the precise result strongly depends on the choice of regulator.

To evaluate instead the energy for $|\mathbf{m}| \gg \sqrt{N}$, we can choose $m^I = (0, m)$ with $m \gg \sqrt{N}$. At leading order in $1/N$ and $1/m$ we can then write

$$E_{(0,m)} = \frac{\Lambda^4}{8\pi^4\rho} \sum_{\mathbf{n}\neq 0}^{|n^I|\leq\frac{1}{2}(\sqrt{N}-1)} \frac{\sin^2\left(\frac{\pi}{N}m n^1\right)}{\mathbf{n}^2} \simeq \frac{\Lambda^4}{2\pi^4\rho} \sum_{n^1,n^2=1}^{\frac{1}{2}\sqrt{N}} \frac{\sin^2\left(\frac{\pi}{N}m n^1\right)}{\mathbf{n}^2}$$

$$\simeq \frac{\Lambda^4}{4\pi^3\rho} \sum_{n^1=1}^{\frac{1}{2}\sqrt{N}} \frac{\sin^2\left(\frac{\pi}{N}m n^1\right)}{n^1} \simeq \frac{\Lambda^4}{8\pi^3\rho} \sum_{n^1\sim\frac{N}{m}}^{\frac{1}{2}\sqrt{N}} \frac{1}{n^1} \simeq \frac{\Lambda^4}{8\pi^3\rho}\ln\left(\frac{m}{\sqrt{N}}\right). \tag{131}$$

Choosing an arbitrary orientation of the wave vector $\mathbf{m}$ one can also check that the result is rotationally invariant up to corrections suppressed by inverse powers of $N$. In the continuum/infinite volume limit, $N \to \infty$ with $a_1 = 2\pi/\Lambda$ fixed, the energy of the states in the adjoint representation is then given by

$$E_{\text{adj}}(p) \simeq \begin{cases} \frac{p^2}{4\rho a_1^2}, & \text{for } p^2 \ll \Lambda^2, \\ \frac{\pi}{\rho a_1^4}\ln\left(\frac{p^2}{\Lambda^2}\right), & \text{for } p^2 \gg \Lambda^2. \end{cases} \tag{132}$$

A few comments are here in order. Notice, first of all, that the $N \to \infty$ limit with $a_1$ fixed is well defined. The dependence on both the fundamental lattice length $a$ and torus radius $r$ drop out. However, in addition to $\rho$, which fully defined the classical theory, a new parameter, $a_1 = 2\pi/\Lambda$, appears quantum mechanically. Secondly the adjoint states are gapless, as intuitively befits the universal long distance quanta of hydrodynamic vorticose motion. Indeed, the states encompassing the gapless region, that is $p < \Lambda$ and $E \propto p^2$, are precisely the vortons we previously identified. Somewhat expectedly the states that approximate a representation of $S\text{Diff}(T^2)$ within $SU(N)$ are the ungapped ones. The existence of vortons and their energy spectrum is the main result of our study.

The long wavelength behaviour $E \propto p^2$ gives way to a logarithmic behaviour $E \propto \log p$ at momenta of the order of the cut-off $\Lambda$. The velocity of the quanta at this change of regime is of order

$$|\frac{\partial E}{\partial p}| \sim \frac{\Lambda}{\rho a_1^2} = \frac{2\pi}{\rho a_1^3} \equiv c_v. \tag{133}$$

It is instructive to compare this result to the dispersion relation of longitudinal phonons. By zooming on incompressible flows, we have excluded the phonons from the very beginning. However, in any fluid they do exist and have linear dispersion relation: $E_{\text{ph}}(p) = c_s p$. For

small enough momenta, we then always have $E_{\text{ph}}(p) \gg E_{\text{adj}}(p)$, which justifies the decoupling of the longitudinal modes. The energies, however, become comparable at momenta of order $\Lambda_{\text{ph}} = \rho a_1^2 c_s = \Lambda \frac{c_s}{c_v}$, which provides another limit of applicability of our model. The simplest option, $c_s \sim c_v$, then identifies $\Lambda$ with the very scale at which fluid compression cannot be neglected.

Consider now different choices for $F$. The quadratic and logarithmic divergence of the relevant sums in respectively eqs. (130) and (131) are easily seen to determine what happens for different choices satisfying eq. (122). For $p \ll \Lambda$ the coefficient controlling $E \propto p^2$ changes by $O(1)$, while for $p \gg \Lambda$ only the finite part of the leading logarithmic contribution is modified.

More dramatic changes occur for regulators not respecting eq. (122), even though these choices are not physically sensible. For instance, for $F(\mathbf{n}) \equiv 1$, $\forall \mathbf{n}$, the energy behaves logarithmically at all momenta: $E_{\text{adj}}(p) \simeq \frac{\Lambda^4}{8\pi^3\rho} \ln(p\,r)$. Moreover the gap is finite: the energy of the state with the softest possible momentum is $E_{\text{adj}}(p = \frac{1}{r}) \approx 7\frac{\Lambda^4}{(2\pi)^4\rho}$. One can also check that any function $F(\mathbf{n})$ that shuts off between $|\mathbf{n}| \sim \sqrt{N}$ and $|\mathbf{n}| \sim N$ leads to a spectrum similar to eq. (132), but where the transition from quadratic to logarithmic behaviour happens at a scale lower than $\Lambda$. In particular, pushing the sharp cut-off to $C\sqrt{N}$ instead of $\sqrt{N}$, with $C > 1$, we have that the quadratic dispersion relation is only valid for states in the adjoint with $p \ll \Lambda^2 C^{-2}$. In that sense, the choice of $C = 1$ maximises the number of charges whose equation of motion eq. (121) matches the Euler equation, while extending the transition scale between the quadratic and logarithmic behaviour.

The main message is that, for the physically sensible regulators eq. (122), the vorton spectrum, while UV physics dependent, is robustly controlled by a quadratic dispersion relation, as shown in eq. (132). For definiteness in what follows we will stick to the choice $F(\mathbf{n}) = 1$ for $|n^I| < \frac{1}{2}(\sqrt{N}-1)$ and $F(\mathbf{n}) = 0$ for $|n^I| > \frac{1}{2}(\sqrt{N}-1)$.

### 5.2.2 Multi-vorton states

With the results obtained so far, we can now show that tensor products of adjoints are also ungapped. Again this matches what we argued in subsection 5.1.2, that only the low momentum portion of these representations properly represents $S$Diff in the $N \to \infty$ limit.

To study the spectrum let us consider a tensor product of a number $M$ of adjoints $|\mathbf{n}_1\rangle \otimes |\mathbf{n}_2\rangle \otimes \cdots \otimes |\mathbf{n}_M\rangle \equiv \bigotimes_j^M |\mathbf{n}_j\rangle$. The vorticity operator of the tensor product space is $\Omega_{tot} = \oplus_j^M \Omega_j$ where

$$\Omega_1(\boldsymbol{x}) = \Omega(\boldsymbol{x}) \otimes \mathbb{1} \otimes \cdots \otimes \mathbb{1}, \quad \Omega_2(\boldsymbol{x}) = \mathbb{1} \otimes \Omega(\boldsymbol{x}) \otimes \cdots \otimes \mathbb{1}, \quad \text{etc.} \tag{134}$$

To study the energy on these states we can consider a wave packet

$$|\{\psi\}\rangle = \sum_{\mathbf{n}_1 \dots \mathbf{n}_M} \left( \bigotimes_J^M \hat{\psi}_j(\mathbf{n}_j) |\mathbf{n}_j\rangle \right), \tag{135}$$

and estimate $\langle\{\psi\}|H|\{\psi\}\rangle$. As we are aiming at the low energy limit, we will focus on wave packets supported in the vorton region $|\mathbf{p}| = |\mathbf{n}/r| \ll \Lambda$. The Hamiltonian consists of a sum over pairs $jk$ involving the bilinears products $\Omega_j \Omega_k$. One can easily see that the diagonal terms $\Omega_1 \Omega_1, \dots \Omega_M \Omega_M$ give each the energy expectation $E_j$ on the corresponding adjoint state

$$E_j = \sum_n |\hat{\psi}_j(\mathbf{n})|^2 E_{\mathbf{n}}, \tag{136}$$

with $E_{\mathbf{n}}$ given by eq. (130). The sum of these contributions $E_1 + \cdots + E_M$ thus corresponds to the energy of $M$ non-interacting vortons: as vortons are ungapped, this energy can be made

arbitrarily small (in the infinite volume limit) by picking individual vorton wave functions $\psi_j$ supported at softer and softer momenta.

Consider now instead the cross terms involving $\Omega_i \Omega_j$, with $i \neq j$, which can be interpreted as giving the interaction energy between vortons. One finds that the contribution of these terms is controlled by the products of vorticity expectation values in eq. (117) of each separate vorton state. In terms of the position space wave packets and already taking the continuum limit we have

$$\left\langle \psi_j \middle| \Omega(\boldsymbol{x}) \middle| \psi_j \right\rangle = \frac{i \, \partial \, \psi_j \wedge \partial \, \psi_j^*}{\rho} \equiv \partial_I J_j^I \,, \tag{137}$$

with

$$J_j^I = \frac{1}{2\rho} \left( -i \psi_j^* \tilde{\partial}^I \psi_j + i \tilde{\partial}^I \psi_j^* \psi_j \right) . \tag{138}$$

Unlike for the diagonal terms, one finds that the $i \neq j$ contributions are not quadratically UV divergent. For wave packets supported at momenta $\ll \Lambda$, we can then remove the regulator and use the long distance Hamiltonian of eq. (69). To estimate the energy we can also directly work at infinite volume. The interaction energy is then given by

$$\frac{\rho}{2} \sum_{i \neq j} \int \Omega_i \frac{1}{-\nabla^2} \Omega_j = \frac{\rho}{2} \sum_{i \neq j} \int d^2 \boldsymbol{x} \, d^2 \boldsymbol{y} J_i^K(\boldsymbol{x}) G_{KL}(\boldsymbol{x} - \boldsymbol{y}) J_j^L(\boldsymbol{y}), \tag{139}$$

$$G_{KL}(x - y) = \frac{2}{\pi} \left[ \delta^{KL} - \frac{2(x - y)^K (x - y)^L}{|\boldsymbol{x} - \boldsymbol{y}|^2} \right] \frac{1}{|\boldsymbol{x} - \boldsymbol{y}|^2} \,, \tag{140}$$

where in the last step we used the explicit expression of the 2D propagator $-1/\nabla^2$, then eq. (137) and finally we integrated by parts. The final expression corresponds to the electrostatic interaction energy among a set of localized charge dipoles with polarization densities $-J_i^I$. As $G_{KL}$ decreases like $|\boldsymbol{x} - \boldsymbol{y}|^{-2}$, the interaction energy can be made arbitrarily small by making the wave packets arbitrarily separated from one another. In the limit of infinite reciprocal separation the only remaining contribution is the diagonal one $E_1 + \cdots + E_M$, corresponding to $M$ non-interacting vortons.

The obvious corollary of this discussion is that multi-vorton states are also ungapped.

## 6 Vorton interactions

Working directly at infinite volume, we have just shown that vortons behave like particles with interactions that vanish at infinite mutual separation. We can then think of a scattering experiment, where, for instance, two originally infinitely separated vorton waved packets are time evolved to within a finite distance where they interact. In this section we will compute the scattering amplitude of such two-vorton process. We shall arrive at that in three steps. The first is a detailed study of the two vorton state at finite $N$. This step is in fact not strictly necessary, given we are in any case interested in scattering at infinite volume. However, we believe it naturally completes the matrix model description of the fluid. The second step is the derivation of a vorton field theory, which describes both the spectrum and the interactions of vortons in the $N \to \infty$ limit. This result summarizes our understanding of the *universal* long distance dynamics of a perfect quantum fluid. As an application of the vorton field theory we shall then compute the scattering amplitude.

### 6.1 Two vorton states

We want to consider representations that belong in the tensor product of two adjoints. Using Young Tableaux such tensor product is seen to decompose into the direct sum of seven

irreducible representations:

$$
\text{(diagram)} \otimes \text{(diagram)} = \left( \text{(diagram)} \oplus \text{(diagram)} \oplus \text{(diagram)} \oplus \bullet \right)_S \oplus \left( \text{(diagram)} \oplus \text{(diagram)} \oplus \text{(diagram)} \right)_A , \quad (141)
$$

where the longest columns have $N-1$ boxes and the bullet stands for the trivial representation. The first four representations are symmetric under exchange of the two adjoints and the latter three are antisymmetric. This decomposition is easy to understand by representing adjoints states as traceless matrices $f^{\,\alpha}_{\,\beta}$, corresponding to the traceless product of a vortex–anti-vortex pair, see eq. (110). The first two terms are respectively symmetric and antisymmetric under exchange of any pair of constituent vortices or constituent anti-vortices. They have dimensions of order $\frac{1}{4}N^4$ each. They are self-conjugated and thus invariant under interchanging the roles of vortices and anti-vortices. Analogously, the first two terms among antisymmetric representations have opposite symmetry properties under exchange of vortices and exchange of anti-vortices. These two representations have also each dimensions of order $\frac{1}{4}N^4$ and are conjugated to one another. Apart from these four large representations, one finds also two adjoint representations and the singlet.

What we have just discussed is purely group theory. We must now view this result from the standpoint of physics. In particular eq. (141) is at odds with the interpretation of the adjoint states in the left hand side as particles in a Fock space: in the tensor product of two particles we would find both single particle states, the adjoints, and the vacuum, the singlet. Notice that, given the Hamiltonian only depends on the $SU(N)$ (or $S\mathrm{Diff}(T^2)$) generators, the adjoints and the singlet on the right hand side are dynamically indistinguishable from the same representations we already studied.[26] Indeed in the particular case of hypothesis **A** of section 3.2, the Peter-Weil theorem implies that each representation $(r, r)$ of $SU(N)_L \times SU(N)_R$ features only once, in such a way that the adjoints and singlets on the right hand side of eq. (141) correspond to states we already considered.

The above obstruction to a particle interpretation has however a rather simple solution. If we fold the left hand side with vorton wave packets, i.e. localized at momenta $p \ll \Lambda$, and then take the $N \to \infty$ limit, the overlap with the adjoints and the singlet on the right hand side goes to zero. Furthermore, up to $O(p^2/\Lambda^2)$ effects, the overlap with the remaining four large representations, reduces in the same limit to two combinations, the symmetric and antisymmetric one: $|\mathbf{m}, \mathbf{n}\rangle_{S/A} = (|\mathbf{m}\rangle \otimes |\mathbf{n}\rangle \pm |\mathbf{n}\rangle \otimes |\mathbf{m}\rangle)/\sqrt{2}$. These corresponds to the two-particle subsector of a Fock space of either bosonic or fermionic identical quanta. Let us see how this works in detail.

The adjoints in the right hand side of eq. (141) arise from tracing one constituent vortex and one anti-vortex from different adjoints, while the vacuum corresponds to tracing both. The states belonging to the two adjoints are then explicitly written as

$$
|\ell\rangle_{S/A} \propto \sideset{}{'}\sum_{\mathbf{m},\mathbf{n}} (J_\ell)^{\alpha}_{\beta} \left( J^*_{\mathbf{m}} \right)^{\gamma}_{\alpha} \left( J^*_{\mathbf{n}} \right)^{\beta}_{\gamma} \left( |\mathbf{m}\rangle \otimes |\mathbf{n}\rangle \pm |\mathbf{n}\rangle \otimes |\mathbf{m}\rangle \right). \quad (142)
$$

Tracing over the three $J$-matrices and normalizing the states we can write[27]

$$
|\ell\rangle_S = \frac{\sqrt{2}}{N} \sideset{}{'}\sum_{\mathbf{m}} \cos\left( \frac{\pi}{2N} \mathbf{m} \wedge \ell \right) \left| \frac{1}{2}(\ell + \mathbf{m}) \right\rangle \otimes \left| \frac{1}{2}(\ell - \mathbf{m}) \right\rangle, \quad (143)
$$

$$
|\ell\rangle_A = \frac{\sqrt{2}}{N} \sideset{}{'}\sum_{\mathbf{m}} i \sin\left( \frac{\pi}{2N} \mathbf{m} \wedge \ell \right) \left| \frac{1}{2}(\ell + \mathbf{m}) \right\rangle \otimes \left| \frac{1}{2}(\ell - \mathbf{m}) \right\rangle. \quad (144)
$$

---

[26] It is interesting to contrast this situation with, for instance, the case of isospin in QCD: unlike $SU(N)$ here, isospin merely constrains the dynamics without fully determining it.

[27] In the normalization factor of $|\ell\rangle_S$ we have approximated $\sqrt{N^2 - 2} \to N$.

In the above equations the sums over $\mathbf{m}$ actually extend over the double fundamental domain, that is $-N + 1 \leq m_I \leq N - 1$, and only the $\mathbf{m}$'s for which the arguments of the kets are integer vectors should be retained. Moreover, the twisted periodicity of eq. (77) should be applied to the kets, whenever their argument is outside the fundamental domain.

The vacuum state corresponds to the symmetric zero-momentum state

$$|\bullet\rangle = |\boldsymbol{\ell} = 0\rangle_S / \sqrt{2} = \frac{1}{N} \sum_{\mathbf{m}}{}' |\mathbf{m}\rangle \otimes |-\mathbf{m}\rangle = \frac{a^2}{N} \sum_{\boldsymbol{x}} |\boldsymbol{x}\rangle \otimes |\boldsymbol{x}\rangle , \tag{145}$$

where in the last step we used the position eigenstates defined in eq. (111). The above three equations all define largely entangled states of two vortons, the majority of which have momenta above the fluid cut-off $\Lambda$. We thus expect their overlap with vorton wave packets to be suppressed. Consider indeed a two vorton wave-packet $|f_1(\boldsymbol{x})\rangle \otimes |f_2(\boldsymbol{x})\rangle$ where $f_1$ and $f_2$ are localised on a distance $L$ larger than the fluid cut-off length $a_1$. The overlap with the "small" representations subspace for such a state is approximately given by

$$\sum_{\boldsymbol{\ell},S,A} \left| (\langle f_1 | \otimes \langle f_2 |) |\boldsymbol{\ell}\rangle_{S/A} \right|^2 \simeq \sum_{\boldsymbol{x}} |f_1(\boldsymbol{x})|^2 |f_2(\boldsymbol{x})|^2 \sim \frac{a^2}{L^2} < \frac{a^2}{a_1^2} = \frac{1}{N} , \tag{146}$$

and vanishes for $N \to \infty$. A similar conclusion holds for the overlap with $|\bullet\rangle$. In the continuum limit, we can thus neglect the overlap of $|\boldsymbol{\ell}\rangle_{S,A}$ and $|\bullet\rangle$ with vorton wave packets.

Consider now the four "big" representation in eq. (141). We can also explicitly write them using projectors. For example, the projector on the symmetric-symmetric states is

$$P_{SS} \propto \sum_{\substack{\mathbf{m}_1,\mathbf{m}_2, \\ \mathbf{n}_1,\mathbf{n}_2}}{}' |\mathbf{m}_1,\mathbf{m}_2\rangle \langle \mathbf{n}_1,\mathbf{n}_2 | \left\{ \left( \left(J_{\mathbf{m}_1}\right)^{\alpha_1}_{\beta_1} \left(J_{\mathbf{m}_2}\right)^{\alpha_2}_{\beta_2} \left(J^*_{\mathbf{n}_1}\right)^{\beta_1}_{\alpha_1} \left(J^*_{\mathbf{n}_2}\right)^{\beta_2}_{\alpha_2} + (\alpha_1 \leftrightarrow \alpha_2) \right) \right.$$
$$\left. + (\beta_1 \leftrightarrow \beta_2) \right\}, \tag{147}$$

where we neglected the subtraction of trace terms that correspond to the singlet and the adjoints, which we already established to be negligible. The other three projectors are obtained by exchanging symmetrisations with anti-symmetrisations accordingly. Two terms in eq. (147) contain a double trace of products of $J$-matrices. They give two momentum conservation Kronecker symbols and correspond to the identity operator on the symmetric or anti-symmetric subspaces, respectively

$$I_{S/A} = \frac{1}{4} \sum_{\mathbf{m}_1,\mathbf{m}_2}{}' \left( |\mathbf{m}_1,\mathbf{m}_2\rangle \langle \mathbf{m}_1,\mathbf{m}_2 | \pm |\mathbf{m}_1,\mathbf{m}_2\rangle \langle \mathbf{m}_2,\mathbf{m}_1 | \right). \tag{148}$$

The other two terms involve the single trace of a product of four $J$-matrices, which produces only conservation of the total momentum. These terms are thus off-diagonal in the individual vorton momentum labels. In respectively the symmetric and anti-symmetric subspaces they correspond to the following operators (the same comments we made below eq. (144) apply to these summations)

$$\Delta_S = \frac{1}{2N} \sum_{\boldsymbol{\ell},\mathbf{m},\mathbf{n}}{}' \cos\left( \frac{\pi}{2N} \mathbf{m} \wedge \mathbf{n} \right) \left| \frac{1}{2}(\boldsymbol{\ell} + \mathbf{n}), \frac{1}{2}(\boldsymbol{\ell} - \mathbf{n}) \right\rangle \left\langle \frac{1}{2}(\boldsymbol{\ell} + \mathbf{m}), \frac{1}{2}(\boldsymbol{\ell} - \mathbf{m}) \right|, \tag{149}$$

$$\Delta_A = \frac{1}{2N} \sum_{\boldsymbol{\ell},\mathbf{m},\mathbf{n}}{}' \sin\left( \frac{\pi}{2N} \mathbf{m} \wedge \mathbf{n} \right) \left| \frac{1}{2}(\boldsymbol{\ell} + \mathbf{n}), \frac{1}{2}(\boldsymbol{\ell} - \mathbf{n}) \right\rangle \left\langle \frac{1}{2}(\boldsymbol{\ell} + \mathbf{m}), \frac{1}{2}(\boldsymbol{\ell} - \mathbf{m}) \right|. \tag{150}$$

Using these elements, the full projectors can be written as

$$P_{SS} = I_S + \Delta_S\,, \qquad P_{AA} = I_S - \Delta_S\,, \tag{151}$$

$$P_{AS} = I_A + \Delta_A\,, \qquad P_{SA} = I_A - \Delta_A\,. \tag{152}$$

Very much as we did previously with the "small" representations, one can study the matrix elements of $\Delta_S$ and $\Delta_A$ on vorton wave packets localised on a distance $L$ larger than the fluid cut-off length $a_1$. In the same notation used previously we find

$$|\langle f_1| \otimes \langle f_2| \Delta_{S,A} |g_1\rangle \otimes |g_2\rangle| \lesssim \frac{a_1^2}{L^2}\,. \tag{153}$$

The overlap is not as suppressed as in eq. (146), but it still becomes negligible for wave packets that are deeply in the vorton region $L \gg a_1$. When projected on vorton wave packets states the four "large" representations reduce in reality to just two representations, corresponding to the symmetric and antisymmetric product of two vortons $|\mathbf{m},\mathbf{n}\rangle_{S/A} = (|\mathbf{m}\rangle \otimes |\mathbf{n}\rangle \pm |\mathbf{n}\rangle \otimes |\mathbf{m}\rangle)/\sqrt{2}$. As already anticipated, when considering vorton wave packets, the content of the seven irreps in eq. (141) reduces to $|\mathbf{m},\mathbf{n}\rangle_S$ and $|\mathbf{m},\mathbf{n}\rangle_A$, corresponding to the two particle Fock-subspace for respectively a boson and a fermion field.

One can go on and investigate the same questions for representations that can be written as tensor products of more than two adjoints. We have not done the exercise, but for a product of $M$ adjoints $|\mathbf{n}_1\rangle \otimes \cdots \otimes |\mathbf{n}_M\rangle$ we expect the irreducible subspaces to correspond to Young tableaux with $M$ boxes, modulo the subtraction of traces. The simplest options, consisting of a row or a column of boxes, correspond respectively to identical bosons or identical fermions. However, the more complex options correspond to mixed statistics.

These results motivate the investigation of a field theoretic realization of vortons.

## 6.2 Vorton field theory

The results of the last two subsections indicate that it should be possible to bypass the truncation of $S\mathrm{Diff}(T^2)$ to $SU(N)$ and focus on a theory where the vortons are the sole degrees of freedom. In order to achieve that, we need to go back to section 3 and seek a different route to quantization. All we need is to find an explicit realization of the algebra of $R$-charges, use it to write the Hamiltonian of eq. (69) and then study the dynamics. The first step is easily taken, as any Lie algebra can be explicitly realized by either bosonic or fermionic ladder operators through the Jordan-Schwinger map [54, 55]. In practice this works as follows. Given any explicit representation $(T^A)^\beta_\alpha$ of the generators $Q^A$, one considers either bosonic or fermionic ladder operators transforming in the same representation, that is either $\Phi^\alpha$ or $\Psi^\alpha$ with commutation relations

$$[\Phi^\alpha, \Phi^\dagger_\beta] = \delta^\alpha_\beta\,, \qquad \{\Psi^\alpha, \Psi^\dagger_\beta\} = \delta^\alpha_\beta\,. \tag{154}$$

The operators $Q^A_b \equiv \Phi^\dagger_\beta \Phi^\alpha (T^A)^\beta_\alpha = \Phi^\dagger T^A \Phi$ and $Q^A_f \equiv \Psi^\dagger_\beta \Psi^\alpha (T^A)^\beta_\alpha = \Psi^\dagger T^A \Psi$ are then both easily seen to represent the algebra.

In our case, a basis of the Lie algebra is given by the local charges $R(\boldsymbol{x})$, whose commutator is the Fourier transform of eq. (66)

$$[R(\boldsymbol{x}), R(\mathbf{y})] = -i \int d^2z\, R(\mathbf{z}) \partial_z \delta^2(\mathbf{z} - \boldsymbol{x}) \wedge \partial_z \delta^2(\mathbf{z} - \mathbf{y})\,. \tag{155}$$

We can then introduce ladder operators, either bosonic or fermionic, transforming in the adjoint of $S\mathrm{Diff}$, that is position space scalar fields $\Phi(\boldsymbol{x})$ or $\Psi(\boldsymbol{x})$ satisfying

$$[\Phi(\boldsymbol{x}), \Phi^\dagger(\mathbf{y})] = \delta^2(\boldsymbol{x} - \mathbf{y})\,, \qquad \{\Psi(\boldsymbol{x}), \Psi^\dagger(\mathbf{y})\} = \delta^2(\boldsymbol{x} - \mathbf{y})\,. \tag{156}$$

Applying the above construction, one then finds that the $R$-algebra is equally well represented as

$$R_b(\boldsymbol{x}) = i\partial\Phi^\dagger \wedge \partial\Phi\,, \qquad \text{or} \qquad R_f(\boldsymbol{x}) = i\partial\Psi^\dagger \wedge \partial\Psi\,. \tag{157}$$

Of course also the sum $R_b + R_f$ offers a representation. In the boson (fermion) case, the Hilbert space is a Fock space constructed from a vacuum state $|0\rangle$, satisfying $\Phi|0\rangle = 0$ ($\Psi|0\rangle = 0$) by repeated action with $\Phi^\dagger$ ($\Psi^\dagger$). As we will now discuss, the particle excitations of either Fock space offer specific incarnations of the vortons.

Focusing for definiteness on the bosonic case, the Hamiltonian is obtained by replacing $\Omega(\boldsymbol{x}) = -R_b(\boldsymbol{x})/\rho$ in the Hamiltonian of eq. (69). Of course we should also take into account the need for a UV regulation of the intermediate $1/\nabla^2$, as evidenced by the study of the $SU(N)$ truncation. Working directly in the infinity volume limit (thus dropping the velocity zero mode $\bar{v} \to 0$) we can write

$$H = \frac{1}{2\rho}\int d^2\boldsymbol{x}\, R_b \frac{F(-\nabla^2/\Lambda^2)}{-\nabla^2} R_b \tag{158}$$
$$= \frac{1}{2\rho}\int d^2\boldsymbol{x}\, d^2\mathbf{y}[i\partial\Phi^\dagger \wedge \partial\Phi](\boldsymbol{x}) G^\Lambda(\boldsymbol{x}-\mathbf{y})[i\partial\Phi^\dagger \wedge \partial\Phi](\mathbf{y})\,,$$

where, as previously, $F(\tau)$ is a UV regulator function satisfying $F \to 0$ for $\tau \to \infty$ and $F \to 1$ for $\tau \lesssim 1$. The UV regulated Green's function $G^\Lambda(\boldsymbol{x}-\mathbf{y})$ is still logarithmically IR divergent. However, as already discussed in section 5.2.2, $R_b$ is a total derivative, and the divergent piece drops out. More precisely we can write

$$R_b(\boldsymbol{x}) = \partial_I J^I\,, \qquad J^I = \frac{i}{2}\left(\Phi^\dagger \tilde{\partial}^I \Phi - \tilde{\partial}^I \Phi^\dagger \Phi\right)\,, \tag{159}$$

so that we can express the Hamiltonian as

$$H = \frac{1}{2\rho}\int d^2\boldsymbol{x}\, d^2\mathbf{y}\, J^K(\boldsymbol{x}) G^\Lambda_{KL}(\boldsymbol{x}-\mathbf{y}) J^L(\mathbf{y})\,, \tag{160}$$

where the long distance behaviour of $G^\Lambda_{KL}$ is given by eq. (140), while at coinciding points its quadratic divergence is regulated to

$$G^\Lambda_{KL}(0) = \left[\frac{\Lambda^2}{8\pi}\int dx F(x)\right]\delta_{KL} \equiv c\frac{\Lambda^2}{8\pi}\delta_{KL}\,. \tag{161}$$

To study the dynamics it is further convenient to write the Hamiltonian in normal ordered form. A straightforward computation then gives

$$H = \frac{1}{2\rho}\left\{c\frac{\Lambda^2}{8\pi}\int d^2\boldsymbol{x}\,\partial\Phi^\dagger\partial\Phi + \int d^2\boldsymbol{x}\, d^2\mathbf{y} : J^K(\boldsymbol{x}) G^\Lambda_{KL}(\boldsymbol{x}-\mathbf{y}) J^L(\mathbf{y}) :\right\}\,. \tag{162}$$

This results makes the dynamics manifest. The degrees of freedom are bosonic vortons with a dispersion relation set by the term quadratic in the field: $E_\mathbf{p} = (c\Lambda^2/16\pi\rho)\mathbf{p}^2$, which nicely matches the result in section 5.2.1. The second term in the Hamiltonian, vanishes on single vorton states, and represents a dipolar interaction among vorton pairs. Also for this term one can check that the matrix elements perfectly match the $SU(N)$ model matrix elements among symmetric bi-adjoints $|\mathbf{m},\mathbf{n}\rangle_S = (|\mathbf{m}\rangle \otimes |\mathbf{n}\rangle + |\mathbf{n}\rangle \otimes |\mathbf{m}\rangle)/\sqrt{2}$ at low momentum ($p \ll \Lambda$). This is not surprising given the computation is fully controlled by the algebra of the $R$-charges. Furthermore, as one can immediately appreciate, eq. (162) makes a specific prediction for the structure of the vorton scattering amplitude. In fact, at Born level the regulator can be dropped,

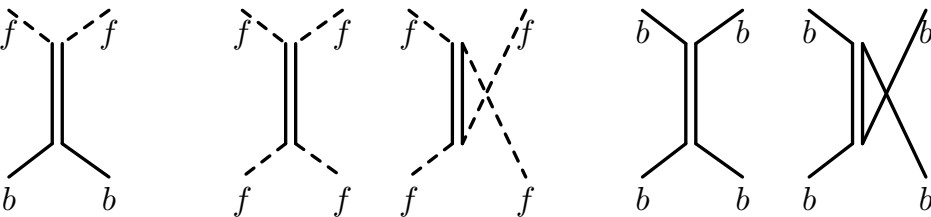

Figure 6: Born level Feynman diagrams relevant for the vorton scattering.

and the amplitude is fully determined by one parameter, $\rho$, with a specific dependence on the external momenta. We will present this computation in the next subsection.

The above discussion goes through basically unchanged for the case of fermionic vortons: the result amounts to replacing $\Phi \to \Psi$ in eqs. (159) and (162), while the matrix elements now match those for the anti-symmetric bi-adjoint $|\mathbf{m}, \mathbf{n}\rangle_A = (|\mathbf{m}\rangle \otimes |\mathbf{n}\rangle - |\mathbf{n}\rangle \otimes |\mathbf{m}\rangle)/\sqrt{2}$ in the $SU(N)$ model. Similarly in a realization with both a boson and a fermion, the Hamiltonian would consist, at the quadratic level, of the sum of the bosonic and fermionic kinetic term, while the current appearing in the interaction would be the sum of the bosonic and fermionic currents. That would make specific predictions for the scattering amplitudes $bb \to bb$, $ff \to ff$ and $bf \to bf$. Notice that individual vorton number is preserved, so that processes like $bb \to ff$ do not occur.

## 6.3 Vorton scattering

We are now ready to apply our results to compute the simplest instance of vorton $S$-matrix: $2 \to 2$ scattering. As stressed in the previous section, the matrix elements in the vorton field theory match those of the $SU(N)$ model in the low energy limit. To perform computations we can thus work directly in the field theory. Writing the $S$-matrix as $S = \mathbb{1} + i\mathbb{M}$, the reduced $2 \to 2$ scattering amplitude $\mathcal{M}$ is defined by

$$\langle \mathbf{k}_1, \mathbf{k}_2 | \mathbb{M} | \mathbf{q}_1, \mathbf{q}_2 \rangle = (2\pi)^3 \delta(E_{\mathbf{k}_1} + E_{\mathbf{k}_2} - E_{\mathbf{q}_1} - E_{\mathbf{q}_2}) \delta^2(\mathbf{k}_1 + \mathbf{k}_2 - \mathbf{q}_1 - \mathbf{q}_2) \mathcal{M}(\mathbf{k}_1, \mathbf{k}_2, \mathbf{q}_1, \mathbf{q}_2). \quad (163)$$

In the Born approximation, $\mathcal{M}$ is directly given by the matrix elements of the interaction Hamiltonian. The corresponding Feynman diagrams for the three cases of interest $bb$, $ff$ and $bf$ are shown in Fig. 6. As also evidenced by the figure, the basic building block is the $bf$ amplitude, which has the structure of a $t$-channel amplitude.

Defining

$$\mathcal{M}_t(\mathbf{k}_1, \mathbf{k}_2, \mathbf{q}_1, \mathbf{q}_2) = \frac{1}{\rho} \frac{(\mathbf{k}_1 \wedge \mathbf{q}_2)(\mathbf{k}_2 \wedge \mathbf{q}_1)}{(\mathbf{q}_1 - \mathbf{k}_1)^2}, \quad (164)$$

we then have

$$\mathcal{M}_{bf}(\mathbf{k}_1, \mathbf{k}_2, \mathbf{q}_1, \mathbf{q}_2) = 2\mathcal{M}_t(\mathbf{k}_1, \mathbf{k}_2, \mathbf{q}_1, \mathbf{q}_2), \quad (165)$$
$$\mathcal{M}_{bb}(\mathbf{k}_1, \mathbf{k}_2, \mathbf{q}_1, \mathbf{q}_2) = \mathcal{M}_t(\mathbf{k}_1, \mathbf{k}_2, \mathbf{q}_1, \mathbf{q}_2) + \mathcal{M}_t(\mathbf{k}_2, \mathbf{k}_1, \mathbf{q}_1, \mathbf{q}_2) \equiv \mathcal{M}_t + \mathcal{M}_u,$$
$$\mathcal{M}_{ff}(\mathbf{k}_1, \mathbf{k}_2, \mathbf{q}_1, \mathbf{q}_2) = \mathcal{M}_t(\mathbf{k}_1, \mathbf{k}_2, \mathbf{q}_1, \mathbf{q}_2) - \mathcal{M}_t(\mathbf{k}_2, \mathbf{k}_1, \mathbf{q}_1, \mathbf{q}_2) \equiv \mathcal{M}_t - \mathcal{M}_u,$$

where, in an obvious notation, we defined a $u$-channel amplitude $\mathcal{M}_u$ obtained by exchanging $\mathbf{k}_1 \leftrightarrow \mathbf{k}_2$ in $\mathcal{M}_t$. Notice that all these amplitudes vanish when any of the external momenta vanish. Like for the vanishing of the vorton energy at zero momentum, this can be intuitively understood by the fact that, at zero momentum, adjoint states becomes locally indistinguishable from the singlet. While eq. (164) makes this property manifest, it is also useful

to implement energy and momentum conservation and write it in terms of unconstrained kinematic variables. In particular, momentum conservation allows to write $\mathbf{q}_{1/2} = \frac{1}{2}(\mathbf{P} \pm \mathbf{Q})$ and $\mathbf{k}_{1/2} = \frac{1}{2}(\mathbf{P} \pm \mathbf{K})$, while energy conservation corresponds to $\mathbf{K}^2 = \mathbf{Q}^2$. Substituting in eq. (165) and taking into account that crossing now corresponds to $\mathbf{K} \to -\mathbf{K}$, we can write (no need to show $\mathcal{M}_{bf} = (\mathcal{M}_{bb} + \mathcal{M}_{ff})/2$, whose expression is too long)

$$\mathcal{M}_{bb} = \frac{1}{4\rho} \left\{ \mathbf{Q}^2 - \mathbf{P}^2 \right\}, \tag{166}$$

$$\mathcal{M}_{ff} = \frac{1}{4\rho} \left\{ \frac{(\mathbf{Q}^2 - \mathbf{P}^2)(\mathbf{P} \cdot \mathbf{Q})(\mathbf{P} \cdot \mathbf{K}) + (\mathbf{Q}^2 + \mathbf{P}^2)(\mathbf{P} \wedge \mathbf{Q})(\mathbf{P} \wedge \mathbf{K})}{\mathbf{P}^2 \mathbf{Q}^2} \right\} \tag{167}$$

$$= \frac{1}{4\rho} \left\{ (\mathbf{Q}^2 - \mathbf{P}^2) \cos\alpha \cos\beta + (\mathbf{Q}^2 + \mathbf{P}^2) \sin\alpha \sin\beta \right\}. \tag{168}$$

Where $\alpha$ ($\beta$) is the angle between $\mathbf{P}$ and $\mathbf{Q}$ ($\mathbf{K}$). The dependence of the amplitudes on the total momentum $\mathbf{P}$ makes evident, as it was to be expected, the absence of Galilean invariance. The vorton dispersion relation $E_\mathbf{k} \sim \mathbf{k}^2$ only accidentally complies with Galilean symmetry. Notice also that, even though the interaction Hamiltonian is non-local, the bosonic amplitude $\mathcal{M}_{bb}$ is polynomial in the momenta. That is not the case for the fermionic one $\mathcal{M}_{ff}$, whose singularity at zero momentum perfectly reflects the non-locality of the Hamiltonian. There must of course exist a change of variables that makes the locality of the Born level bosonic $2 \to 2$ amplitude manifest. We have not investigated that, but it would be interesting to do so and see if all the bosonic tree level amplitudes are made local that way.

Another property of $\mathcal{M}_{bb}$ is its independence of the angular variables: in terms of partial waves for $\mathbf{Q}$ (and $\mathbf{K}$) it is a pure $s$-wave. This result follows from the quadratic dependence on the momenta and from Bose symmetry, which dictates invariance of $\mathcal{M}_{bb}$ under the independent sign flip of $\mathbf{Q}$ and $\mathbf{K}$. These properties constrain $\mathcal{M}_{bb}$ to be a linear function of $\mathbf{P}^2$ and $\mathbf{Q}^2$. In fact $\mathcal{M}_{bb}$ has the additional property of vanishing for $|\mathbf{P}| = |\mathbf{Q}|$, that is for incoming, and outgoing, orthogonal momenta: $\mathbf{q}_1 \cdot \mathbf{q}_2 = \mathbf{k}_1 \cdot \mathbf{k}_2 = 0$. A pair of bosonic vortons thus has the peculiarity of not diffusing when crossing at 90 degrees. The fermionic amplitude $\mathcal{M}_{ff}$ does not share this property, though it has other zeroes. In particular it vanishes for $(\alpha, \beta)$ equal to either $(\pi/2, 0)$ or $(0, \pi/2)$. In terms of momenta, this means that in the scattering of two vortons with the same energy, i.e. $|\mathbf{k}_1| = |\mathbf{k}_2|$, the final state momenta $\mathbf{q}_1$ and $\mathbf{q}_2$ cannot be parallel. Similarly for parallel initial state momenta, the two final state vortons cannot have the same energy. Of course these properties follow from the dipolar nature of the vorton-vorton interaction, see eq. (162), and from the relation between dipole and momentum, see eq. (159).

The Born level is a good approximation as long as the coupling is weak. A natural measure of coupling strength is provided by the scattering phases. These are readily read off the $S$-matrix on states labelled by energy, momentum and angular momentum, where the latter is just the the quantum number canonically conjugated to the angles $\alpha$, or $\beta$. One finds that, given the vorton mass $\mu = c(8\pi\rho/\Lambda^2)$, the scattering phases scale like $\mathcal{M}\mu \sim k^2/\Lambda^2$. This result also simply follows from dimensional analysis, given that $\mathcal{M}$ has dimension $[k^2/\rho] = 1/[mass]$ and given that the vorton mass is the only mass parameter we can use to normalize the amplitude so as to make it dimensionless. A quick study of the 1-loop corrections also shows that the loop expansion is controlled by $k^2/\Lambda^2$. We conclude that the coupling strength is $\sim k^2/\Lambda^2$ so that the system is strongly coupled at the cut-off, i.e. for $k \sim \Lambda$, but weakly coupled at all lower momenta. Notice that, even though the interaction Hamiltonian, when written in terms of the canonical field $\Phi$, purely depends on the classical parameter $\rho$, it should not come as a surprise that the interaction strength is instead controlled by the cut-off $\Lambda$. That is because the existence of asymptotic states and of an $S$-matrix is guaranteed precisely by the existence of the UV cut-off $\Lambda$. That makes it more intuitively clear that the strength of

the interaction is in the end controlled by $k^2/\Lambda^2$.

## 6.4 Effective field theory approach

The results of the last two sections compel us to try and investigate vorton QFT from the modern effective field theory (EFT) standpoint. In this section we will make a brief foray into that, focusing on the bosonic vorton QFT. The need for a deeper study will emerge. We leave that to future work [52].

The idea is to formulate vorton QFT in the path integral approach on the basis of symmetry principles and by working order by order in the derivative expansion. Notice however that the use of spacial locality as a constitutive principle will be limited, given vortons are endowed with long range dipolar interactions.

In the path integral approach, the first step is to write the classical Lagrangian corresponding to the Hamiltonian of eq. (158). Adding an auxiliary field $A(\boldsymbol{x},t)$ such Lagrangian can be written as

$$\mathcal{L} = i\Phi^\dagger \dot{\Phi} + \frac{1}{\rho}\left[i(\partial \Phi^\dagger \wedge \partial \Phi)A + \frac{1}{2}\partial A \frac{1}{F(-\partial^2/\Lambda^2)}\partial A\right], \tag{169}$$

where $F(\tau)$ is the same regulator function defined in eq. (158). By the above Lagrangian, $\Phi$ and $\Phi^\dagger$ are canonically conjugated variables, and moreover one can easily see that by eliminating the auxiliary field $A$ one gets back the Hamiltonian pf eq. (158). Notice that the Lagrangian is non-local only at distance scales $\sim 1/\Lambda$, and because of the non-trivial regulator function $F$. On the other hand, the exchange of $A$ produces a specific long-distance non-locality in $H$. As, by hypothesis, $F(\tau)$ is regular around $\tau = 0$, it's low momentum expansion defines a subset of the higher derivative corrections to the leading long-distance dynamics.

The above Lagrangian is invariant under two sets of continuous transformations. The first is given by

$$A(\boldsymbol{x},t) \to A(\boldsymbol{x},t) + \alpha(t), \tag{170}$$

for any $\alpha(t)$. While the invariance of the quadratic term is obvious, that of the term linear in $A$ is proven by noticing that $\partial \Phi^\dagger \wedge \partial \Phi$ is a total derivative and by integrating by parts. Notice that this symmetry forbids $\dot{A}^2$, thus making $A$ an auxiliary field. The second set of transformations consists of volume preserving mappings in $(\Phi^\dagger, \Phi)$ space, that is transformations that preserve the two form $d\Phi^\dagger \wedge d\Phi$. As $\Phi(\boldsymbol{x})$ and $\Phi^\dagger(\boldsymbol{x})$ are canonically conjugated, these correspond to canonical transformations that do not mix variables at different space points. At the infinitesimal level, such transformations can be written in terms of the Poisson bracket with a generating function $\mathcal{F}$

$$\delta\Phi = \partial_{\Phi^\dagger}\mathcal{F}(\Phi^\dagger, \Phi), \qquad \delta\Phi^\dagger = -\partial_\Phi \mathcal{F}(\Phi^\dagger, \Phi), \tag{171}$$

with $\mathcal{F}^\dagger = -\mathcal{F}$ in order to satisfy the consistency condition $\delta\Phi^\dagger = (\delta\Phi)^\dagger$. The vorticity $\partial \Phi^\dagger \wedge \partial \Phi$ is easily seen to be exactly invariant, while for the kinetic term one finds exact invariance up to a time derivative[28]

$$\delta(i\Phi^\dagger \dot{\Phi}) = -i\partial_\Phi \mathcal{F}\dot{\Phi} + i\Phi^\dagger \frac{d}{dt}(\partial_{\Phi^\dagger}\mathcal{F}) = i\frac{d}{dt}\left(-\mathcal{F} + \Phi^\dagger \partial_{\Phi^\dagger}\mathcal{F}\right). \tag{172}$$

---

[28]This result is the field theoretic version of the invariance of $\int p\dot{q}$ under canonical transformations in analytical mechanics. It can also be quickly understood by noticing that the kinetic term involves the integral along the time line of the 1-form $K = \Phi^\dagger d\Phi$, whose external derivative is just the volume 2-form: $dK = d\Phi^\dagger \wedge d\Phi$. As the latter is invariant under volume preserving transformations, the variation $\delta K$ must be closed, $d\delta K = 0$. Since the topology of $\Phi$-space is trivial, the variation is actually exact $\delta K = dG$, for some $G$, and therefore the action is invariant up to boundary terms.

The vast majority of choices for $\mathcal{F}$ leads to non-linear transformations, but the subset $\mathcal{F} = i\alpha\Phi^\dagger\Phi + \beta\Phi^{\dagger 2} - \beta^*\Phi^2$ (with real $\alpha$) leads to a linearly realized $SL(2,R)$ subgroup

$$\delta\Phi = i\alpha\Phi + 2\beta\Phi^\dagger, \qquad \delta\Phi^\dagger = -i\alpha\Phi^\dagger + 2\beta^*\Phi. \tag{173}$$

The constant shift $\Phi \rightarrow \Phi + c$, $\Phi^\dagger \rightarrow \Phi + c^*$, is obviously also a symmetry, generated by $\mathcal{F} = c\Phi^\dagger - c^*\Phi$.

While we arrived at eq. (169) following the thread of our $SU(N)$ rigid body construction, the symmetries we have just identified offer an alternative route. One can indeed convince oneself that eq. (169) with $F(-\partial^2/\Lambda^2) \rightarrow 1$ is the lowest order local action, counting powers of fields and derivatives, that is compatible with eqs. (170) and (172). This result places vorton field theory in the standard framework of effective field theory, according to which the dynamics of systems is constructed on the basis of symmetries and of an expansion in powers of fields and derivatives.

Notice that, in particular, the symmetries forbid a pure gradient term $\partial\Phi^\dagger\partial\Phi$ in the quadratic action. Because of that, the solutions of the linearized classical equations of motion[29] satisfy a trivial dispersion relation $\omega_\mathbf{k} = 0$ for any $\mathbf{k}$. Thus they do not propagate.

We just outlined the situation when treating $\Phi$ and $\Phi^\dagger$ as classical variables. As symmetries underlie the structure of eq. (169), it matters to investigate their fate in the quantum mechanical treatment. In particular, for non-linear symmetries the concern is the path integral measure, or equivalently the UV regulator.[30] In many standard examples of effective field theories based on symmetries and derivative expansion, like the pion Lagrangian of QCD, dimensional regularization offers the most convenient approach, as it generally respects all the symmetries. An exception is represented by the symmetries that rely on the dimensionality of space-time, like chiral symmetry. Unfortunately vorton field theory appears to belong in this class, given the symmetries and the Lagrangian are based on the 2-dimensional Levi-Civita tensor $\epsilon^{IJ}$. Moreover dimensional regularization eliminates all power divergences (or, better, consistently hides them in the definition of composite operators) but in the vorton system a quadratic divergence saturated at the UV cut-off scale $\Lambda$ crucially generates a gradient kinetic term, see eq. (162), which makes quanta propagate, bringing asymptotic states into existence. How should we then proceed in our case? Indeed we already know of one option, which is precisely where we started from: UV-regulate by truncating $S\mathrm{Diff}(T^2)$ to $SU(N)$. Space is consequently latticized and the vorton field $\Phi$ reduced to the discrete set of the $N^2-1$ components of the $SU(N)$ adjoint. The procedure would then amount to repeating the steps we followed in section 6.2 but working with $SU(N)$ rather than directly with $S\mathrm{Diff}(T^2)$. We could then proceed either by path integral or by canonical quantization and study the quantum mechanical fate of the analogue of eq. (171) in the $SU(N)$ model. In the canonical approach, the main effect of quantization is that the ordering of $\Phi$ and $\Phi^\dagger$ matters when considering a generic generating function $\mathcal{F}$. One can however easily see that for the case of *harmonic* generator $\mathcal{F} = f(\Phi) - f(\Phi)^\dagger$ there is no ordering problem given there is no operator product involving both $\Phi$ and $\Phi^\dagger$. This subclass of transformations can then straightforwardly be adapted to the $SU(N)$ regulated theory. We leave the study of the general case to future work.

Another fact of the quantum theory is that symmetries can be spontaneously broken. More precisely, symmetries that act linearly on the classical fields may end up being non-linearly realized in the quantum theory. In our case the linear symmetry subgroup $SL(2,R)$ is indeed spontaneously broken to $U(1)$ in the Fock vacuum. Specifically, the transformations associated with $\beta$ in eq. (173) are broken, while phase rotations $\Phi \rightarrow e^{i\alpha}\Phi$ stay unbroken. An order

---

[29]That is solutions for which $\Phi$ is small enough that the quartic term can be neglected.

[30]In the canonical approach that role is played by operator ordering and by the associated possible UV divergences.

parameter of the breaking is for instance given by the local operator $\Phi\Phi^\dagger$, given one has

$$\langle 0|\,\delta_\beta(\Phi^2)(\boldsymbol{x})\,|0\rangle = \langle 0|\,2\Phi\Phi^\dagger(\boldsymbol{x})\,|0\rangle > 0\,. \tag{174}$$

Indeed the expectation value in the last expression is UV divergent and formally equal to $2\delta^2(\mathbf{0})$. When considering UV regulated quantities, like for instance the Hamiltonian, the relevant equation would entail a UV smeared composite operator of the form

$$\langle 0|\,\delta_\beta\left[\int d^2\mathbf{y}\, f(\mathbf{y})\Phi(\boldsymbol{x}+\mathbf{y}/2)\Phi(\boldsymbol{x}-\mathbf{y}/2)\right]|0\rangle = 2\,\langle 0|\int d^2\mathbf{y}\, f(\mathbf{y})\Phi(\boldsymbol{x}+\mathbf{y}/2)\Phi^\dagger(\boldsymbol{x}-\mathbf{y}/2)\,|0\rangle$$
$$= 2 f(\mathbf{0})\,. \tag{175}$$

For a regulator function $f(\mathbf{y})$ that integrates to unity and is localized over the fluid cut-off length, we have $f(\mathbf{0}) \sim \Lambda^2/4\pi$. Notice that $SL(2,R)$ forbids the addition of an arbitrary gradient term $\partial\Phi^\dagger\partial\Phi$ in eq. (162). Therefore the term that appears after normal ordering and its effects on the spectrum must be controlled by the spontaneous breaking of $SL(2,R)$. A remarkable aspect here is that spontaneous symmetry breaking leads to the appearance of effects that depend on UV physics. It would definitely be interesting to study in detail the regulated Ward identities (for instance using the $SU(N)$ truncation) and explore the role of single and double vorton states in the light of Goldstone theorem. We leave this study to future work.

## 7 Summary

As this is a long and winding paper, we shall here offer a detailed summary.

The initial object of our study has been a dynamical system whose configuration space is $S\mathrm{Diff}(T^2)$, the group of area preserving mappings of the 2-torus $T^2$ onto itself. In practice, given two sets of coordinates $\phi^a$, $a=1,2$, and $x^I$, $I=1,2$ both parametrizing $T^2$, the dynamical variables are given by the time-dependent mappings $\phi \to x(\phi,t)$ subject to the constraint $\det(\partial\phi/\partial x)=1$. With the velocity $v^I = \dot{x}^I(\phi,t)$ and the Lagrangian of eq. (47), the system classically defines the incompressible perfect 2D fluid. The $\phi$'s and the $x$'s, correspond to respectively the *Lagrangian* and the *Eulerian* descriptions of the flow. The goal has been to study the system quantum mechanically.

Our system belongs to the general class of mechanical systems whose configuration space is a Lie group $G$ [30,31]. This class generalizes the notion of rigid body, given the case $G=SO(3)$ coincides with the ordinary rigid body of mechanics. At least when $G$ is finite dimensional, once the irreducible representations of $G$ are known, the steps to derive the quantum description are clear. The first step is dictated by the Peter-Weyl theorem, which establishes that the collection of entries $D^r_{\alpha_r\beta_r}(g(\pi))$ of all irreducible representations offers a complete orthonormal basis of functions on $G$ (see the discussion around eq. (34)). Each irrep $r$ of dimension $d_r$ then corresponds to a Hilbert subspace of dimension $d_r^2$. The action of $G$ on each index of $D^r_{\alpha_r\beta_r}(g(\pi))$ realizes a doubling $G_L \times G_R$ of $G$, with respect to which the subspace transforms like $(r,r)$. This doubling already exists classically. Moreover even though only $G_L$ is exact in general, $G_R$ is crucial to describe the dynamics. Indeed the Hamiltonian is a quadratic form in the generators of $G_R$ (see eq. (69)). The second step in the solution of the quantum theory, is the diagonalization of the Hamiltonian in each separate $(r,r)$ block. Each block then gives $d_r$ generally distinguished eigenvalues, with each eigenvalue $d_r$-degenerate because of $G_L$.

The extension of the above construction to the fluid entails a technical difficulty and a conceptual conundrum. The technical difficulty is that $S\mathrm{Diff}(T^2)$ is an infinite Lie group for which we do not possess ready made explicit irreducible representations. We dealt with this

issue by using that $S\text{Diff}(T^2) = \lim_{N\to\infty} SU(N)$. Working at finite $N$, we then identified a physically sensible way to take the $N \to \infty$ limit.

The conceptual conundrum is tied to the very existence of dual coordinates $\boldsymbol{\phi}$ (Lagrangian) and $\boldsymbol{x}$ (Eulerian), with $S\text{Diff}(T^2)_L$ and $S\text{Diff}_R(T^2)$ acting respectively on the first and on the second. The position space velocity $\boldsymbol{v}(\boldsymbol{x}, t)$, and more precisely its vorticity $\boldsymbol{\nabla} \wedge \boldsymbol{v}$, coincides with the $S\text{Diff}(T^2)_R$ generators and is consequently $S\text{Diff}(T^2)_L$ invariant. As discussed in section 3, this implies that there exist twice as many local degrees of freedom than we can measure with $\boldsymbol{v}(\boldsymbol{x}, t)$ and associated with the internal variables $\boldsymbol{\phi}$. Classically this seems irrelevant, as each $\boldsymbol{\phi}$ configuration lives a life of its own, but not so when considering quantum superpositions: measurements of $\boldsymbol{v}(\boldsymbol{x}, t)$ are then generally described by a density matrix. We considered two alternative attitudes towards this fact (labelled **A** and **B2**). The first is just to accept it. The second is to do away with $S\text{Diff}(T^2)_L$ and to only represent the $S\text{Diff}(T^2)_R$ generators $\boldsymbol{\nabla} \wedge \boldsymbol{v}(\boldsymbol{x}, t)$. The latter choice is equivalent to giving up the group manifold configuration space we started with. Our impression is that this second option is more likely to be concretely realized in a laboratory.

As the energy spectrum and the position space dynamics are determined by the $S\text{Diff}_R(T^2)$ algebra, the above two options only differ in the eigenvalue degeneracy. The study of their dynamics thus technically coincide. Using the truncation $S\text{Diff}_R(T^2) \to SU(N)$, we pursued that study starting in section 4. A crucial consequence of the truncation is the reduction of $T^2$ to an $N \times N$ lattice. The $N^2$ values of the now discretized vorticity on each lattice site provide the $N^2 - 1$ generators of $SU(N)$ (the apparent mismatch of $N^2$ and $N^2 - 1$ is accounted for by the vanishing of the vorticity zero mode on $T^2$). However, the discretized vorticity algebra ($\equiv SU(N)$) approximates $S\text{Diff}(T^2)$ only at wavelengths $\gg \sqrt{N}a$, with $a$ the fundamental lattice length. This led us to identify $\Lambda = 1/\sqrt{N}a$ with the fundamental cut-off of the fluid effective theory and to construct a suitably regulated Hamiltonian (see eq. (120)).

Armed with our construction, the study of the spectrum reduced to computing, irrep by irrep, the eigenvalues of a quadratic form in the $SU(N)$ generators. A detailed study of the fundamental and the adjoint representations allowed us to draw a general picture. Our results are neatly stated in the limit where $N \to \infty$ with $\Lambda$ fixed, in which the lattice becomes a continuum and the volume of $T^2$ goes to infinity like $N/\Lambda^2$. We found that fundamental $\square$ and antifundamental $\bar{\square}$ are fully degenerate and gapped at $\omega_\square \sim \Lambda^4/\rho$. This result, which is fully dictated by dimensional analysis when taking $\Lambda$ and the mass density $\rho$ as the sole inputs, coincides with Landau's estimate for the gap of the transverse hydrodynamic modes. Landau then concluded that the only ungapped mode of quantum hydrodynamics is the longitudinal one, in compliance with the observed properties of superfluids at zero temperature. Our finding however differs from Landau's when considering the states in the adjoint representation. These are not degenerate and satisfy an ungapped dispersion relation that at low momentum takes the quadratic form $\omega(k) = c\Lambda^2 k^2/\rho$. That is the main result of our paper.

The absence of a gap in the adjoint can be qualitatively understood on the basis of group theory and locality. As we show, the $N$ basis states of $\square$ can be chosen to correspond to single vortices localized on each of the $N$ sites of the coarse grained $\sqrt{N} \times \sqrt{N}$ lattice, see Fig. 3. Analogously, the antifundamental $\bar{\square}$ has a basis of localized anti-vortices. Now, $\square \otimes \bar{\square} = \mathbf{Adj} \oplus \mathbf{1}$ and locality imply that the adjoint states are locally indistinguishable from the trivial representation $\mathbf{1}$ when their momentum is sent to zero. By continuity, in this limit their energy must then also vanish like it vanishes for $\mathbf{1}$. This argument can easily be extended to prove the gaplessness of any representation that can be written as a tensor product of an equal number of $\square$ and $\bar{\square}$. That is precisely the subset that trivially realizes the $Z_N$ center of $SU(N)$ and which can equivalently be written as tensor products of adjoints. Indeed, as we prove in section 5.2.2, the low end of the energy spectrum for the tensor product of $q$ adjoints has the form $E = \omega(k_1) + \cdots + \omega(k_q)$ compatible with $q$ quanta that are weakly coupled at long distance

and have the same dispersion relation as the adjoint. We named these quanta *vortons*.

As it is implicit from $\mathbf{Adj} \in \Box \otimes \bar{\Box}$, the vortons are a vortex-antivortex pair. More precisely, for momentum $\mathbf{p}$ they correspond to a pair separated by a distance $\pi\tilde{\mathbf{p}}/\Lambda^2$ (see eq. (114)). In the low momentum regime $p \ll \Lambda$, they can be viewed as point-like objects, given they carry a vorticity dipole smaller than the cut-off length $1/\Lambda$. Instead, for $p \gtrsim \Lambda$, the dipole length exceeds $1/\Lambda$ and the components of the pair can be resolved within the effective theory. In this regime, their dispersion relation also shifts to $\omega \sim (\Lambda^4/\rho) \log p/\Lambda$ which suits expectations for a macroscopic vortex antivortex configuration. Indeed the high momentum ($p \gtrsim \Lambda$) adjoint states correspond to $SU(N)$ generators that do not properly approximate the $S\mathrm{Diff}(T^2)$ algebra. In view of that, we concluded that only vortons with momentum $p \lesssim \Lambda$ should be considered a universal consequence of $S\mathrm{Diff}(T^2)$, while the $p \gtrsim \Lambda$ modes should be discarded as a spurious feature of the truncation. Moreover, as the latter modes feature well separated vortices, we also inferred that $\Box, \bar{\Box}$ and all the states involving well separated vortices should be discarded. In particular this implies that irreps that do not feature vortons should be fully discarded.

That conclusion also fits well our general expectation that states that do not purely involve vortons are gapped. While we did not provide a general mathematical proof, that expectation is based on the analytical and numerical study of various non trivial irreps (see the Appendix) and on the absence of any obvious reason for these states to be ungapped. A deeper perspective would undoubtedly be gained by exploring in which way the gapless vortons are mandated by Goldstone theorem [52].[31] The relevant symmetries would either be $S\mathrm{Diff}_L$ in the original 2D fluid or the transformation of eq. (171) in vorton QFT. We left that study to future work and focused on the study of multi-vorton states.

Our construction, besides implying vortons behave as free particles when infinitely separated, also completely fixes the structure of their interaction up to a proportionality constant. In section 6 we investigated the most basic implications. For that purpose we made specific choices for the structure ($\equiv$ irrep content) of the Hilbert space. Notice the $(r, r)$ blocks of the original perfect fluid system imply (for $N \to \infty$) an infinite number of vorton flavors, given every flux configuration is in turn degenerate under the action of $S\mathrm{Diff}_L$. That is another perspective on the bizarre nature of this system. We thus considered it more plausible to realize $S\mathrm{Diff}_R$ only. Furthermore, motivated by the particle nature of vortons, we realized $S\mathrm{Diff}_R$ through ladder field operators that create and destroy vortons. We provided two examples based on respectively bosonic and fermionic vortons. Within this set up, see section 6.3, we studied $2 \to 2$ scattering at Born level. The amplitude dependence on the kinematical variables is completely fixed by the algebra, with distinctive features that would unmistakably identify these systems experimentally. For instance the scattering amplitude for identical bosons vanishes when the incoming momenta are mutually orthogonal. In the scattering of identical fermions, instead, one finds that two collinear vortons with different energies won't scatter into two vortons with identical energies. All these amplitudes are quadratic in the overall momentum and vanish in the soft limit as it suits hydrodynamic modes. We also found that the Born approximation breaks down for momenta of order $\Lambda$, at which the vortons are therefore strongly coupled. Remarkably, in spite of this fact, our construction is technically consistent also at momenta above $\Lambda$, only its interpretation as a fluid becomes untenable.

# 8  Outlook

We can force ourselves to a synthesis by considering three questions:
Did we solve our original problem? What did we learn? Where do we go from here?

The answer to the first question is: yes, at least in 2D, there exists a consistent quantum

---

[31]We thank Alberto Nicolis for rightly pressuring us on this.

mechanical perfect fluid. This affirmative answer however comes with a qualification that brings us to the second question. In our original incompressible fluid, each velocity/vorticity configuration has an internal infinite degeneracy associated with the action of $S\text{Diff}_L$ on the Lagrangian coordinates of the fluid. We regard this property as suspect. Interestingly, however, we could construct a variety of systems that do not possess the internal infinite degeneracy and yet are endowed with a local vorticity satisfying the quantum mechanical Euler's equation (eq. (71)) of ordinary fluids. We aided ourselves in the construction of these systems with a regulation $S\text{Diff}(T^2) \to SU(N)$, but in the end our results do not depend structurally on the details of the regulation. Our main result is that all these systems feature ungapped quanta, the vortons, whose flow configuration is a vorticity dipole and whose interactions are fully fixed. We thus labelled vorton QFT the simplest and perhaps most plausible incarnations. The vortons sharply distinguish our systems from superfluids, where all modes carrying vorticity are gapped.

The obvious goal at this point is to try and find out if vorton QFT can be realized experimentally. A first issue to investigate towards that goal concerns the robustness of the symmetries at the basis of vorton QFT. A preliminary study was undertaken in section 6.4, though a more thorough study is warranted. In particular one should understand the occurrence on the ungapped vortons according to the finite density analogue of Goldstone's theorem. In parallel, it also seems mandatory to explore the relation between vorton QFT and the quantum Hall system, where occupied Landau levels do behave like a sort of incompressible fluid. An even simpler but related system to consider is 2D electrodynamics in a constant magnetic field, which is dually equivalent to a 2D superfluid. As charged particles are dual to vortices, particle-antiparticle bound states seem the natural candidates to reproduce the dynamics of vortons. Indeed a preliminary study [52] indicates that is the case, but only up to contact interaction terms and up to the obvious constant gap in energy associated with the rest mass of particle and antiparticle. These facts further motivate the study of the robustness of the symmetries that constrain vorton QFT.

If the above program ends up consolidating vorton QFT, it will then be interesting to consider further properties of its dynamics. One particularly interesting issue is the semiclassical limit of the flow, which should arise for sufficiently large vorton density. One is easily convinced that the critical request is that the flow velocity collectively generated by the vortons is larger than their individual velocity. A simple estimate shows this requires vorton densities $\gg \Lambda^2$. It would be interesting to study quantum effects by expanding around such semiclassical flows. In fact that brings to mind the possible use of our insights in the effective description of fluids, like those occurring in finite temperature QFT, that do not correspond to QFTs around a vacuum (see e.g. [56]). At this stage we have no concrete idea to offer, but a fresh look at those systems with our results in mind may teach us something. The extension of our analysis to 3D fluids is also in our mind, even though it appears a long shot at this moment.

In the end, we have as many questions as when we started. Luckily they are different.

# Acknowledgments

We would like to thank Alexander Abanov, Andrea Cappelli, Vadim Cheianov, Gabriel Cuomo, Sergey Dubovsky, Angelo Esposito, Fanny Eustachon, Eren Firat, Ben Gripaios, Brian Henning, Alexander Monin, Alberto Nicolis, Riccardo Penco, Federico Piazza, Tudor Ratiu, Slava Rychkov and George Savvidy for insightful dicussions and for honest criticism. RR offers one very special thank to Solomon Endlich, Walter Goldberger and Ira Rothstein for the hard work and endless debates that animated the abandoned project upon which much of the present paper was based.

**Funding information** R.R. is partially supported by the Swiss National Science Foundation under contract 200020-188671 and through the National Center of Competence in Research SwissMAP.

# A  Energy spectrum beyond fundamental states

Obtaining an analytic formula for the spectrum of representations beyond the fundamental and the adjoint is a challenging task. However, there are still interesting features of the spectrum that one can derive for an arbitrary representation. In what follows we discuss two such results: the $N$-fold degeneracy associated with most of the energy eigenstates and the statistics of the energy eigenvalues. The latter will allow us to put a lower bound on the number of vortices in a gapless state. At the end of the section we describe the results of the numerical diagonalisation of the Hamiltonian on two-vortex states.

## A.1   $N$-fold degeneracy of the states

The first interesting feature of the spectrum is that for most $SU(N)$ representations all energy eigenstates are $N$-fold degenerate, as we have seen for the fundamental. To see this, let us consider the products of $n$ fundamental states, the $n$-vortex states. The basis states $|\alpha_1, \ldots, \alpha_n\rangle \equiv |\alpha_1\rangle \otimes \cdots \otimes |\alpha_n\rangle$ are eigenstates of the elementary translations $T_2$ of eq. (87) with eigenvalues $\omega^{\sum_i \alpha_i} \equiv e^{-ip_2 a}$. The second component of the momentum $p_2 = -\frac{1}{r}(\sum_i \alpha_i \mod N)$ is periodic and takes $N$ different integer values in units of $\frac{1}{r}$, which we chose to be in the range $-\frac{1}{2}(N-1) \le p_2 r \le \frac{1}{2}(N-1)$. The space of all $n$-vortex states can be split into $N$ subspaces of fixed $p_2$ value. Since the Hamiltonian of eq. (124) commutes with $T_2$ it does not mix the subspaces with different values of $p_2$ and takes a block-diagonal form:

$$\langle p_2', \mu' | H | p_2, \mu \rangle = \delta_{p_2}^{p_2'} H_{\mu', \mu}^{(p_2)}. \tag{A.1}$$

Here the $\mu$ label distinguishes among the states with fixed $p_2$. The other translation acts on the basis states as $T_1 |\alpha_1, \ldots, \alpha_n\rangle = |\alpha_1 + 1 \mod N, \ldots, \alpha_n + 1 \mod N\rangle$ and maps each fixed $p_2$ subspace to another one with $p_2$ reduced by $n \mod N$ in units of $\frac{1}{r}$. If $n$ and $N$ do not have any non-trivial common divisors then shifting by $n \mod N$ runs all $N$ different values of $p_2$. The Hamiltonian also commutes with $T_1$, which means that all $N$ blocks in the Hamiltonian are identical and do not depend on the value of $p_2$. This introduces an $N$-fold degeneracy of energy eigenstates that is labeled by $p_2$. If the greatest common divisor of $n$ and $N$ is larger than one, then the $N$ $p_2$ blocks in the Hamiltonian are split into $\gcd(n, N)$ equal subsets connected by the action of $T_1$. The degeneracy of such states is smaller and is given by $\frac{N}{\gcd(n,N)}$.[32]

We have seen before that for the fundamental representation, one-vortex, this degeneracy fixes the spectrum completely. For two-vortex states, the degeneracy is also $N$, since we take $N$ to be odd. Then the space of $N^2$ two-vortex states splits in $N$ subspaces of $N$ states each, which can be further reduced to $(N-1)/2$ antisymmetric and $(N+1)/2$ symmetric states. We shall discuss the results of numerical diagonalisation of the Hamiltonian on the two-vortex states at the end of this section. Since the adjoint states are made of $N$ fundamental vortices, we expect no large $N$ degeneracy in the energy eigenvalues given $gcd(N, N) = N$. This

---

[32]In the magnetic field analogy the $n$-vortex state behaves like a particle of charge $n$ that lives in a magnetic field. The flux per plaquette $\Phi = 2\pi n/Ne$ is such that $T_1 T_2 = e^{-ie\Phi} T_2 T_1$, which holds for the translations acting on the $n$-vortex states. Making the parallel with the quantum Hall literature [57] we have $\Phi = \frac{n}{N}\Phi_0$, with $\Phi_0$ the fundamental flux. If $n$ and $N$ share a common divisor we have to consider $N/\gcd(n, N)$ instead of $N$, such that the spectrum admits an $N/\gcd(n, N)$ degeneracy. The basis states are eigenstates of $T_2$ with $-\frac{1}{2}(N-1) \le p_2 r \le \frac{1}{2}(N-1)$ and are eigenstates of $T_1^{N/\gcd(n,N)}$ such that $p_1$ is confined to take one of $\gcd(n, N)$ values. This defines the magnetic Brillouin zone.

matches the absence of large degeneracies observed in the analytic expression of the spectrum in eq. (129). The same is true for any other representation in the tensor product of any multiple of $N$ fundamentals, for which no degeneracy is required by translational invariance.

## A.2 Statistics of the energy eigenvalues

More information on the spectrum can be obtained from the statistics of energy eigenvalues. Indeed, it is easy to obtain a mean energy over all $n$-vortex states. Starting from the eq. (124), and using the same regulator as in eq. (130), the Hamiltonian on these states takes the form

$$\langle \alpha_1, \dots, \alpha_n | H | \beta_1, \dots, \beta_n \rangle$$
$$= \frac{\Lambda^4}{32\pi^4\rho} \sum_{\mathbf{n}\neq\mathbf{0}}^{|n^I|\leq\frac{1}{2}(\sqrt{N}-1)} \frac{1}{\mathbf{m}^2} \left( n\,\delta^{\alpha_1}_{\beta_1}\cdots\delta^{\alpha_n}_{\beta_n} + \sum_{i\neq j}^{n}(J^{\dagger}_{\mathbf{m}})^{\alpha_i}_{\beta_i}(J_{\mathbf{m}})^{\alpha_j}_{\beta_j}\overbrace{\delta^{\alpha_1}_{\beta_1}\cdots\delta^{\alpha_n}_{\beta_n}}^{\text{no }i,j} \right). \quad \text{(A.2)}$$

Since the $J_{\mathbf{m}}$ matrices are traceless only the first term contributes to the trace of the Hamiltonian over all $n$-vortices states:

$$\sum_{\alpha_1,\dots,\alpha_n} \langle \alpha_1, \dots, \alpha_n | H | \alpha_1, \dots, \alpha_n \rangle = N^n\, n\, \frac{\Lambda^4}{32\pi^4\rho} \sum_{\mathbf{n}\neq\mathbf{0}}^{|n^I|\leq\frac{1}{2}(\sqrt{N}-1)} \frac{1}{\mathbf{m}^2} = N^n\, n\, E_\square. \quad \text{(A.3)}$$

Since there are $N^n$ $n$-vortex states the mean energy is $n$ times the energy of a single vortex and is thus growing logarithmically with $N$ in the continuum limit. In order to place bounds on the energy eigenvalues, we have to further look into the maximal deviation from the mean, hoping this way to identify the gapped representations. In order to implement such bounds let us consider traces of higher powers of the Hamiltonian or, rather, of higher powers of the deviation of the Hamiltonian from its mean $\delta H = H - nE_\square \mathbb{1}$. The latter is simply given by the second term in eq. (A.2). The quadratic dispersion of the energy eigenvalues is given by

$$\overline{\delta H^2} \equiv N^{-n} \sum_{\substack{\alpha_1,\dots,\alpha_n \\ \beta_1,\dots,\beta_n}} \langle \alpha_1, \dots, \alpha_n | \delta H | \beta_1, \dots, \beta_n \rangle \langle \beta_1, \dots, \beta_n | \delta H | \alpha_1, \dots, \alpha_n \rangle$$

$$= N^{-n} \left(\frac{\Lambda^4}{32\pi^4\rho}\right)^2 \sum_{\substack{\alpha_1,\dots,\alpha_n \\ \beta_1,\dots,\beta_n}} \sum_{\mathbf{n},\mathbf{m}\neq\mathbf{0}}^{\sqrt{N}} \frac{1}{\mathbf{m}^2\,\mathbf{n}^2} \sum_{\substack{i\neq j \\ i'\neq j'}}^{n} (J^{\dagger}_{\mathbf{m}})^{\alpha_i}_{\beta_i}(J_{\mathbf{m}})^{\alpha_j}_{\beta_j}(J^{\dagger}_{\mathbf{n}})^{\beta_{i'}}_{\alpha_{i'}}(J_{\mathbf{n}})^{\beta_{j'}}_{\alpha_{j'}} \overbrace{\delta^{\alpha_1}_{\beta_1}\cdots\delta^{\alpha_n}_{\beta_n}}^{\text{no }i,j}\overbrace{\delta^{\beta_1}_{\alpha_1}\cdots\delta^{\beta_n}_{\alpha_n}}^{\text{no }i',j'}$$

$$= 2n(n-1)\left(\frac{\Lambda^4}{32\pi^4\rho}\right)^2 \sum_{\mathbf{m}\neq\mathbf{0}}^{\sqrt{N}} \frac{1}{\mathbf{m}^4}, \quad \text{(A.4)}$$

where we have used a shorthand to indicate the endpoints of the sum over $\mathbf{n}$. After taking the traces only the terms with $i = i'$, $j = j'$ or $i = j'$, $j = i'$ survive and all give equal contributions proportional to a momentum conservation $\delta(\mathbf{m} + \mathbf{n})$ or $\delta(\mathbf{m} - \mathbf{n})$ respectively. The remaining momentum sum is converging and does not depend on $N$ in the continuum limit. The fact that energy dispersion remains finite while the mean is growing with $N$ is not enough to prove the absence of the gapless modes since the total number of eigenvalues is growing as $N^n$, which is much faster than the mean. Eq. (A.4) only implies a conservative bound on the maximal deviation of the energy eigenvalues from the mean, $\delta H^2_{\max} \leq N^n\,\overline{\delta H^2}$, or

$$\frac{\delta H_{\max}}{\overline{H}} \leq const \cdot \sqrt{\frac{n-1}{n}} \frac{N^{n/2}}{\ln N}. \quad \text{(A.5)}$$

Since the ratio on the right hand side is growing with $N$ we cannot constrain the smallest energy eigenvalue at large $N$. The inequality is saturated if just one of the $N^n$ eigenvalues deviates maximally from the mean while the rest are exactly equal to the mean, which is not expected to represent the actual spectrum. The $N$-fold degeneracy discussed in the previous section would soften the bound slightly, replacing $N^{n/2} \to N^{(n-1)/2}$ if the spectrum were precisely $N$ times degenerate as it is for $n = 2$.

In order to obtain a useful constraint one should consider the trace of higher powers of $\overline{\delta H}$. The maximal deviation of the eigenvalues from the mean can be constrained as $\delta H_{\max}^k \leq N^n \overline{\delta H^k}$, and choosing $k \gtrsim n \ln N$ should be enough to counter the growth in the dimension of the $n$-vortex subspace. For a general $k$ we propose a very crude estimate of $\overline{\delta H^k}$. Starting from eq. (A.2), we see that every term in the momentum sum in $\delta H$ is in turn a sum of $n(n-1)$ terms corresponding to the choice of two different indices out of $n$. After taking the trace over $\delta H^k$, we end up with a product of at least one and at most $k$ momentum conserving $\delta$-symbols. For instance at $k = 3$ we expect momentum conservation contributions $\delta(\mathbf{m}_1 + \mathbf{m}_2)\delta(\mathbf{m}_1 - \mathbf{m}_3)\delta(\mathbf{m}_2 - \mathbf{m}_3)$ or $\delta(\mathbf{m}_1 + \mathbf{m}_2 + \mathbf{m}_3)$. Since every momentum label is included in at least one momentum conservation $\delta$, the momentum sums will converge at large momenta with a result independent of $N$ in the large $N$ limit. The momentum sums can thus grow at most as $const^k$ with the constant independent of both $N$ and $n$. We can therefore place a conservative upper bound on the trace of $\delta H^k$ as

$$\overline{\delta H^k} \leq \left( \frac{\Lambda^4}{32\pi^4 \rho} \right)^k const^k\, n^{2k}\,. \tag{A.6}$$

It leads to the following constraint on the maximal relative deviation of the energy eigenvalues from the mean:

$$\frac{\delta H_{\max}}{\overline{H}} \leq const \cdot n\, \frac{N^{n/k}}{\ln N} \xrightarrow[k\to\infty]{} const\, \frac{n}{\ln N}\,. \tag{A.7}$$

The right hand side monotonically decreases as $k$ grows. Since the bound is valid for any $k$ we can optimise it by taking the limit of large $k$, or $\frac{k}{n\ln N} \to \infty$ to be precise. The latter bound implies that the maximal relative deviation from the mean energy for the states with a fixed number of vortices $n$ is vanishing in the continuum limit and there could not be any gapless states unless the number of vortices grows as well, at least as $n \gtrsim \ln N$. In particular, no states with a finite number of vortices can be gapless in the continuum limit. Unfortunately, the bound is not strong enough to imply that the number of elementary vortices in a gapless state should be at least $N$.

## A.3 Numerical diagonalisation: Two-vortex and vortex-anti-vortex states

To confirm our theoretical prediction on the gapped nature of the 2-vortex spectrum, we here numerically diagonalise the Hamiltonian at fixed $N$. For the 2-vortex state, the $N$ fold degeneracy actually reduces the problem to a diagonalisation of a simple $N \times N$ matrix, as opposed to $N^2 \times N^2$. In particular we can focus on states at a fixed momentum value $p_2$ and diagonalise the Hamiltonian in this subspace: given all subspaces with fixed $p_2$ have identical spectrum, this will suffice. In practice we considered the states $|\alpha, -\alpha\rangle$ that correspond to $p_2 = 0$. The numerical results are displayed in Fig. 7 where we plot the distribution of the eigenvalues for $N = 91^2$. The eigenvalues are concentrated around $\overline{H} = 2E_\square$, that is twice the energy of a single fundamental vortex. This agrees with the result in eq. (A.3) for the mean energy of a 2-vortex state. There is a sharp fall off of the eigenvalue density at some finite value below $2E_\square$, which hints at the fact that there are no gapless states in the large $N$ limit. This matches our expectation.

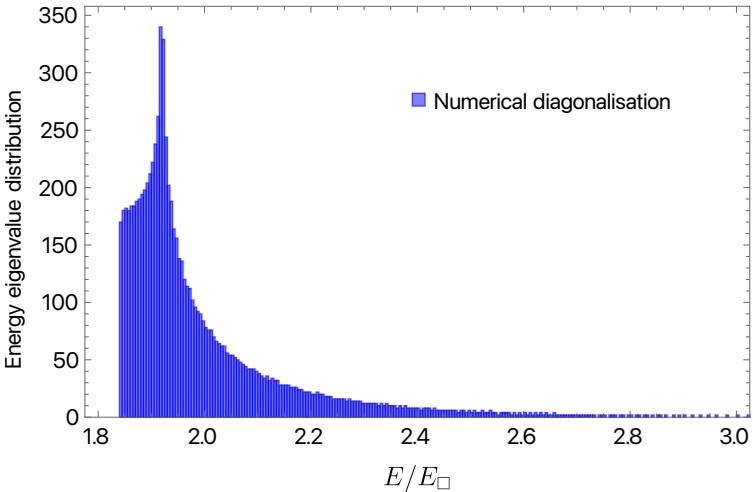

Figure 7: Energy eigenvalue distribution functions of two-vortex states for $N = 91^2$ obtained by exact numerical diagonalisation. Energy is measured in the units of single vortex energy. The $N$ fold degeneracy of the spectrum is not represented.

The numerical results can be well understood if we model the 2-vortex state using a Coulomb interaction between vortices localized on a single fluid cell. The energy of a state with one vortex located at the origin and another at the position $\boldsymbol{x}$ is written as $E = 2E_\square + E_{int}(\boldsymbol{x})$. The interaction energy is associated with the second term of eq. (A.2) and depends only on the relative separation between the vortices. This translation invariance is identified with the aforementioned $N$-fold degeneracy in the 2-vortex spectrum. The configurations with nearby vortices will correspond to the highest energy. The maximum energy, $E = 4E_\square$, is reached for superposed vortices which act as a single vortex with twice the vorticity. At larger separation, the energy the vortex pair decreases, while the number of allowed configurations goes up. That explains the fall-off towards large energies of the energy eigenvalue density observed in Fig. 7. For very large separations the picture changes as the number of available states decreases again. Namely the points on the torus with $\pi r \leq |\boldsymbol{x}| \leq \sqrt{2}\pi r$ are found only in the corners of the square $|x^I| \leq \pi r$. The minimal energy corresponds to the maximal possible vortex separation $|\boldsymbol{x}| = \sqrt{2}\pi r$. The corresponding interaction energy, even if negative, does not feature a logarithmic enhancement and thus cannot compensate for the free part $2E_\square$. In the continuum limit the gap of the 2-vortex states is logarithmically large. More precisely the majority of the states has energy just below $2E_\square$ and corresponds to vortices separated by large distances $|\boldsymbol{x}| \sim r$.

The same logic can be used to explain the spectrum of adjoint states of eq. (129) represented here in Fig. 8. Indeed, an adjoint state is a product of a fundamental and anti-fundamental state, i.e., a vortex–anti-vortex pair. Given that the vorticity field of an anti-vortex is negative, the interaction energy is now precisely the opposite of the 2-vortex case. The interpretation is now different however, as the naive Coulomb model predicts an $N$ fold degeneracy that is not present in the adjoint spectrum. In particular the adjoint state energy of eq. (129) depends on $\mathbf{m}$, defined on the finer lattice, whereas the Coulomb model for the two vortex state assumed localization over the coarser lattice. The naive model, however, can explain the main features of the adjoint states spectrum. The minimum energy is zero and it is attained when the vortex and anti-vortex are on top of each other. The corresponding configuration has vanishing vorticity and is just the vacuum state. The rest of the $N^2 - 1$ states belong to the adjoint representation and their energy spectrum has a gap that vanishes in the continuum

limit. The low lying modes with the quadratic dispersion relation of eq. (132) correspond to the tail of the interaction energy distribution function. The peak is centered slightly above $2E_\square$ and corresponds to adjoint modes associated with large momenta and an energy with a logarithmic momentum dependence.

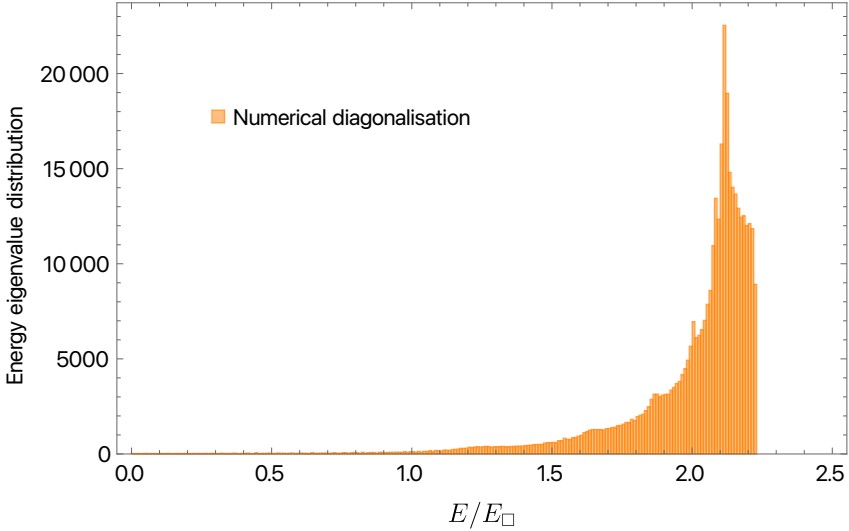

Figure 8: Energy eigenvalue distribution function of adjoint states for $N = 25^2$ from eq. (129). Energy is measured in the units of single vortex energy. The absence of $N$ fold degeneracy in the spectrum implies that it is more expensive to compute than for the two vortex state.

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
