# Peer review of "The Quantum Perfect Fluid in 2D"

_SciPost Physics, doi:SciPost Phys. 17, 019 (2024)_

## Round 1 · Author Response

Dear Editor and Referees,

We thank the editor and referees for a careful reading of the manuscript, interesting questions and comments, and we apologize for the extended delay in our response. We will respond promptly to any further comments. In the following, we go over the comments made by the referees, address them, and indicate any changes made to the manuscript in the "List of changes" section.

Sincerely,

Aurélien Dersy, on behalf of the authors.

Referee 1: 1) We corrected the typo. 2 ) We comment on the comparison between our results and those of ref.28 in p.28, adding a new footnote 19. In ref.28 the correlators of n-pt functions are computed on the vacuum and found to be non-vanishing. In contrast, our ground state, the singlet representation, is characterized by a vanishing of all the charges and velocity correlators.

Referee 2: 1) Response to flat band remark.We think that the situation we are facing in the quantum fluid does not resemble that of the Landau levels. First of all, as explained in the long paragraph below eq.(9), $\omega=0$, for the transverse modes, is a {\it{classical}} dispersion relation. {\it{Quantum mechanically}} it implies that the states are fully delocalized in field space so that the energy levels cannot be computed by the standard methods of weakly coupled field theory. In other words, to start, we do not even know what the quantum mechanical energy levels are! The goal of the paper is precisely to find out what these levels are. The case of Landau levels is different in that, first of all, the quantum mechanical energy levels are known. Moreover on a finite area, they happen to have a finite degeneracy, which, from a UV perspective, is only a mild degeneracy: while the ground state modes do not even have definite 2-momentum, for very excited modes we can treat the magnetic field as a small perturbation and construct states that have approximately definite 2-momentum and energy growing quadratically. Notice in passing that, as we explain in the paper, the $N$ states of the $SU(N)$ fundamental in our quantum fluid are fully analogue to the set of lowest Landau levels in a system with area $N$ in unit of the magnetic flux. As concerns perturbing the $\omega=0$ classical dispersion relation, notice that this is discussed at the bottom of page 9, where we recall the results of ref. 20: the degeneracy of the classical dispersion relation can be lifted by perturbing the lagrangian with terms that violate the internal volume preserving symmetry but this results in a completely different system. In view of all the above, we believe the addition of a comment on Landau levels would not add clarity. 2) Response to the heat capacity remark: When dealing with the doubling of the degrees of freedom we discuss two alternatives: option A and option B2. Indeed the entropy (and heat capacity) of option A is twice that of option B2. We view this doubling as a feature that distinguishes the two options. We believe only a concrete realization of the quantum fluid can select between them.

Referee 3:
1) Regarding membranes: We thank the referee for directing us towards the original works on relativistic 2D membranes. We have included further references and outlined the main differences, as we see them, between our work and the preceding one. As a matter of fact, the only paper that played a role in the development of our project is the one by Pope and Stelle of 1989, where the $S_{diff}\sim \lim_{N\to \infty} SU(N)$ relation is neatly derived. However, as that paper was certainly influenced by preceding literature, we now asked ourselves to what extent our work might have been influenced by previous work and/or overlapped with it. So we scouted the literature looking for work on connecting $S_{diff}$, $SU(N)$, the relativistic membrane and the incompressible fluid. Honestly, we must admit that we have not been able to find clear statements, or at least clear enough for us to understand them. Indeed we do not even fully understand the connection between the relativistic membrane and the incompressible non-relativistic fluid. Certainly the equivalence does not apply to the domain wall, that is the infinite extended 2+1-brane embedded in Minkowski space. That system should be viewed as an effective field theory with a cut-off scale controlled by the tension, and consisting at low energy of a single derivatively coupled massless "branon" excitation, as discussed for instance in doi:10.1103/PhysRevD.59.085009. So, if anything, the equivalence may apply to the study of the "finite size" quantum fluctuating 2-membrane, open or closed, where $S_{diff}$ surfaces as a residual gauge symmetry when working in light cone gauge (doi:10.1088/0264-9381/7/1/015). In that context, the closest we got to finding a connection with the non-relativistic fluid is the paper by Bordermann and Hoppe of 1993 (doi:10.1016/0370-2693(93)91002-5), where however what emerges is an irrotational fluid, with no vorticity. Nowhere were we able to find a concrete result for an energy spectrum which we could compare to. We believe the main rift separating our work from all the preceding one stems from our effective field theory perspective (which we also think is the only one that can be meaningfully taken for both fluids and membranes). A consequence of that is that we are forced to interpret the relation $S_{diff}\sim SU(N)$ modulo a truncation (realized through the choice of a class of Hamiltonians) that preserves only the relevant low momentum tail ($|\mathbf{n}| \lsim \sqrt N$) of the algebra, where the structure constants approximately coincide. That is essential in the derivation of all our results, in particular the spectrum. Fortunately the gross details of our results, in particular the vorton dispersion relation are robust and apply over a broad range of Hamiltonians. Whether that can make contact with a real world system is another question, which we honestly have not yet been able to address, and which motivates us to explore further. We have not found any trace of the necessity for a truncation in that previous literature. We track that fact to the lack of an EFT perspective.
2) Regarding the additional UV regularization : The $SU(N)$ regulation of $S_{diff}$ is conceptually important, in that it clarifies that the UV can be regulated by simply deforming the symmetry, rather than breaking it by hand. In other words, the reduction to a rigid body with $SU(N)$ internal symmetry and a particular inertia tensor (a class of them indeed) gives us confidence that we are not missing anything. But the $SU(N)$ and $S_{diff}$ algebra only coincide in the infrared tail and thus an additional regulation step is needed to ensure that Euler's equations for the low momentum modes are respected. It is interesting that the embedding of $SU(N)$ in $S_{diff}$ gives rise to a double lattice structure, a fine and a coarse one, with the latter providing the natural physical cut-off of the fluid EFT.We comment on this point in the main text further and contrast our approach with other works using the truncation of $S$Diff without an additional regularization. Also, the additional states present in the $SU(N)$ theory, isolated gapped vortices like one can find in a superfluid, while not part of the fluid EFT, still illuminate by contrast the peculiar nature of the vortons. But finally, and most importantly, as made clear in the study of vorton scattering, the EFT is strongly coupled at the scale $\Lambda$, which makes the computation of the vorton dispersion relation in the continuum EFT ambiguous. The existence of an explicit class of $SU(N)$ regulated theories where everything can be computed explicitly is a reassuring existence proof. Once that is granted, one can work in the continuum for most things. Yet again, the very fact that the existence of the asymptotic vorton states relies on effects at a UV cut-off where the coupling is irremediably strong is a measure of the unconventionality of the fluid EFT. Of course, the best way to sort out any possible doubts would be to find a concrete realization.
3) Regarding the UV cutoff: The decoupling of the charges $\tilde{R}_{\mathbf{m}}$ with $|\mathbf{m}|\geq \sqrt{N}$ is indeed only guaranteed once we assign infinite inertia to the UV modes. Failing to provide that constraint yields a theory which cannot be identified with quantum hydrodynamics anymore. The validity of this (unconventional) EFT construction itself depends on the way that the cutoff is implemented. For a suitable family of cutoffs we are left with a proper quantum fluid theory, where the modes satisfy Euler's equation and where the vortons have a spectrum quadratic in momentum. In particular, as long as $F(\mathbf{n})\rightarrow 0$ for $|\mathbf{n}|\geq \sqrt{N}$, then any regulator function $F(\mathbf{n})$ is suitable and the main features of the IR physics (such as the quadratic dispersion relation) do not depend on the precise choice of the regulator. We comment on the extension to cutoffs with $F(\mathbf{n})\rightarrow 0$ for $|\mathbf{n}|\geq C \sqrt{N}$ in the main text.
4) Regarding fractons: We thank the referee for pointing out to us the potential connection of our model with exotic fracton phases of matter. Fractons are non-propagating quasi-particles, where a single quasiparticle, in isolation, is unable to move. Only by forming bound states such as dipoles, can a fracton move (see https://doi.org/10.1142/S0217751X20300033). This constraint comes from a set of higher moment conservation laws, such as multiple moments of charge density. In our case, this conservation law relates to the different moments of vorticity. It seems that, in our model, the states we found in the fundamental representation are akin to the non-propagating fractons, while the vortons are similar to the fracton bound states, moving in a direction perpendicular to their dipole moment. However, we have not gone beyond this superficial connection and leave the rigorous exploration of the analogy for future work.
5) We refer to our answer to the Referee 1 for our treatment of Ref.28
6) Regarding multi-vorton states: What is peculiar about our construction is that all the left charges annihilate the ground state, like for a rigid body. Thus all correlators vanish on the ground state and in particular there is no 2-vorton cut. Correspondingly to prepare multi adjoint (vorton) states we must act on the ground state with operators that are not L-invariant. The situation is the same as for the rigid body: in order to set it in motion starting from its ground state one needs to kick it in some specific point, that is one needs an external source coupled to the Euler angles, which are not L-invariant. It seems that is the way things unavoidably are for the rigid body and all its generalizations. Perhaps a different perspective on this can be obtained by starting directly with the vorton EFT. This EFT, a fact which we did not yet fully understand at the completion of this paper, can be directly derived from the Clebsch variable formulation of the fluid (shown for instance in doi:10.1088/0305-4470/37/42/R01). In that formulation, the original left $S_{diff}$ is replaced by another, inequivalent, $S_{diff}$. Amusingly, as we are in the process of exploring, also this other formulation lends itself to an $SU(N)$ regulation, but one which is not quite equivalent to a rigid body. We hope the exploration of this other approach may help us develop a more satisfactory picture.
7) Regarding the extension to other 2D manifolds: In this paper we have only restricted ourselves to the 2D manifold consisting of a square torus. As pointed out in another comment, embeddings of $S$Diff exist for other 2D surfaces, where different Lie algebras can provide suitable finite truncations. Suppose we focused on other topologically equivalent tori, realised by picking different lattices. In that case, we do not expect our results to change much, as the algebra for $S$Diff will be isomorphic for all these tori (see 10.1088/0264-9381/7/1/015). Provided we restrict the rigid body to modes below our cutoff $\Lambda$ it seems reasonable to expect that the spectrum of the adjoint would similarly contain gapless excitations. For this feature to emerge in Eq.(130) we only needed this cutoff to be imposed, along with a sum over $\mathbf{n}$ of the form $\frac{(\mathbf{m}\wedge \mathbf{n})}{\mathbf{n}^2}$, which is generic for any suitable approximation of $S$Diff. For other 2D surfaces, like the Klein bottle, the algebra of $S$Diff is modified in such a way that the usual adjoint representation does not diagonalize the rigid body Hamiltonian. However, after diagonalization we would expect similar gapless vorton excitations. We have not studied the case of a compact surface with a boundary, leaving it for future work and indeed do not discard the possibility of finding either vortex or anti-vortex edge states.

---

## Round 1 · List of Changes

1) We correct the sentence on p.7 starting with "That is because time independent transverse modes..." 2) On p.8 (last paragraph before 1.3) we reformulate our comparison to ref.28, starting from "Indeed the results of this paper, which ..." 3) In the second paragraph of section 4 (p.22) we refer the reader to some of the original works regarding quantized membranes (refs 41, 42, 43) and discuss the usage of the finite truncation of $S$Diff there (starting at "This realization was made in..."). We mention the extension of the finite truncation construction to other 2D surfaces in a new footnote 14 on p.22, referring to ref 41. 4) On p.22 we add references in a new footnote 15 discussing the usage of the finite truncation construction in classical statistical mechanics (refs 44,45), membranes (ref 46) and quantum complexity (ref 47). The footnote extends to p.23. 5) At the top of p.24 (paragraph after eq.79, starting with "The fact that the commutation relations...") and at the top of p.37 (paragraph after eq.121 starting with "This separates our work from...") we add comments regarding the $|\mathbf n| \ll \sqrt{N}$ limit which emerges as a consequence of our EFT perspective, setting our work apart from the original literature discussing the finite truncation of $S$Diff. 6) On p.28 (end of the last paragraph before section 5.1.1, starting with "This is to be contrasted...") we contrast our results with ref 28. The comparison is made in detail in a new footnote 19. 7) On p.40, in the second to last paragraph (starting with "In particular, pushing the") we add a comment regarding the choice of cutoff taken.

---

## Editorial Decision

published